# THE IMPLICIT BIAS OF STOCHASTIC ADAGRAD-NORM ON SEPARABLE DATA

## ABSTRACT

This paper explores stochastic adaptive gradient descent, i.e., stochastic AdaGrad-Norm, with applications to linearly separable data sets. For the stochastic AdaGrad-Norm equipped with a wide range of sampling noise, we demonstrate its almost surely convergence result to the $\mathcal{L}^2$ max-margin solution. This means that stochastic AdaGrad-Norm has an implicit bias that yields good generalization, even without regularization terms. We show that the convergence rate of the direction is $o(1/\ln^{\frac{1-\epsilon}{2}} n)$. Our approach takes a novel stance by explicitly characterizing the $\mathcal{L}^2$ max-margin direction. By doing so, we overcome the challenge that arises from the dependency between the stepsize and the gradient, and also address the limitations in the traditional AdaGrad-Norm analysis.

## 1 INTRODUCTION

With the growth of computing power in recent years, various models like neural networks have gained the ability to perfectly fit training data. These models, exceeding the data's capacity, are referred to as over-parametrized models. Over-parametrized models often exhibit numerous global optimums, yielding a zero training loss, yet exhibiting substantial disparities in test performance (Wu et al., 2018; Chatterji et al., 2022). Fascinatingly, investigations have indicated that optimization algorithms tend to converge towards those optimal points associated with a good generalization (Zhang et al., 2021). This intriguing phenomenon is referred to as the implicit bias of optimizers and is widely speculated to exist (Neyshabur et al., 2014; Zhang et al., 2005; Keskar et al., 2017; Wilson et al., 2017).

Evidence of implicit bias has been established under different settings. For the linear classification task with cross-entropy loss, Soudry et al. (2018) demonstrate that gradient descent (GD) converges to the $\mathcal{L}^2$ max-margin solution. This solution is also called the hard support vector machine (hard SVM) solution, which is commonly known. This revelation underscores that even fundamental optimizers like GD have an implicit bias. Subsequent endeavors have extended their work, adapting GD into stochastic gradient descent (SGD), momentum-based SGD (mSGD), and deterministic adaptive gradient descent (AdaGrad-Diagonal) (Gunasekar et al. (2018); Qian & Qian (2019); Wang et al. (2021b;a); Wu et al. (2021)). However, to the best of our knowledge, there is no work that proves the existence of implicit bias in the stochastic AdaGrad-Norm method. It is worth doing this since this method is widely used in most of the practical systems Duchi et al. (2010); Streeter & Mcmahan (2010); Lacroix et al. (2018), machine learning, and so on.

The iterates generated by the stochastic AdaGrad-Norm method enjoy the following dynamics (see Streeter & Mcmahan (2010); Ward et al. (2020)):

$$S_n = S_{n-1} + \left\| \nabla g(\theta_n, \xi_n) \right\|^2, \quad \theta_{n+1} = \theta_n - \frac{\alpha_0}{\sqrt{S_n}} \nabla g(\theta_n, \xi_n), \tag{1}$$

where $g(\theta)$ refers to the objective function, $\nabla g(\theta, \xi_n)$ is an unbiased estimation of the gradient $\nabla g(\theta)$ with $\{\xi_n\}$ being mutual independent. $S_n$ is the cumulative stochastic gradient norm, and $\alpha_0 > 0$ represents the constant step size. We define a $\sigma$-filtration $\mathcal{F}_n := \sigma\{\theta_1, \xi_1, \xi_2, \dots, \xi_{n-1}\}$. A critical question then arises:

*Can stochastic AdaGrad-Norm converge to the $\mathcal{L}^2$ max-margin solution?*

If the answer is true, we can show that stochastic AdaGrad-Norm has an implicit bias.

**Formulation of the convergence**   We investigate the linear classification problem with linearly separable data set $\{(x_i, y_i)\}_{i=1}^N$, where $y_i \in \{0, 1\}$. The $\mathcal{L}^2$ max-margin solution set $\theta^*/\|\theta^*\|$ as the set of all unit vectors that maximizes the margin between positive data $(y_i = 1)$ and negative data $(y_i = 0)$, i.e.,

$$\frac{\theta^*}{\|\theta^*\|} := \left\{ \frac{\theta}{\|\theta\|} \,\middle|\, \theta \in \arg\max_{\phi \in \mathbb{R}^d} \min_{1 \le i \le N} \left\{ \frac{\mathrm{sgn}(y_i - 0.5)(x_i^\top \phi)}{\|\phi\|} \right\} \right\}, \tag{2}$$

where $\|\cdot\|$ denotes $\ell_2$ norm. Denote the cross-entropy loss $g(\theta) = \frac{1}{N} \sum_{i=1}^N g(\theta, x_i)$, where $g(\theta, x_i) = -y_i \ln(\hat{y}_i) - (1 - y_i) \ln(1 - \hat{y}_i)$ and $\hat{y}_i = \frac{1}{1 + e^{-\theta^\top x_i}}$. Our main goal is to show that running stochastic AdaGrad-Norm 1 on the cross-entropy loss $g(\theta)$ obtains $\frac{\theta_n}{\|\theta_n\|} \to \frac{\theta^*}{\|\theta^*\|}$ $a.s.$,

For a detailed description of the problem formulation and its background, please refer to Section 2.

**Challenges in Analyzing stochastic AdaGrad-Norm**   Compared to SGD, mSGD, and deterministic AdaGrad-Diagonal, the analysis of stochastic AdaGrad-Norm presents distinct challenges arising from the following four aspects.

(I) Given the $\sigma$-algebra $\mathcal{F}_n$, the adaptive step size $\alpha_0 / \sqrt{\sum_{i=1}^n \|\nabla g(\theta_i, \xi_i)\|^2}$ in *Equation* 1 is a random variable, and is conditionally dependent of $\nabla g(\theta_n, \xi_n)$. Handling the terms $\frac{\alpha_0}{\sqrt{S_n}} \nabla f(\theta_n)^\top \nabla g(\theta_n, \xi_n)$ and $\frac{\alpha_0^2}{S_n} \|\nabla g(\theta_n, \xi_n)\|^2$ becomes complicated due to this conditional dependency, where $f(\theta) := 1 - ((\theta^\top \theta^*)/((\|\theta\| + 1)\|\theta^*\|))$ and $\theta^*$ is a max margin vector. In fact, the conditional expectation terms cannot be straightforwardly calculated by $\frac{\alpha_0}{\sqrt{S_n}} \nabla f(\theta_n)^\top \nabla g(\theta_n)$ and $\frac{\alpha_0^2}{S_n} \mathbb{E}_{\xi_n} \left( \|\nabla g(\theta_n, \xi_n)\|^2 \right)$. This challenge has been effectively resolved in (Jin et al., 2022; Faw et al., 2022; Wang et al., 2023). Faw et al. (2022) addressed this issue by scaling down $1/\sqrt{S_n}$ to $1/\sqrt{S_{n-1} + \|\nabla g(\theta_n)\|^2}$. In Jin et al. (2022); Wang et al. (2023), authors transformed $1/\sqrt{S_n}$ into $1/\sqrt{S_{n-1}} + 1/\sqrt{S_{n-1}} - 1/\sqrt{S_n}$ to obtain a new recurrence relation, where the conditional dependence issue no longer exists. The technique employed in Jin et al. (2022) to solve this issue is also utilized in the proof of this paper.

(II) Even when demonstrating the last-iterate convergence of the objective function $g(\theta_n) \to 0$, it only implies $\theta_n \to \infty$, leaving the limit of the $\mathcal{L}^2$ max-margin direction, i.e., $\theta_n/\|\theta_n\|$, unknown. Since the $\mathcal{L}^2$ max-margin direction is important in some machine learning problems, such as classification, we must conduct additional effort to establish convergence of the $\mathcal{L}^2$ max-margin direction. Moreover, the relevant techniques used to prove the last-iterate convergence for stochastic AdaGrad-Norm cannot be directly applied to establish the corresponding results for implicit bias. We will explain why the techniques cannot be transferred in Section 4 after Theorem 4.1.

(III) Previous results on the implicit bias of SGD and mSGD are based on the assumption that the sampling noise is chosen properly (see Section 3 for more details). Specifically, they assume the strong growth property holds for the sampling noise, i.e., $\mathbb{E}_{\xi_n} \|\nabla g(\theta, \xi_n)\|^2 \le M \|\nabla g(\theta_n)\|^2$. In contrast, the stochastic AdaGrad-Norm method is not related to the choice of sampling noise. Thus, the strong growth property is not required in our analysis.

(IV) For the stochastic AdaGrad-Norm, the properties of the generated iterate points $\theta_n$ are sensitive to the distance between $\theta_n$ and the stationary point. Such a challenge does not exist in previous settings. For example, considering deterministic or stochastic algorithms under a quadratic growth condition, this challenge is successfully bypassed by considering the dynamic system in different segments. However, for the stochastic AdaGrad-Norm, the segment of iterates near and far from the stationary point is highly random, making the previous technique unavailable. Therefore, it becomes challenging in this setting,

**Related Works**   There are only a few work that is related to this topic. For example, Soudry et al. (2018) prove that GD converges to the $\mathcal{L}^2$ max-margin solution for linear classification tasks with exponential-tailed loss. Their result is improved by Nacson et al. (2019) latterly. For SGD and

momentum-based SGD, Wang et al. (2021a) prove the convergence to the $\mathcal{L}^2$ max-margin solution for linear classification task with exponential-tailed loss and regular sampling noise.

For deterministic AdaGrad-Diagonal, (Soudry et al., 2018; Gunasekar et al., 2018; Qian & Qian, 2019) claim that it does not converge to the $\mathcal{L}^2$ max-margin solution as the non-adaptive methods do (e.g. SGD, GD). Instead, for stochastic AdaGrad-Norm, Jin et al. (2022) presents the last-iterate convergence. Wang et al. (2023) and Faw et al. (2022) obtained the convergence rates of stochastic AdaGrad-Norm. The characterization of the converging point (like implicit bias) of stochastic AdaGrad-Norm remains unknown.

**Contributions** In this paper, we present a conclusive response to the aforementioned question. Specifically, we provide rigorous proof demonstrating the convergence of the stochastic AdaGrad-Norm method to the $\mathcal{L}^2$ max-margin solution almost surely. This result emphasizes that the resultant classification hyperplane closely conforms to the solution obtained through the application of the hard Support Vector Machine (see Theorems 4.2 and 4.3).

In comparison to previous works that mainly focused on regular sampling noise Wang et al. (2021b), our study stands out by its ability to handle a wide range of stochastic settings (Assumption 3.1). Specifically, our study can be applied to any stochastic algorithms with bounded noise, i.e., $\nabla g(\theta, \xi_n) = \nabla g(\theta) + \xi_n$, (for some $\xi_n, \sup_{n \geq 1} \|\xi_n\| < +\infty$), and the stochastic algorithms with regular sampling noise.

Our technical contributions are summarized as follows:

(I) We begin by adopting a divide-and-conquer approach, simultaneously applying a specific indicator function at both ends of the stochastic dynamical system. This novel approach allows us to analyze the generated iterate points' properties properly. When the iterate point is close to the stationary point, we leverage second-order information from the loss function to provide a deeper characterization of the algorithm's behavior. Conversely, when the iterate point is far from the stationary point, we establish a local strong growth property. Combining these two scenarios, and by exploiting the separability property inherent in the dataset, we conclusively demonstrate that the AdaGrad-Norm algorithm converges towards a max-margin solution.

(II) In a parallel line of investigation, we employ the martingale method to establish the almost everywhere convergence result. This pivotal outcome enables us to convert the convergence order of the partition vector into an order related to the iterates' norm, specifically, $\left\| \frac{\theta_n}{\|\theta_n\|} - \frac{\theta^*}{\|\theta^*\|} \right\|^2 = O(\|\theta_n\|^{-\alpha})$ $(\forall \, 0 < \alpha < 1)$ $a.s..$ By combining this result with the earlier amplitude findings, we ultimately derive the convergence rate of the partition vector as $\min_{1 \leq k \leq n} \left\| \frac{\theta_k}{\|\theta_k\|} - \frac{\theta^*}{\|\theta^*\|} \right\| = o\left( \ln^{-\frac{1-\epsilon}{2}} n \right)$ $(\forall \, \epsilon > 0)$ $a.s..$

## 2 PROBLEM FORMULATION

In this section, we give the detailed formulation of our aimed problem. We consider the linear classification problem with linearly separable data set $\{(x_i, y_i)\}_{i=1}^N$, where $y_i \in \{0, 1\}$. Here, separability means that there exists a vector $\theta_0 \in \mathbb{R}^d$, such that for any $y_i = 1$, $\theta_0^\top x_i > 0$, and for any $y_i = 0$, $\theta_0^\top x_i < 0$. Meanwhile, we call $\theta_0$ as a margin vector. The setting has been considered in many existing works (Soudry et al. (2018); Wang et al. (2021a); Qian & Qian (2019)).

Denote $\| \cdot \|$ as the $\ell_2$ norm. Denote the $\mathcal{L}^2$ max-margin solution set $\theta^*/\|\theta^*\|$ as the set of all unit vectors that maximizes the margin between the positive data $(y_i = 1)$ and the negative data $(y_i = 0)$, which can be formulation by Equation 2. Equivalently, it is also common in the literature to denote $\frac{\theta^*}{\|\theta^*\|} := \left\{ \frac{\theta}{\|\theta\|} \,\middle|\, \theta \in \arg\min_{\phi \in \mathbb{R}^d} \left\{ \|\phi\| \,\middle|\, \mathrm{sgn}(y_i - 0.5)(\phi^\top x_i) \geq 1, \, \forall i \right\} \right\}$. The two definitions are equivalent.

We set the cross-entropy loss as our loss function, i.e., $g(\theta) = \frac{1}{N} \sum_{i=1}^N g(\theta, x_i)$, where $g(\theta, x_i) = -y_i \ln(\hat{y}_i) - (1 - y_i) \ln(1 - \hat{y}_i)$ and $\hat{y}_i = \frac{1}{1 + e^{-\theta^\top x_i}}$. This loss function is widely used in logistic regression. This is a special case of the exponential-tail loss, as discussed in Soudry et al. (2018);

Wang et al. (2021a). Since the choice of cross-entropy loss does not affect the validity of our analysis, while the use of exponential-tail loss does increase many tedious notations, we present our results under the logistic regression setting in the rest of this paper for brevity. Our results can easily be generalized to the stochastic AdaGrad-Norm method with tight exponential-tail loss [1]. For function $g$, we have the following property.

**Property 1.** *The gradient of the loss function, denoted as $\nabla g(\theta)$, satisfies Lipschitz continuity, i.e., $\forall\, \theta_1,\, \theta_2 \in \mathbb{R}^d$, there is $\|\nabla g(\theta_1) - \nabla g(\theta_2)\| \leq c\|\theta_1 - \theta_2\|$, where $c$ is the Lipschitz constant of the function $\nabla g(\theta)$.*

Due to the particularity of classification problems, the global optimal point do not exists. When $\theta_n$ tends to infinity along a certain margin vector, the value of the loss function tends to zero. For any $\epsilon > 0$ and any margin vector $e$, there exists a positive constant $N_0$ associated with $e$, such that for any $\theta/\|\theta\| = e$ and $\|\theta\| > N_0$, we have $g(\theta) < \epsilon$, i.e.,

$$\lim_{\|\theta\|\to+\infty, \theta/\|\theta\|=e} g(\theta) = 0,$$

where $e$ is a margin vector of the data set $\{(x_i, y_i)\}_{i=1}^N$. However, we are more interested in the case that $e$ is a $\mathcal{L}^2$ max-margin vector, which has better generalization.

In the following, we will give the convergence results of the stochastic AdaGrad-Norm method, described in (1), with the aforementioned objective function $g(\theta)$.

## 3 Noise Model Assumption

The results we are going to present hold for the natural noise model induced by mini-batch sampling. Nevertheless, to incorporate a broader family of noise model, such as the bounded variance model, we present a general noise model under which we derive our main results.

We first give our assumption on the unbiased estimation $\nabla g(\theta, \xi_n)$ of the gradient. Here, unbiasedness implies that $\mathbb{E}_{\xi_n} \nabla g(\theta, \xi_n) = \nabla g(\theta)$.

**Assumption 3.1.** *There exist $M_0 > 0$, $a > 0$, such that the variance of $\nabla g(\theta, \xi_n)$ satisfies*

$$\mathbb{E}_{\xi_n} \left\|\nabla g(\theta, \xi_n)\right\|^2 \leq M_0 \left\|\nabla g(\theta)\right\|^2 + a.$$

*Meanwhile, there exist $\delta_0 > 0$, $\hat{K} > 0$, such that when $g(\theta) < \delta_0$, there is $\|\nabla g(\theta, \xi_n)\| \leq \hat{K}$.*

Remarkably, the Assumption 3.1 differs from that in the existing works on the implicit bias of stochastic algorithms, in which regular sampling noise is taken into consideration. In contrast, we consider all estimation noise in the assumption, which includes the regular sampling noise (see the following remark).

**Regular Sampling Noise**   The regular sampling noise is given by

$$\nabla g(\theta, \xi_n) = \frac{1}{|C_i|} \sum_{\bar{x} \in C_i} \nabla g(\theta, \bar{x}),$$

where $C_i$ is a randomly selected mini-batch from the given data set. Through Lemma 8 in Wang et al. (2021b), we know that sampling noise satisfies the *strong growth condition*, i.e., $\mathbb{E}_{\xi_n} \|\nabla g(\theta, \xi_n)\|^2 \leq \tilde{M} \|\nabla g(\theta)\|^2$.

Since any subset (mini-batch) of a linearly separable data set is separable, we know that $\theta$ satisfying $g(\theta) < \delta_0$ is a margin vector of $\{x_i, y_i\}$ by Lemma A.10 with $\delta_0 = (\ln 2)/2N$. Then by Lemma A.8, we have

$$\|\nabla g(\theta, \xi_n)\| = \frac{1}{|C_i|} \left\|\sum_{\bar{x} \in C_i} \nabla g(\theta, \bar{x})\right\| \leq \frac{1}{|C_i|} \sum_{\bar{x} \in C_i} \left\|\nabla g(\theta, \bar{x})\right\| \leq \frac{k_2}{|C_i|} \sum_{\bar{x} \in C_i} g(\theta, \bar{x})$$

$$\leq \frac{k_2 N}{|C_i|} g(\theta) < \frac{k_2}{|C_i|} \cdot \frac{\ln 2}{2} =: \hat{K}.$$

Hence the regular sampling noise satisfies Assumption 3.1.

---

[1] We will demonstrate the easiness of this generalization in Appendix B.10.

# 4 MAIN RESULTS

Now, we are ready to present our main results. Below, we present the last-iterate convergence result of stochastic AdaGrad-Norm, which was first proven by Jin et al. (2022).

**Theorem 4.1.** *(Theorem 3 in Jin et al. (2022)) Suppose that Assumption 3.1 holds. Consider the classification problem with the cross-entropy loss on a linearly separable data set (Section 2). For the stochastic AdaGrad-Norm method given in Equation 1 equipped with step size $\alpha_0 > 0$ and initial parameter $\theta_1 \in \mathbb{R}^d$, we have $g(\theta_n) \to 0$ a.s., and $\|\theta_n\| \to +\infty$ a.s..*

This proof to this theorem can be found in Jin et al. (2022), but in order to make the paper self-contained, we provide the proof of this theorem in Appendix B.8. Below, we point out that the method in Jin et al. (2022) cannot be directly applied to the analysis for the implicit bias. The authors Jin et al. (2022) construct a recursive iterative inequity therein for $g(\theta)$, i.e.,

$$g(\theta_{n+1}) - g(\theta_n) \leq \frac{k}{S_{n-1}} + c_n \tag{3}$$

with $\sum_{n=1}^{+\infty} c_n < +\infty$ and $k > 0$. Then, their goal is to prove that the difference between $\|\nabla g(\theta_{n+1})\|^2$ and $\|\nabla g(\theta_n)\|^2$ becoming sufficiently small as the iterations progress. To do so, they try to bound $\|\nabla g(\theta_{n+1})\|^2 - \|\nabla g(\theta_n)\|^2$ via $g(\theta_{n+1}) - g(\theta_n)$ and inequity $\|\nabla g(\theta)\|^2 \leq 2cg(\theta)$ for Lipschitz constant $c$ of $\nabla g$. However, to obtain the implicit bias, the techniques in Jin et al. (2022) become unsuitable due to the nuanced nature of our constructed Lyapunov function, i.e., $\|\theta_n/\|\theta_n\| - \theta^*/\|\theta^*\|\|^2$. Specifically, the terms $\nabla(\|\theta_n/\|\theta_n\| - \theta^*/\|\theta^*\|\|^2)^\top \nabla g(\theta_n, \xi_n)/\sqrt{S_n}$ and $\|\theta_n/\|\theta_n\| - \theta^*/\|\theta^*\|\|^2$ lack a clear and evident quantitative relationship, making it difficult for us to obtain *Equation* 3. Consequently, novel methods and techniques become imperative to address this challenge.

Next, we present the almost surely convergence analysis of the $\mathcal{L}^2$ max-margin direction $\theta_n/\|\theta_n\|$.

**Theorem 4.2.** *Suppose that Assumption 3.1 holds. Consider the classification problem with the cross-entropy loss on a linearly separable data set (Section 2). For the stochastic AdaGrad-Norm method given in Equation 1 equipped with step size $\alpha_0 > 0$ and initial parameter $\theta_1 \in \mathbb{R}^d$, we have*

$$\frac{\theta_n}{\|\theta_n\|} \to \frac{\theta^*}{\|\theta^*\|} \quad a.s.,$$

*where $\theta^*/\|\theta^*\|$ is the $\mathcal{L}^2$ max-margin solution.*

In Theorem 4.2, we prove that the stochastic AdaGrad-Norm method has the implicit bias to find the $\mathcal{L}^2$ max-margin solution.

Since the full proof is long, we move it to Appendix B.9. A proof sketch now follows, offering an overview of the core arguments constituting the proof.

*Proof Sketch.* Given

$$f(\theta) := 1 - \frac{\theta^\top \hat{\theta}^*}{\|\theta\| + 1}$$

with $\hat{\theta}^* := \theta^*/\|\theta^*\|$, which tends to $\left\|\frac{\theta}{\|\theta\|} - \frac{\theta^*}{\|\theta^*\|}\right\|^2$ as $\theta \to +\infty$. We then prove $f(\theta_n) \to 0$ a.s..

**Step 1**: In this step, we construct a recursive inequality for $f(\theta_n)$. We derive that

$$\mathbb{E}\left(f(\theta_{n+1})\right) - \mathbb{E}\left(f(\theta_n)\right) \leq -\mathbb{E}\left(\left(\frac{\hat{\theta}^*\|\theta_n\| - \frac{\theta_n \theta^\top \hat{\theta}^*}{\|\theta_n\|}}{(\|\theta_n\| + 1)^2}\right)^\top \frac{\alpha_0 \nabla g(\theta_n)}{\sqrt{S_{n-1}}}\right) + \mathbb{E}\left(G_n\right), \tag{4}$$

where

$$G_n := \left|\left(\frac{\hat{\theta}^*\|\theta_n\| - \frac{\theta_n \theta^\top \hat{\theta}^*}{\|\theta_n\|}}{(\|\theta_n\| + 1)^2}\right)^\top \alpha_0 \nabla g(\theta_n, \xi_n)\left(\frac{1}{\sqrt{S_{n-1}}} - \frac{1}{\sqrt{S_n}}\right)\right| + \frac{T_n \alpha_0^2 \|\nabla g(\theta_n, \xi_n)\|^2}{S_n}$$

$$+ \frac{\alpha_0 \hat{\theta}^{*\top} \nabla g(\theta_n, \xi_n)}{(\|\theta_n\| + 1)^2 \sqrt{S_{n-1}}} + \frac{N^2 \max_{1 \leq i \leq N}\{\|x_i\|^2\}}{2k_1^2 \ln^2 2} \cdot \frac{\|\nabla g(\theta_n)\|^2}{\sqrt{S_{n-1}}},$$

where $T_n$ is defined in *Equation* 66. It can be shown that $\sum_{n=1}^{+\infty} \mathbb{E}(G_n) < +\infty$ (see the specific proof in Appendix B.9). Thus, we focus on studying the first term on the right-hand side of *Equation* 4.

**Step 2** In this step, we focus on decomposing the first term in *Equation* 4.

$$\mathbb{E}\left(\left(\frac{\hat{\theta}^*\|\theta_n\| - \frac{\theta_n \theta^\top \hat{\theta}^*}{\|\theta_n\|}}{(\|\theta_n\|+1)^2}\right)^\top \frac{\nabla g(\theta_n)}{\sqrt{S_{n-1}}}\right) \le \mathbb{E}\left(\frac{1}{N\sqrt{S_{n-1}}} \sum_{i=1}^N \psi_i \frac{\theta_n{}^\top x_i - \hat{\theta}^{*\top} x_i \|\theta_n\|}{(\|\theta_n\|+1)^2}\right)$$
$$:= \mathbb{E}(H_n),$$

where the definition of $f_{x_i}(\theta, x_i)$ can refer *Equation* 41 in Appendix B.9 and $\psi_i := \text{sgn}(y_i - 0.5)$. We then prove that the right-hand side of the above inequality is negative. Denote the index of the support vector as $\mathbf{i}_n := \{i | i = \arg\min_{1 \le i \le N} \psi_i \theta_n{}^\top x_i / \|\theta_n\|\}$, and $i_n$ is a element of $\mathbf{i}_n$. Then we have $\exists\, \hat{k}_0 > 0$, such that

$$H_n \le \frac{f_{x_i}(\theta_n, x_{i_n})\|\theta_n\|}{N(\|\theta_n+1\|)^2\sqrt{S_{n-1}}}\left(\sum_{i\in\mathbf{i}_n} \psi_i\left(\frac{\theta_n{}^\top x_i}{\|\theta_n\|} - \hat{\theta}^{*\top} x_i\right)\right.$$
$$\left. + \hat{k}_0 \sum_{i\notin\mathbf{i}_n}^N \frac{\psi_i}{e^{(d_{n,i}-d_{n,i_n})(\|\theta_n\|+1)}}\left(\frac{\theta_n{}^\top x_i}{\|\theta_n\|} - \hat{\theta}^{*\top} x_i\right)\right),$$

(5)

where $d_{n,i} := |\theta_n^\top x_i|/\|\theta_n\|$. The first term of the above inequality is negative.

**Step 3**: In this step, we give a bound of the second term of *Equation* 5. We employ the divide and conquer method to handle the second term of *Equation* 5. We also classify the discussion based on the distance between the partition vector $\theta_n/\|\theta_n\|$ and the max-margin vector $\hat{\theta}^*$. As a result, we construct two events $\mathcal{C}_n^+ := \{\|(\theta_n/\|\theta_n\|) - \hat{\theta}^*\| \ge \mathcal{L}\}$, $\mathcal{C}_n^- := \{\|(\theta_n/\|\theta_n\|) - \hat{\theta}^*\| < \mathcal{L}\}$. In the case where $\mathcal{C}_n^-$ occurs, that is, when $\theta/\|\theta\|$ is to $\hat{\theta}^*$, we have the following geometric relationship lemma:

**Lemma 4.1.** *Let $\{x_i\}_{i=1}^N$ be $d$-dimensional vectors. Then there is a vector $x_\theta$ such that $|\theta^\top x_\theta|/\|\theta\| := \min_{1\le i\le N}\{|\theta^\top x_i|/\|\theta\|\}$. Let $\theta^*/\|\theta^*\|$ as the max-margin vector. Then there exists $\delta_0 > 0$, $\hat{r} > 0$, such that for all $\theta/\|\theta\| \in U(\theta^*/\|\theta^*\|, \delta_0)/\{\theta^*/\|\theta^*\|\}$, where $U(\theta^*/\|\theta^*\|, \delta_0)$ means $\delta_0$-neighborhood of vector $\theta^*/\|\theta^*\|$, it holds $\left|\frac{\theta^\top x_i}{\|\theta\|} - \frac{\theta^* x_i}{\|\theta^*\|}\right| < \hat{r}\left|\frac{\theta^\top x_\theta}{\|\theta\|} - \frac{\theta^* x_\theta}{\|\theta^*\|}\right| (\forall\, i \in [1, N])$.*

Through this lemma we can obtain following inequity:

$$\sum_{i\notin\mathbf{i}_n} \mathbf{1}_{\mathcal{C}_n^-} \frac{\psi_i}{e^{(d_{n,i}-d_{n,i_n})(\|\theta_n\|+1)}}\left(\frac{\theta_n{}^\top x_i}{\|\theta_n\|} - \hat{\theta}^{*\top} x_i\right) \le \hat{k}_0 \hat{c} N \frac{\hat{U}}{\|\theta_n\|+1}$$
$$+ \hat{k}_0 \frac{N\hat{r}}{e^{\hat{U}}}\left|\frac{\theta_n{}^\top x_{i_n}}{\|\theta_n\|} - \hat{\theta}^{*\top} x_{i_n}\right|,$$

where $\hat{U}$ is an undetermined constant. Similarly, where $\mathcal{C}_n^+$ occurs, we get

$$\sum_{i\notin\mathbf{i}_n} \mathbf{1}_{\mathcal{C}_n^+} \frac{\psi_i}{e^{(d_{n,i}-d_{n,i_n})(\|\theta_n\|+1)}}\left(\frac{\theta_n{}^\top x_i}{\|\theta_n\|} - \hat{\theta}^{*\top} x_i\right) \le \frac{N\cdot\tilde{M}_1}{e^{s'\|\theta_n\|}} + \hat{k}_1 \frac{N}{e^{\hat{U}}}\left|\frac{\theta_n{}^\top x_{i_n}}{\|\theta_n\|} - \hat{\theta}^{*\top} x_{i_n}\right|,$$

where $M_1$ is a constant. Combining, we get

$$\sum_{i\notin\mathbf{i}_n} \frac{\psi_i}{e^{(d_{n,i}-d_{n,i_n})(\|\theta_n\|+1)}}\left(\frac{\theta_n{}^\top x_i}{\|\theta_n\|} - \hat{\theta}^{*\top} x_i\right) \le (\hat{k}_0\hat{r} + \hat{k}_1)\frac{N}{e^{\hat{U}}}\left|\frac{\theta_n{}^\top x_{i_n}}{\|\theta_n\|} - \hat{\theta}^{*\top} x_{i_n}\right| + \frac{N\cdot\tilde{M}_1}{e^{s'\|\theta_n\|}}$$
$$+ \hat{k}_0\hat{c}N\frac{\hat{U}}{\|\theta_n\|+1}.$$

By adjusting the value of $\hat{U}$, we can always cancel out the first term with the half of the negative term in *Equation* 5, and then we only need to prove that the remainder term can be neglected. That

is to prove

$$\sum_{n=1}^{+\infty} \mathbb{E}\left( \frac{f_{x_i}(\theta_n, x_{i_n})\|\theta_n\|}{N(\|\theta_n + 1\|)^2 \sqrt{S_{n-1}}} \cdot \left( \frac{N \cdot \tilde{M}_1}{e^{s'\|\theta_n\|}} + \hat{k}_0 \hat{c} N \frac{\hat{U}}{\|\theta_n\| + 1} \right) \right) < +\infty.$$

**Step 4** In this step, we will prove the convergence of the series sum in the final step of the third step. We prove this conclusion by the following lemma:

**Lemma 4.2.** *Consider the AdaGrad Equation 1 under our problem setting in Section 2 and Assumption 3.1. We have for any $\alpha_0 > 0$, $\alpha > 0$, $\theta_1$, there is $\sum_{k=2}^{n} \mathbb{E}\left( \frac{\|\nabla g(\theta_k)\|^2}{\sqrt{S_{k-1}} g(\theta_k) \ln^{1+\alpha}(g(\theta_k))} \right) < +\infty.$*

**Step 5** Through the above steps, we have obtained the following recursive formula:

$$\mathbb{E}(f(\theta_{n+1}|\mathcal{F}_n) - f(\theta_n) \leq -\frac{1}{2} \frac{f_{x_i}(\theta_n, x_{i_n})\|\theta_n\|}{N(\|\theta_n + 1\|)^2 \sqrt{S_{n-1}}} \sum_{i \in \mathbf{i}_n} \psi_i \left( \frac{\theta_n^\top x_i}{\|\theta_n\|} - \hat{\theta}^{*\top} x_i \right) + c_n,$$

where $\sum_{n=1}^{+\infty} c_n < +\infty$. According to the martingale difference sum convergence theorem, we can conclude that $f(\theta_n)$ convergence almost surely. Then, we prove by contradiction that this limit can only be 0. We assume that this assumption is not 0, and immediately derive a contradiction from the following result:

$$\sum_{n=2}^{+\infty} \frac{\|\theta_n\| f_{x_{i_n}}(\theta_n, x_{i_n})}{N(\|\theta_n\| + 1)^2 \sqrt{S_{n-1}}} > q_1 \sum_{n=1}^{+\infty} \left( \ln \|\theta_{n+1}\| - \ln \|\theta_n\| \right) - q_2 \sum_{n=1}^{+\infty} \frac{\|\nabla g(\theta_n, \xi_n)\|^2}{\|\theta_n\|^2 S_n} = +\infty \ a.s..$$

Therefore, we have proved this theorem. $\qquad\square$

The previous works (Soudry et al., 2018; Gunasekar et al., 2018; Qian & Qian, 2019) point out that the $\mathcal{L}^2$ max-margin direction of the AdaGrad method depends on the initial point and step size. Hence, it is not as predictable and robust as the non-adaptive methods (e.g., SGD, GD). However, the claim only holds true for the deterministic AdaGrad-diagonal method, which is described by the system $\theta_{n+1} = \theta_n - \eta \mathbf{G}_n^{-1/2} \nabla g(\theta_n)$, where $\mathbf{G}_n \in \mathbb{R}^{d \times d}$ is a diagonal matrix such that, $\forall i: \mathbf{G}_n[i, i] = \sum_{k=0}^{n} (\nabla g(\theta_k)[i])^2$. Nonetheless, it is crucial to emphasize the substantial distinctions inherent in the properties of the algorithm under discussion when compared to the stochastic AdaGrad-Norm method. Specifically, the stochastic AdaGrad-Norm method maintains a uniform step size consistently across all components, leading to fundamental differences in the analytical methods and techniques that are used to prove the convergence of these two algorithms. For the AdaGrad-diagonal algorithm, we are able to compute the key component, denoted as $-\nabla f(\theta_n)^\top (\theta_{n+1} - \theta_n)$, which determines the update direction of the decision boundary, analogous to *Equation* 42. This computation yields the following expression:

$$\mathbb{E}(\nabla f(\theta_n)^\top G_n^{-\frac{1}{2}} \nabla g(\theta_n))$$
$$= \mathbb{E}\left( \frac{1}{N\sqrt{S_{n-1}}} \sum_{i=1}^{N} \text{sgn}(y_i - 0.5) f_{x_i}(\theta_n, x_i) \left( \frac{\theta_n^\top G_n^{-\frac{1}{2}} x_i - \hat{\theta}^{*\top} G_n^{-\frac{1}{2}} x_i \|\theta_n\|}{(\|\theta_n\| + 1)^2} \right. \right.$$
$$\left. \left. - \frac{\theta_n^\top G_n^{-\frac{1}{2}} x_i}{2(\|\theta_n\| + 1)^2} \left\| \frac{\theta_n}{\|\theta_n\|} - \hat{\theta}^* \right\|^2 \right) \right).$$

Here, we have omitted higher-order terms from consideration. It is worth noting that, given the diagonal matrix structure of $G_n$ with distinct diagonal elements, as the iterations progress, our pursuit effectively converges towards identifying the max-margin vector associated with the dataset $\{G_\infty^{-\frac{1}{2}} \cdot x_i, y_i\}_{i=1}^{N}$. This differs from the previous result.

Finally, we present the convergence rate analysis of the stochastic AdaGrad-Norm method, as shown in Theorem A.4.

**Theorem 4.3.** *Suppose that Assumption 3.1 holds. Consider the classification problem with the cross-entropy loss on a linearly separable data set (Section 2). For the stochastic AdaGrad-Norm method given in Equation 1 equipped with step size $\alpha_0 > 0$ and initial parameter $\theta_1 \in \mathbb{R}^d$, we have $\min_{1 \leq k \leq n} \left\| \theta_k / \|\theta_k\| - \theta^* / \|\theta^*\| \right\| = o\left(1 / \ln^{\frac{1-\epsilon}{2}} n\right) \ (\forall \, 0 < \epsilon < 1) \ a.s..$ where $\theta^* / \|\theta^*\|$ is the $\mathcal{L}^2$ max-margin solution.*

This theorem presents the convergence rate $o\left(1 \Big/ \ln^{\frac{1-\epsilon}{2}} n\right) \ \forall \, \epsilon > 0 \ a.s.$ of the $\mathcal{L}^2$ max-margin direction. This achievement is also new to the literature.

Comparative analysis against corresponding GD results, given by Soudry et al. (2018), reveals that the convergence rate for both $g(\theta_n)$ and $\theta_n/\|\theta_n\|$ within stochastic AdaGrad-Norm is comparatively slower. This observation isn't unexpected, as the stochastic AdaGrad-Norm method uses a decreasing step size, which will be much smaller than that used in GD as iteration grows. However, for GD, one has to verify whether the step size $\alpha$ satisfies $\alpha < 2\beta^{-1}\sigma_{\max}^{-2}(X)$ (Soudry et al. (2018)), where $X$ is the data matrix, $\sigma_{\max}(\cdot)$ denotes the maximal singular value and $\beta$ is a constant characterized by loss function $g$. This checking rule requires an extra burden of hyperparameter tuning. In contrast, the stochastic AdaGrad-Norm method uses simple step sizes.

The proof strategy of this theorem is very similar to that of Theorem 4.2. We only need to replace the function $f(\theta)$ in the proof of Theorem 4.2 with $\|\theta\|^\alpha \cdot f(\theta)$ $(\forall \, 0 < \alpha < 1)$.

*Proof.* For any $0 < \alpha < 1$, we construct a function $r(\theta) := \|\theta\|^\alpha \cdot f(\theta)$ $(0 < \alpha < 1)$, where $f$ is defined in *Equation* 65. Then we calculate $\nabla r(\theta)$, acquiring $\nabla r(\theta) = \nabla(\|\theta\|^\alpha)^\top f(\theta) + (\nabla f(\theta))^\top \|\theta\|^\alpha = \frac{\alpha \frac{\theta}{\|\theta\|} \cdot f(\theta)}{\|\theta\|^{1-\alpha}} + \|\theta\|^\alpha \nabla f(\theta)$, and $\|\nabla^2 r(\theta)\| = O((\|\theta\| + 1)^{\alpha-2})$. Meanwhile, we assign the Lipschitz constant of $\nabla^2 r(\theta)$ as $c_1$. Then we get

$$r(\theta_{n+1}) - r(\theta_n) \leq \nabla r(\theta_n)^\top (\theta_{n+1} - \theta_n) + \|\nabla^2 r(\theta_n)\| \cdot \|\theta_{n+1} - \theta_n\|^2 + c_1 \|\theta_{n+1} - \theta_n\|^3$$

$$\leq -\alpha \frac{\alpha_0 (\frac{\theta_n}{\|\theta_n\|})^\top \nabla g(\theta_n, \xi_n) f(\theta_n)}{\sqrt{S_n} \|\theta_n\|^{1-\alpha}} - \|\theta_n\|^\alpha \frac{\alpha_0 \nabla f(\theta_n)^\top \nabla g(\theta_n, \xi_n)}{\sqrt{S_n}} + q_0 \frac{\alpha_0^2 \|\nabla g(\theta_n, \xi_n)\|^2}{(\|\theta_n\| + 1)^{2-\alpha} S_n}$$

$$+ c_1 \alpha_0^3 \frac{\|\nabla g(\theta_n, \xi_n)\|^2}{S_n^3}.$$

$$(6)$$

Notice that $\nabla f(\theta_n) = \frac{\theta - \hat{\theta}^* \|\theta_n\|}{(\|\theta\| + 1)^2} - \frac{\theta}{2(\|\theta\| + 1)^2} \left\| \frac{\theta_n}{\|\theta_n\|} - \hat{\theta}^* \right\|^2 - \frac{\hat{\theta}^*}{(\|\theta_n\| + 1)^2}.$ For the first term and second term in the right-hand of *Equation* 6, we know that

$$-\mathbb{E}\left( \alpha \frac{\alpha_0 (\frac{\theta_n}{\|\theta\|})^\top \nabla g(\theta_n, \xi_n) f(\theta_n)}{\sqrt{S_n} \|\theta_n\|^{1-\alpha}} + \|\theta_n\|^\alpha \frac{\alpha_0 \nabla f(\theta_n)^\top \nabla g(\theta_n, \xi_n)}{\sqrt{S_n}} \Big| \mathcal{F}_n \right)$$

$$\leq -\alpha \frac{\alpha_0 (\frac{\theta_n}{\|\theta_n\|})^\top \nabla g(\theta_n) f(\theta_n)}{\sqrt{S_n} \|\theta_n\|^{1-\alpha}} + \|\theta_n\|^\alpha H_n + \|\theta_n\|^\alpha \frac{\alpha_0}{\sqrt{S_n}} \frac{\|\theta_n\|^2}{2(\|\theta_n\| + 1)^2} \cdot \frac{\theta_n^\top \nabla g(\theta_n)}{\|\theta_n\|^2} \left\| \frac{\theta_n}{\|\theta_n\|} - \hat{\theta}^* \right\|^2,$$

where $H_n$ is defined in *Equation* 43. Through Theorem 4.2, we know the vector $\theta_n/\|\theta_n\|$ tend to the max-margin vector almost surely, which means $\frac{\theta_n^\top \nabla g(\theta_n)}{\|\theta_n\|^2} < 0$ when $n$ is sufficient large. Then,

$$-\mathbb{E}\left( \alpha \frac{\alpha_0 (\frac{\theta_n}{\|\theta\|})^\top \nabla g(\theta_n, \xi_n) f(\theta_n)}{\sqrt{S_n} \|\theta_n\|^{1-\alpha}} + \|\theta_n\|^\alpha \frac{\alpha_0 \nabla f(\theta_n)^\top \nabla g(\theta_n, \xi_n)}{\sqrt{S_n}} \Big| \mathcal{F}_n \right)$$

$$\leq (1 - \alpha) \frac{\alpha_0 (\frac{\theta_n}{\|\theta_n\|})^\top \nabla g(\theta_n) f(\theta_n)}{\sqrt{S_n} \|\theta_n\|^{1-\alpha}} + \|\theta_n\|^\alpha H_n$$

$$+ \|\theta_n\|^\alpha \frac{\alpha_0}{\sqrt{S_n}} \left| 1 - \frac{\|\theta_n\|^2}{(\|\theta_n\| + 1)^2} \right| \cdot \frac{|\theta_n^\top \nabla g(\theta_n)|}{\|\theta_n\|^2} |f(\theta_n)|$$

$$+ \|\theta_n\|^\alpha \frac{\alpha_0}{\sqrt{S_n}} \frac{\|\theta_n\|^2}{(\|\theta_n\| + 1)^2} \cdot \frac{|\theta_n^\top \nabla g(\theta_n)|}{\|\theta_n\|^2} \left| f(\theta_n) - \frac{1}{2} \left\| \frac{\theta_n}{\|\theta_n\|} - \hat{\theta}^* \right\|^2 \right|$$

$$\leq \|\theta_n\|^\alpha H_n + O\left( \frac{\|\nabla g(\theta_n)\|^2}{\sqrt{S_{n-1}} g(\theta_n) \ln^{2-\alpha}(g(\theta_n))} \right).$$

Through *Equation* 52, we have

$$\|\theta_n\|^\alpha H_n = O\left(\frac{\|\nabla g(\theta_n)\|^2}{\sqrt{S_{n-1}}g(\theta_n)\ln^{2-\alpha}(g(\theta_n))}\right).$$

Then we use Lemma 4.2 and obtain

$$\sum_{n=1}^{+\infty} -\mathbb{E}\left(\alpha\frac{\alpha_0(\frac{\theta_n}{\|\theta\|})^\top \nabla g(\theta_n, \xi_n)f(\theta_n)}{\sqrt{S_n}\|\theta_n\|^{1-\alpha}} + \|\theta_n\|^\alpha\frac{\alpha_0\nabla f(\theta_n)^\top \nabla g(\theta_n, \xi_n)}{\sqrt{S_n}}\Big|\mathcal{F}_n\right)$$
$$< O\left(\sum_{n=1}^{+\infty}\frac{\|\nabla g(\theta_n)\|^2}{\sqrt{S_{n-1}}g(\theta_n)\ln^{2-\alpha}(g(\theta_n))}\right) < +\infty \ a.s.. \tag{7}$$

For the third term in the right-hand of *Equation* 6, we have $\exists \, Q_1 > 0$, such that

$$\sum_{n=1}^{+\infty}\frac{\alpha_0^2\|\nabla g(\theta_n, \xi_n)\|^2}{(\|\theta_n\|+1)^{2-\alpha}S_n} \le Q_1\sum_{n=1}^{+\infty}\frac{\|\nabla g(\theta_n, \xi_n)\|^2}{\ln^{2-\alpha}(g(\theta_n))S_n} \le Q_1\sum_{n=1}^{+\infty}\frac{\|\nabla g(\theta_n, \xi_n)\|^2}{\ln^{2-\alpha}(S_n)S_n}$$
$$+ Q_1\sum_{n=1}^{+\infty}\frac{\|\nabla g(\theta_n, \xi_n)\|^2 g(\theta_n)}{\ln^{2-\alpha}(g(\theta_n))\sqrt{S_n}}. \tag{8}$$

For the fourth term in the right-hand of *Equation* 6, we know

$$\sum_{n=1}^{+\infty}c_1\alpha_0^3\frac{\|\nabla g(\theta_n, \xi_n)\|^2}{S_n^3} < +\infty \ a.s. \tag{9}$$

Substitute *Equation* 7, *Equation* 8 and *Equation* 9 into *Equation* 6, we get

$$\sum_{n=1}^{+\infty}\left(\mathbb{E}\left(r(\theta_{n+1})\big|\mathcal{F}_n\right) - r(\theta_n)\right) < +\infty \ a.s..$$

By *The Martingale Convergence Theorem*, we get $\lim_{n\to+\infty} r(\theta_n) < +\infty \ a.s.$ That is, for any $0 < \alpha < 1$, we have

$$f(\theta_n) = O(\|\theta_n\|^{-\alpha}) \ a.s..$$

By the arbitrariness of $\alpha$, we know the $O$ can be written as $o$, so

$$\min_{1\le k\le n}\left\|\frac{\theta_k}{\|\theta_k\|} - \frac{\theta^*}{\|\theta^*\|}\right\| = o\left(\min_{1\le k\le n}\|\theta_k\|^{\frac{-\alpha}{2}}\right) = o\left(\ln^{-\frac{\alpha}{2}}\min_{1\le k\le n}g(\theta_k)\right) \ (\forall \, 0 < \alpha < 1) \ a.s..$$

Through Lemma A.4 and Lemma A.8, we know

$$\min_{1\le k\le n}g(\theta_k) \le \sqrt{\frac{1}{k_1^2}\min_{1\le k\le n}\{\|\nabla g(\theta_k)\|^2\}} \le \sqrt{\frac{\sqrt{\hat{K}n}}{nk_1}\sum_{k=2}^{+\infty}\frac{\|\nabla g(\theta_k)\|^2}{\sqrt{S_{k-1}}}} = O(n^{-\frac{1}{4}}) \ a.s..$$

As a result, we know

$$\min_{1\le k\le n}\left\|\frac{\theta_k}{\|\theta_k\|} - \frac{\theta^*}{\|\theta^*\|}\right\| = o\left(\ln^{-\frac{1-\epsilon}{2}}n\right) \ (\forall \, \epsilon > 0) \ a.s..$$

This completes the proof. $\qquad\qquad\qquad\qquad\qquad\qquad\qquad\qquad\qquad\qquad\qquad\qquad\qquad$ $\square$

## 5 CONCLUSION

This paper focuses on the convergence analysis of the stochastic AdaGrad-Norm method, a widely used variant of the AdaGrad method, with linearly separable data sets. While previous perspectives often suggest that AdaGrad's convergence might hinge on initialization and step size, our findings present a contrasting view. Specifically, we establish that stochastic AdaGrad-Norm exhibits an implicit bias, consistently converging towards the $\mathcal{L}^2$ max-margin solution, even without regularization terms. Furthermore, we present the convergence rates for the $\mathcal{L}^2$ max-margin solution, offering comprehensive insights into the algorithm's convergence dynamics.

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

# A USEFUL LEMMAS

**Lemma A.1.** *(Lemma 6 in Jin et al. (2022)) Suppose that $\{X_n\} \in \mathbb{R}^d$ is a non-negative sequence of random variables. If it holds that $\sum_{n=0}^{\infty} \mathbb{E}(X_n) < +\infty$, then $\sum_{n=0}^{\infty} X_n < +\infty$ holds almost surely.*

**Lemma A.2.** *(Wang et al., 2019) Suppose that $\{X_n\} \in \mathbb{R}^d$ is an $\mathcal{L}_2$ martingale difference sequence, and $(X_n, \mathcal{F}_n)$ is an adaptive process. Suppose it holds that*

$$\sum_{n=1}^{\infty} \mathbb{E}(\|X_n\|^2) < +\infty, \quad or \quad \sum_{n=1}^{\infty} \mathbb{E}(\|X_n\|^2 | \mathcal{F}_{n-1}) < +\infty.$$

*Then, it holds $\sum_{k=0}^{\infty} X_k < +\infty$ almost surely.*

**Lemma A.3.** *(Lemma 13 in Jin et al. (2022)) Consider the AdaGrad Equation 1 under our problem setting given in Section 2 and Assumption 3.1. It holds that for any $\alpha_0 > 0, \theta_1$,*

$$\sum_{k=3}^{n} \mathbb{E}\left(\frac{\|\nabla g(\theta_k)\|^2}{S_{k-1}^{\frac{1}{2}+\epsilon}}\right) < +\infty.$$

**Lemma A.4.** *Consider the AdaGrad Equation 1 under our problem setting in Section 2 and Assumption 3.1. We have for any $\alpha_0 > 0, \theta_1$, there is*

$$\sum_{n=2}^{+\infty} \mathbb{E}\left(\frac{\|\nabla g(\theta_n)\|^2}{\sqrt{S_{n-1}}}\right) < +\infty, \quad and \quad \sum_{n=2}^{+\infty} \frac{\|\nabla g(\theta_n)\|^2}{\sqrt{S_{n-1}}} < +\infty \ \ a.s..$$

**Lemma A.5.** *Consider the AdaGrad Equation 1 under our problem setting in Section 2 and Assumption 3.1, and function $f(\theta) := 1 - \frac{\theta^\top \hat{\theta}^*}{\|\theta\|+1}$, where $\hat{\theta}^*$ ($\|\hat{\theta}^*\| = 1$) is a max margin vector. We have for any $\alpha_0 > 0, \theta_1, \exists \, r_0 > 0, \tilde{M}_0 > 0$, such that*

$$\mathbb{E}\left(\frac{(\nabla f(\theta_n))^\top \nabla g(\theta_n)}{\sqrt{S_{n-1}}}\right)$$

$$\leq \mathbb{E}\left(\frac{f_{x_{i_n}}(\theta_n, x_{i_n})\|\theta_n\|}{N(\|\theta_n\|+1)^2\sqrt{S_{n-1}}}\left(\frac{1}{2}\psi_{i_n}\left(\frac{\theta_n^\top x_{i_n}}{\|\theta_n\|} - \hat{\theta}^{*\top}x_{i_n}\right) + \frac{r_0}{\|\theta_n\|+1} + \frac{\tilde{M}_0}{e^{s'\|\theta_n\|}}\right)\right)$$

$$+ \frac{N\max_{1\leq i\leq N}\{\|x_i\|^2\}}{4c\ln 2} \cdot \mathbb{E}\left(\frac{\|\nabla g(\theta_n)\|^2}{\sqrt{S_{n-1}}}\right),$$

*where $\psi_i := sgn(y_i - 0.5)$ and*

$$f_{x_i}(\theta, x_i) := \begin{cases} f(\theta, x_i), & \text{if } y_i = 0, \\ 1 - f(\theta, x_i), & \text{if } y_i = 1, \end{cases}$$

$$f(\theta, x_i) := \frac{1}{1 + e^{-sgn(y_i - 0.5)\theta^\top x_i}}.$$

**Lemma A.6.** *$x_1$ and $x_2$ are two $d$-dimensional. Then there is a vector $\theta \in \mathbb{R}^d$ which hold $|\theta^\top x_1|/\|\theta\| < |\theta^\top x_2|/\|\theta\|$. We assign $\theta^* := \arg\min_{\{\theta\|\theta^\top x_1|=|\theta^\top x_2|\}} \|\theta\|$. Then there exists $\delta_0 > 0, \hat{r} > 0$, such that*

$$\left|\frac{\theta^\top x_2}{\|\theta\|} - \frac{\theta^* x_2}{\|\theta^*\|}\right| < \hat{r}\left|\frac{\theta^\top x_1}{\|\theta\|} - \frac{\theta^* x_1}{\|\theta^*\|}\right| \tag{10}$$

*for any $\theta$ satisfying $\theta/\|\theta\| \in U(\theta^*, \delta_0)/\{\theta^*\}$.*

**Lemma A.7.** *(Lemma 10 in Jin et al. (2022)) Suppose $f(x) \in C^1$ $(x \in \mathbb{R}^N)$ with $f(x) > -\infty$ and its gradient satisfying the following Lipschitz condition*

$$\|\nabla f(x) - \nabla f(y)\| \leq c\|x - y\|,$$

*then $\forall \, x_0 \in \mathbb{R}^N$, there is*

$$\|\nabla f(x_0)\|^2 \leq 2c(f(x_0) - f^*),$$

*where $f^* = \inf_{x \in \mathbb{R}^N} f(x)$.*

**Lemma A.8.** $\{\hat{x}_i, \hat{y}_i\}$ *is a linear separable data set and $\hat{g}(\theta)$ is the loss of logistic regression. Then we have that if $\theta$ is a margin vertor of $\{\hat{x}_i, \hat{y}_i\}$, the loss function will hold that*

$$k_1 \hat{g}(\theta) \leq \|\nabla \hat{g}(\theta)\| \leq k_2 \hat{g}(\theta),$$

*where $k_1 > 0,\ k_2 > 2$ are two constant.*

**Lemma A.9.** *For a linear separable data set $S$, we assum its max-margin vertor as $\theta^*/\|\theta^*\|$, Then exists a constant $\tilde{\delta}_0 > 0$, making for any $\theta/\|\theta\| \in U(\theta^*/\|\theta^*\|, \tilde{\delta}_0)$, $\theta/\|\theta\|$ is a margin vector.*

**Lemma A.10.** *If a vector $\theta \in \mathbb{R}^d$ is not a margin vector, it will make $g(\theta) > (\ln 2)/N$.*

# B PROOFS OF LEMMAS AND THEOREMS

## B.1 THE PROOF OF LEMMA A.4

*Proof.* Based on calculations, it is easy to observe that when $\|\nabla g(\theta)\| \to 0$, there is $\|\nabla^2 g(\theta)\| = \Theta(\|\nabla g(\theta)\|)$ (Here, the norm represents the maximum eigenvalue of the Hessian matrix.). That means existing $\tilde{d}_0 > 0, \tilde{\delta}_1 > 0$, such that for any $\|\nabla g(\theta)\| < \tilde{\delta}_1$, there is $\|\nabla^2 g(\theta)\| \leq \tilde{d}_0 \|\nabla g(\theta)\|$. Then we assign $\delta_1 := \min\{\ln 2/N, \tilde{\delta}_1/k_2\}$. Lemma A.8 and Lemma A.10, we know when $g(\theta) < \delta_1$, there is $\|\nabla^2 g(\theta)\| \leq \tilde{d}_0 \|\nabla g(\theta)\|$. Then we define $S^{(\delta_2)} := \{\theta | g(\theta) < \delta_2 := \min\{\delta_0, \delta_1\}\}$, where $\delta_0$ defined in Assumption 1. We know that within the set $S^{(\delta_2)}$, the Hessian matrix is Lipschitz continuous. We define $\hat{c}$ as the Lipschitz constant of the Hessian matrix. We consider an event $\mathcal{B}_n := \{\theta_n \in S^{(\delta_2)}\}$. Meanwhile, we assign its complementary event as $\mathcal{B}_n^{(-)}$. Then, through the third-order Taylor expansion, we have

$$\mathbf{1}_{\mathcal{B}_n}\big(g(\theta_{n+1}) - g(\theta_n)\big) \leq -\mathbf{1}_{\mathcal{B}_n} \alpha_0 S_n^{-\frac{1}{2}} \nabla g(\theta_n)^\top \nabla g(\theta_n, \xi_n) + \mathbf{1}_{\mathcal{B}_n} \frac{\alpha_0^2 \|\nabla^2 g(\theta_n)\| \cdot \|\nabla g(\theta_n, \xi_n)\|^2}{S_n}$$

$$+ \mathbf{1}_{\mathcal{B}_n} \frac{\hat{c} \alpha_0^3 \|\nabla g(\theta_n, \xi_n)\|^3}{S_n^{\frac{3}{2}}}.$$

Combining Assumption 3.1, we can get

$$\mathbf{1}_{\mathcal{B}_n}\big(g(\theta_{n+1}) - g(\theta_n)\big) \leq -\mathbf{1}_{\mathcal{B}_n} \frac{\alpha_0 \nabla g(\theta_n)^\top \nabla g(\theta_n, \xi_n)}{\sqrt{S_{n-1}}} + \mathbf{1}_{\mathcal{B}_n} t_0 \left(\frac{1}{\sqrt{S_{n-1}}} - \frac{1}{\sqrt{S_n}}\right)$$

$$+ \mathbf{1}_{\mathcal{B}_n} \frac{\tilde{d}_0 \alpha_0^2 \|\nabla g(\theta_n)\| \cdot \|\nabla g(\theta_n, \xi_n)\|^2}{S_n} + \mathbf{1}_{\mathcal{B}_n} \frac{\hat{c} \hat{K} \alpha_0^3 \|\nabla g(\theta_n, \xi_n)\|^2}{S_n^{\frac{3}{2}}}, \tag{11}$$

where $t_0 = \alpha_0 \hat{\delta}_1 \hat{K}$. For the third term on the right side, we have

$$\mathbf{1}_{\mathcal{B}_n} \frac{\tilde{d}_0 \alpha_0^2 \|\nabla g(\theta_n)\| \cdot \|\nabla g(\theta_n, \xi_n)\|^2}{S_n} = \mathbf{1}\left(\|\nabla g(\theta_n)\| < \frac{2\alpha_0 \tilde{d}_0 \hat{K}_0}{\sqrt{S_n}}\right)\mathbf{1}_{\mathcal{B}_n} \frac{\tilde{d}_0 \alpha_0^2 \|\nabla g(\theta_n)\| \cdot \|\nabla g(\theta_n, \xi_n)\|^2}{S_n}$$

$$+ \mathbf{1}\left(\|\nabla g(\theta_n)\| \geq \frac{2\alpha_0 \tilde{d}_0 \hat{K}_0}{\sqrt{S_n}}\right)\mathbf{1}_{\mathcal{B}_n} \frac{\tilde{d}_0 \alpha_0^2 \|\nabla g(\theta_n)\| \cdot \|\nabla g(\theta_n, \xi_n)\|^2}{S_n}. \tag{12}$$

Then we can acquire

$$\mathbf{1}_{\mathcal{B}_n} \frac{\tilde{d}_0 \alpha_0^2 \|\nabla g(\theta_n)\| \cdot \|\nabla g(\theta_n, \xi_n)\|^2}{S_n} \leq 2\hat{d}_0^2 \alpha_0^3 \hat{K}^2 \mathbf{1}_{\mathcal{B}_n} \frac{\|\nabla g(\theta_n, \xi_n)\|^2}{S_n^{\frac{3}{2}}}$$

$$+ \frac{\alpha_0}{2\hat{K}^2} \mathbf{1}_{\mathcal{B}_n} \frac{\|\nabla g(\theta_n)\|^2 \|\nabla g(\theta_n, \xi_n)\|^2}{\sqrt{S_n}}.$$

Substitute above inequity into *Equation* 11, and make the mathematical expectation, getting

$$\mathbb{E}\left(\mathbf{1}_{\mathcal{B}_n}\big(g(\theta_{n+1}) - g(\theta_n)\big)\right) \leq -\frac{1}{2}\mathbb{E}\left(\mathbf{1}_{\mathcal{B}_n} \frac{\alpha_0 \|\nabla g(\theta_n)\|^2}{\sqrt{S_{n-1}}}\right) + t_0 \mathbb{E}\left(\frac{1}{\sqrt{S_{n-1}}} - \frac{1}{\sqrt{S_n}}\right)$$

$$+ \mathbb{E}\left(\frac{(\hat{c}\hat{K} + 2\hat{d}_0^2 \hat{K}^2)\|\nabla g(\theta_n, \xi_n)\|^2}{S_n^{\frac{3}{2}}}\right). \tag{13}$$

We make $\mathbb{E}\left(\mathbf{1}_{\mathcal{B}_n}\left(g(\theta_{n+1})\right)\right)$ to $\mathbb{E}\left(\mathbf{1}_{\mathcal{B}_{n+1}}\left(g(\theta_{n+1})\right)\right) + \mathbb{E}\left((\mathbf{1}_{\mathcal{B}_n} - \mathbf{1}_{\mathcal{B}_{n+1}})\left(g(\theta_{n+1})\right)\right)$, acquiring

$$
\begin{aligned}
&\mathbb{E}\left(\mathbf{1}_{\mathcal{B}_{n+1}}\left(g(\theta_{n+1})\right) - \mathbb{E}\left(\mathbf{1}_{\mathcal{B}_n} g(\theta_n)\right)\right) \\
&\leq -\frac{1}{2}\,\mathbb{E}\left(\mathbf{1}_{\mathcal{B}_n} \frac{\alpha_0 \|\nabla g(\theta_n)\|^2}{\sqrt{S_{n-1}}}\right) + t_0\,\mathbb{E}\left(\frac{1}{\sqrt{S_{n-1}}} - \frac{1}{\sqrt{S_n}}\right) \\
&\quad + \mathbb{E}\left(\frac{(\hat{c}\hat{K} + 2\hat{d}_0^2 \hat{K}^2)\|\nabla g(\theta_n, \xi_n)\|^2}{S_n^{\frac{3}{2}}}\right) - \mathbb{E}\left((\mathbf{1}_{\mathcal{B}_n} - \mathbf{1}_{\mathcal{B}_{n+1}})\left(g(\theta_{n+1})\right)\right).
\end{aligned}
\tag{14}
$$

We notice

$$
\begin{aligned}
&-\mathbb{E}\left((\mathbf{1}_{\mathcal{B}_n} - \mathbf{1}_{\mathcal{B}_{n+1}}) g(\theta_{n+1})\right) \\
&= -\mathbb{E}\left((\mathbf{1}_{\mathcal{B}_n} - \mathbf{1}_{\mathcal{B}_n}\mathbf{1}_{\mathcal{B}_{n+1}}) g(\theta_{n+1})\right) - \mathbb{E}\left((\mathbf{1}_{\mathcal{B}_n}\mathbf{1}_{\mathcal{B}_{n+1}} - \mathbf{1}_{\mathcal{B}_{n+1}}) g(\theta_{n+1})\right) \\
&\leq \min\{\delta_0, \delta_1\} \cdot \mathbb{E}\left((\mathbf{1}_{\mathcal{B}_n} - \mathbf{1}_{\mathcal{B}_n}\mathbf{1}_{\mathcal{B}_{n+1}})\right) + \min\{\delta_0, \delta_1\} \cdot \mathbb{E}\left((\mathbf{1}_{\mathcal{B}_n}\mathbf{1}_{\mathcal{B}_{n+1}} - \mathbf{1}_{\mathcal{B}_{n+1}})\right) \\
&= \min\{\delta_0, \delta_1\} \cdot \mathbb{E}\left(\mathbf{1}_{\mathcal{B}_n} - \mathbf{1}_{\mathcal{B}_{n+1}}\right).
\end{aligned}
$$

we getting

$$
\mathbb{E}\left(\mathbf{1}_{\mathcal{B}_{n+1}}\left(g(\theta_{n+1})\right) - \mathbb{E}\left(\mathbf{1}_{\mathcal{B}_n} g(\theta_n)\right)\right) \leq -\frac{1}{2}\,\mathbb{E}\left(\mathbf{1}_{\mathcal{B}_n} \frac{\alpha_0 \|\nabla g(\theta_n)\|^2}{\sqrt{S_{n-1}}}\right) + t_0\,\mathbb{E}\left(\frac{1}{\sqrt{S_{n-1}}} - \frac{1}{\sqrt{S_n}}\right)
$$

$$
+ \mathbb{E}\left(\frac{(\hat{c}\hat{K} + 2\hat{d}_0^2 \hat{K}^2)\|\nabla g(\theta_n, \xi_n)\|^2}{S_n^{\frac{3}{2}}}\right) - \min\{\delta_0, \delta_1\} \cdot \mathbb{E}\left(\mathbf{1}_{\mathcal{B}_n} - \mathbf{1}_{\mathcal{B}_{n+1}}\right).
$$

Then we make a sum, acquiring

$$
\sum_{n=1}^{+\infty} \mathbb{E}\left(\mathbf{1}_{\mathcal{B}_n} \frac{\alpha_0 \|\nabla g(\theta_n)\|^2}{\sqrt{S_{n-1}}}\right) < +\infty.
\tag{15}
$$

Then we consider the case when $\mathcal{B}_n^{(-)}$ occurs. We know $\nabla g$ must hold the Lipschitz condition; we assign its Lipschitz constant as $c$. Then we get

$$
\mathbf{1}_{\mathcal{B}_n^{(-)}}\left(g(\theta_{n+1}) - g(\theta_n)\right) \leq -\mathbf{1}_{\mathcal{B}_n^{(-)}} \alpha_0 S_n^{-\frac{1}{2}} \nabla g(\theta_n)^\top \nabla g(\theta_n, \xi_n) + \mathbf{1}_{\mathcal{B}_n^{(-)}} \frac{\alpha_0^2 c \|\nabla g(\theta_n, \xi_n)\|^2}{S_n}.
$$

First, we have

$$
\begin{aligned}
&\mathbf{1}_{\mathcal{B}_n}^{(-)}\left(g(\theta_{n+1}) - g(\theta_n)\right) \leq -\mathbf{1}_{\mathcal{B}_n}^{(-)} \frac{\alpha_0 \nabla g(\theta_n)^T \nabla g(\theta_n, \xi_n)}{\sqrt{S_n}} + \mathbf{1}_{\mathcal{B}_n}^{(-)} \frac{c\alpha_0^2}{2} \frac{\|\nabla g(\theta_n, \xi_n)\|^2}{S_n} \\
&\leq -\mathbf{1}_{\mathcal{B}_n}^{(-)} \frac{\alpha_0}{2}\left(\frac{1}{M+1} \frac{\|\nabla g(\theta_n, \xi_n)\|^2}{\sqrt{S_n}} + (M+1)\frac{\|\nabla g(\theta_n)\|^2}{\sqrt{S_n}}\right) \\
&\quad + \mathbf{1}_{\mathcal{B}_n}^{(-)} \frac{\alpha_0}{2} \frac{1}{\sqrt{S_{n-1}}}\left\|\frac{1}{\sqrt{M+1}}\nabla g(\theta_n, \xi_n) - \sqrt{M+1}\nabla g(\theta_n)\right\|^2 + \mathbf{1}_{\mathcal{B}_n}^{(-)} \frac{c\alpha_0^2}{2} \frac{\|\nabla g(\theta_n, \xi_n)\|^2}{S_n} \\
&\leq \mathbf{1}_{\mathcal{B}_n}^{(-)} \frac{\alpha_0}{2}(M+1)\left(\frac{\|\nabla g(\theta_{n-1})\|^2}{\sqrt{S_{n-1}}} - \frac{\|\nabla g(\theta_n)\|^2}{\sqrt{S_n}}\right) \\
&\quad + \mathbf{1}_{\mathcal{B}_n}^{(-)} \frac{\alpha_0}{2}\left(\frac{1}{M+1} \frac{\|\nabla g(\theta_n, \xi_n)\|^2}{\sqrt{S_{n-1}}} + \mathbf{1}_{\mathcal{B}_n}^{(-)} \frac{(M-1)\|\nabla g(\theta_n)\|^2}{\sqrt{S_{n-1}}} - \frac{(M+1)\|\nabla g(\theta_{n-1})\|^2}{\sqrt{S_{n-1}}}\right) \\
&\quad + \mathbf{1}_{\mathcal{B}_n}^{(-)} \frac{c\alpha_0^2}{2} \frac{\|\nabla g(\theta_n, \xi_n)\|^2}{S_n} + X_n,
\end{aligned}
\tag{16}
$$

where $X_n$ is defined as follow

$$
X_n := \mathbf{1}_{\mathcal{B}_n}^{(-)} \frac{\alpha_0}{\sqrt{S_{n-1}}} \nabla g(\theta_n)^T\left(\nabla g(\theta_n) - \nabla g(\theta_n, \xi_n)\right),
$$

and $M := 2M_0 + 2(a/k_1^2\delta_2^2) - 1$. Then we can find

$$
\begin{aligned}
\left\|\nabla g(\theta_n)\right\|^2 &= \left\|\nabla g(\theta_{n-1}) + \big(\nabla g(\theta_n) - \nabla g(\theta_{n-1})\big)\right\|^2 \\
&\leq \left\|\nabla g(\theta_{n-1})\right\|^2 + \frac{2\alpha_0 c}{\sqrt{S_{n-1}}}\left\|\nabla g(\theta_{n-1})\right\|\left\|\nabla g(\theta_{n-1},\xi_{n-1})\right\| + c^2\alpha_0^2\frac{\left\|\nabla g(\theta_{n-1},\xi_{n-1})\right\|^2}{S_{n-1}}.
\end{aligned}
\tag{17}
$$

Then we multiple $M + \frac{1}{2}$ on the both side of above inequity, acquiring

$$
\begin{aligned}
\left(M + \frac{1}{2}\right)\left\|\nabla g(\theta_n)\right\|^2 &\leq \left(M + \frac{1}{2}\right)\left\|\nabla g(\theta_{n-1})\right\|^2 + \left(M + \frac{1}{2}\right)\frac{2\alpha_0 c}{\sqrt{S_{n-1}}}\left\|\nabla g(\theta_{n-1})\right\|\left\|\nabla g(\theta_{n-1},\xi_{n-1})\right\| \\
&\quad + \left(M + \frac{1}{2}\right)c^2\alpha_0^2\frac{\left\|\nabla g(\theta_{n-1},\xi_{n-1})\right\|^2}{S_{n-1}}.
\end{aligned}
$$

Noting

$$
\left(M+\frac{1}{2}\right)\frac{2\alpha_0 c}{\sqrt{S_{n-1}}}\left\|\nabla g(\theta_{n-1})\right\|\left\|\nabla g(\theta_{n-1},\xi_{n-1})\right\| \leq \frac{1}{2}\left\|\nabla g(\theta_{n-1})\right\|^2 + 2\left(M+\frac{1}{2}\right)^2\alpha_0^2 c^2\frac{\left\|\nabla g(\theta_{n-1},\xi_{n-1})\right\|^2}{S_{n-1}}.
$$

We get that

$$
\left(M+\frac{1}{2}\right)\left\|\nabla g(\theta_n)\right\|^2 \leq (M+1)\left\|\nabla g(\theta_{n-1})\right\|^2 + \left(2\left(M+\frac{1}{2}\right)^2\alpha_0^2 c^2 + \left(M+\frac{1}{2}\right)c^2\alpha_0^2\right)\frac{\left\|\nabla g(\theta_{n-1},\xi_{n-1})\right\|^2}{S_{n-1}},
$$

that is

$$
\begin{aligned}
&(M-1)\left\|\nabla g(\theta_n)\right\|^2 + \frac{M_0 + \frac{a}{k_1^2\delta_2^2}}{M+1}\left\|\nabla g(\theta_n)\right\|^2 \\
&\leq -\left\|\nabla g(\theta_n)\right\|^2 + (M+1)\left\|\nabla g(\theta_{n-1})\right\|^2 + \left(2\left(M+\frac{1}{2}\right)^2\alpha_0^2 c^2 + \left(M+\frac{1}{2}\right)c^2\alpha_0^2\right)\frac{\left\|\nabla g(\theta_{n-1},\xi_{n-1})\right\|^2}{S_{n-1}}.
\end{aligned}
$$

Then we multiple $\mathbf{1}_{\mathcal{B}_n}^{(-)}/\sqrt{S_{n-1}}$ on both side of above inequity, and noting where $\mathbf{1}_{\mathcal{B}_n}^{(-)} = 1$, there is

$$
\frac{M_0 + \frac{a}{k_1^2\delta_2^2}}{M+1}\left\|\nabla g(\theta_n)\right\|^2 \geq \frac{1}{M+1}\mathbb{E}(\|\nabla g(\theta_n,\xi_n)\|^2|\mathcal{F}_n),
$$

getting

$$
\begin{aligned}
&(M-1)\,\mathbb{E}\left(\mathbf{1}_{\mathcal{B}_n}^{(-)}\frac{\left\|\nabla g(\theta_n)\right\|^2}{\sqrt{S_{n-1}}}\right) + \frac{1}{M+1}\,\mathbb{E}\left(\mathbf{1}_{\mathcal{B}_n}^{(-)}\frac{\left\|\nabla g(\theta_n,\xi_n)\right\|^2}{S_{n-1}}\right) \\
&\leq -\mathbb{E}\left(\mathbf{1}_{\mathcal{B}_n}^{(-)}\frac{\left\|\nabla g(\theta_n)\right\|^2}{\sqrt{S_{n-1}}}\right) + (M+1)\,\mathbb{E}\left(\mathbf{1}_{\mathcal{B}_n}^{(-)}\frac{\left\|\nabla g(\theta_{n-1})\right\|^2}{\sqrt{S_{n-1}}}\right) \\
&\quad + \left(2\left(M+\frac{1}{2}\right)^2\alpha_0^2 c^2 + \left(M+\frac{1}{2}\right)c^2\alpha_0^2\right)\mathbb{E}\left(\mathbf{1}_{\mathcal{B}_n}^{(-)}\frac{\left\|\nabla g(\theta_{n-1},\xi_{n-1})\right\|^2}{S_{n-1}^{\frac{3}{2}}}\right).
\end{aligned}
\tag{18}
$$

Substitute it into *Equation* 16, we get

$$
\begin{aligned}
&\mathbb{E}\left(\mathbf{1}_{\mathcal{B}_{n+1}}^{(-)}\big(g(\theta_{n+1})\big)\right) - \mathbb{E}\left(\mathbf{1}_{\mathcal{B}_n}^{(-)}g(\theta_n)\right) \leq -\frac{\alpha_0}{2}\,\mathbb{E}\left(\mathbf{1}_{\mathcal{B}_n}^{(-)}\frac{\left\|\nabla g(\theta_n)\right\|^2}{\sqrt{S_{n-1}}}\right) \\
&\quad + \left(2\left(M+\frac{1}{2}\right)^2\alpha_0^2 c^2 + \left(M+\frac{1}{2}\right)c^2\alpha_0^2\right)\mathbb{E}\left(\mathbf{1}_{\mathcal{B}_n}^{(-)}\frac{\left\|\nabla g(\theta_{n-1},\xi_{n-1})\right\|^2}{S_{n-1}^{\frac{3}{2}}}\right) \\
&\quad + \mathbb{E}\left(\mathbf{1}_{\mathcal{B}_n}^{(-)}\frac{\alpha_0}{2}(M+1)\left(\frac{\left\|\nabla g(\theta_{n-1})\right\|^2}{\sqrt{S_{n-1}}} - \frac{\left\|\nabla g(\theta_n)\right\|^2}{\sqrt{S_n}}\right)\right) + \frac{c\alpha_0^2}{2}\,\mathbb{E}\left(\frac{\left\|\nabla g(\theta_n,\xi_n)\right\|^2}{S_n}\right) \\
&\quad + \mathbb{E}\left((\mathbf{1}_{\mathcal{B}_n} - \mathbf{1}_{\mathcal{B}_{n+1}})\big(g(\theta_{n+1})\big)\right).
\end{aligned}
\tag{19}
$$

Then we use inequality $2a^T b \leq \lambda\|a\|^2 + \frac{1}{\lambda}\|b\|^2$ $(\lambda > 0)$ on *Equation* 17 to get

$$
\begin{aligned}
\left\|\nabla g(\theta_n)\right\|^2 - \left\|\nabla g(\theta_{n-1})\right\|^2 &\leq \frac{\left\|\nabla g(\theta_{n-1})\right\|^2}{2(M+1)} \\
&\quad + \frac{2\alpha_0^2 c^2(M+1)}{S_{n-1}}\left\|\nabla g(\theta_{n-1},\xi_{n-1})\right\|^2 + \frac{\alpha_0^2 c^2}{S_{n-1}}\left\|\nabla g(\theta_{n-1},\xi_{n-1})\right\|^2.
\end{aligned}
\tag{20}
$$

Multiple both sides of *Equation* 20 by $\mathbf{1}_{\mathcal{B}_n}^{(-)}/\sqrt{S_{n-1}}$ and notice $S_{n-2} \le S_{n-1} \le S_n$, then we have

$$
\begin{aligned}
\mathbf{1}_{\mathcal{B}_n}^{(-)} &\left( \frac{\left\| \nabla g(\theta_n) \right\|^2}{\sqrt{S_n}} - \frac{\left\| \nabla g(\theta_{n-1}) \right\|^2}{\sqrt{S_{n-1}}} \right) \\
&\le \frac{\mathbf{1}_{\mathcal{B}_n}^{(-)} \left\| \nabla g(\theta_{n-1}) \right\|^2}{2(M+1)\sqrt{S_{n-2}}} + \frac{\alpha_0^2 c^2 (2M+3) \mathbf{1}_{\mathcal{B}_n}^{(-)}}{S_{n-1}^{\frac{3}{2}}} \left\| \nabla g(\theta_{n-1}, \xi_{n-1}) \right\|^2 \\
&\le \frac{\mathbf{1}_{\mathcal{B}_n}^{(-)} \mathbf{1}_{\mathcal{B}_{n-1}}^{(-)} \left\| \nabla g(\theta_{n-1}) \right\|^2}{2(M+1)\sqrt{S_{n-2}}} + (\mathbf{1}_{\mathcal{B}_n}^{(-)} - \mathbf{1}_{\mathcal{B}_{n-1}}^{(-)} \mathbf{1}_{\mathcal{B}_n}^{(-)}) \frac{\left\| \nabla g(\theta_{n-1}) \right\|^2}{2(M+1)\sqrt{S_{n-2}}} \\
&\quad + \frac{\alpha_0^2 c^2 (2M+3) \mathbf{1}_{\mathcal{B}_n}^{(-)}}{S_{n-1}^{\frac{3}{2}}} \left\| \nabla g(\theta_{n-1}, \xi_{n-1}) \right\|^2 \\
&\le \frac{\mathbf{1}_{\mathcal{B}_{n-1}}^{(-)} \left\| \nabla g(\theta_{n-1}) \right\|^2}{2(M+1)\sqrt{S_{n-2}}} + \mathbf{1}_{\mathcal{B}_{n-1}} \frac{\left\| \nabla g(\theta_{n-1}) \right\|^2}{2(M+1)\sqrt{S_{n-2}}} \\
&\quad + \frac{\alpha_0^2 c^2 (2M+3) \mathbf{1}_{\mathcal{B}_n}^{(-)}}{S_{n-1}^{\frac{3}{2}}} \left\| \nabla g(\theta_{n-1}, \xi_{n-1}) \right\|^2 .
\end{aligned}
\tag{21}
$$

Substitute *Equation* 21 into *Equation* 18, acquiring

$$
\begin{aligned}
\mathbb{E} \left( \mathbf{1}_{\mathcal{B}_{n+1}}^{(-)} \left( g(\theta_{n+1}) \right) - \mathbb{E} \left( \mathbf{1}_{\mathcal{B}_n}^{(-)} g(\theta_n) \right) \right) &\le -\frac{\alpha_0}{4} \mathbb{E} \left( \mathbf{1}_{\mathcal{B}_n}^{(-)} \frac{\left\| \nabla g(\theta_n) \right\|^2}{\sqrt{S_{n-1}}} \right) \\
+ \left( 2\left(M+\frac{1}{2}\right)^2 \alpha_0^2 c^2 + \left(M+\frac{1}{2}\right) c^2 \alpha_0^2 \right) \mathbb{E} &\left( \mathbf{1}_{\mathcal{B}_n}^{(-)} \frac{\left\| \nabla g(\theta_{n-1}, \xi_{n-1}) \right\|^2}{S_{n-1}^{\frac{3}{2}}} \right) \\
+ \frac{c\alpha_0^2}{2} \mathbb{E} \left( \frac{\|\nabla g(\theta_n, \xi_n)\|^2}{S_n} \right) + \mathbb{E} &\left( \frac{\alpha_0^2 c^2 (2M+3)(M+1) \mathbf{1}_{\mathcal{B}_n}^{(-)}}{2 S_{n-1}^{\frac{3}{2}}} \left\| \nabla g(\theta_{n-1}, \xi_{n-1}) \right\|^2 \right) \\
+ \mathbb{E} \left( (\mathbf{1}_{\mathcal{B}_n} - \mathbf{1}_{\mathcal{B}_{n+1}}) \left( g(\theta_{n+1}) \right) \right) + \mathbb{E} &\left( \mathbf{1}_{\mathcal{B}_{n-1}} \frac{\left\| \nabla g(\theta_{n-1}) \right\|^2}{2(M+1)\sqrt{S_{n-2}}} \right).
\end{aligned}
\tag{22}
$$

We know when $\mathcal{B}_n^{(-)}$ occurs, through Lemma A.8, there is $\|\nabla g(\theta)\| > \delta_2' := k_1 \cdot \min\{\delta_0, \delta_1\}$. That means

$$
\mathbb{E} \left( \|\nabla g(\theta_n, \xi_n)\|^2 \big| \mathcal{F}_n \right) \le M \|\nabla g(\theta_n)\|^2 + a \le \left( M + \frac{a}{\delta_2'^2} \right) \|\nabla g(\theta_n)\|^2 .
$$

so we get

$$
\sum_{k=1}^n \mathbb{E} \left( \mathbf{1}_{B_n^{(-)\prime}} \frac{\left\| \nabla g(\theta_k, \xi_k) \right\|^2}{S_k} \right) \le \left( M + \frac{a}{\delta_2^2} \right) \mathbb{E} \left( \frac{\left\| \nabla g(\theta_k) \right\|^2}{S_k} \right).
$$

Through Lemma A.3, we can get

$$
\mathbb{E} \left( \frac{\left\| \nabla g(\theta_k) \right\|^2}{S_k} \right) < \tilde{o} \, \mathbb{E} \left( \frac{\left\| \nabla g(\theta_k) \right\|^2}{S_k^{\frac{3}{4}}} \right) < +\infty .
$$

We back to *Equation* 14, we can get

$$
\begin{aligned}
\mathbb{E} \left( (\mathbf{1}_{\mathcal{B}_n} - \mathbf{1}_{\mathcal{B}_{n+1}}) \left( g(\theta_{n+1}) \right) \right. &\le \mathbb{E} \left( \mathbf{1}_{\mathcal{B}_n} g(\theta_n) \right) - \mathbb{E} \left( \mathbf{1}_{\mathcal{B}_{n+1}} \left( g(\theta_{n+1}) \right) \right) + t_0 \mathbb{E} \left( \frac{1}{\sqrt{S_{n-1}}} - \frac{1}{\sqrt{S_n}} \right) \\
&+ \mathbb{E} \left( \frac{(\hat{c}\hat{K} + 2\hat{d}_0^2 \hat{K}^2) \|\nabla g(\theta_n, \xi_n)\|^2}{S_n^{\frac{3}{2}}} \right),
\end{aligned}
\tag{23}
$$

which means

$$\sum_{n=1}^{+\infty} \mathbb{E}\left((\mathbf{1}_{\mathcal{B}_n} - \mathbf{1}_{\mathcal{B}_{n+1}})(g(\theta_{n+1}))\right) < +\infty \tag{24}$$

Substitute *Equation* 23 and *Equation* 24 into *Equation* 22, and make a sum, we get

$$\sum_{n=1}^{+\infty} \mathbb{E}\left(\mathbf{1}_{\mathcal{B}_n}^{(-)} \frac{\alpha_0 \|\nabla g(\theta_n)\|^2}{\sqrt{S_{n-1}}}\right) < +\infty. \tag{25}$$

Combine *Equation* 15 and *Equation* 25, we get

$$\sum_{n=1}^{+\infty} \mathbb{E}\left(\frac{\alpha_0 \|\nabla g(\theta_n)\|^2}{\sqrt{S_{n-1}}}\right) < +\infty.$$

With this, we complete the result. □

## B.2  THE PROOF OF LEMMA A.8

*Proof.* We can get

$$\hat{g}(\theta) = -\frac{1}{N} \sum_{i=1}^{N} \left(\hat{y}_i \ln\left(\frac{1}{1 + e^{\mathrm{sgn}(\hat{y}_i - 0.5)\theta^\top \hat{x}_i}}\right) + (1 - \hat{y}_i) \ln\left(1 - \frac{1}{1 + e^{\mathrm{sgn}(\hat{y}_i - 0.5)\theta^\top \hat{x}_i}}\right)\right).$$

Due to $\theta$ is a margin vector, we can get

$$\hat{g}(\theta) = -\frac{1}{N} \sum_{i=1}^{N} \ln\left(1 - \frac{1}{1 + e^{|\theta^\top \hat{x}_i|}}\right).$$

Since $1/(1 + e^{|\theta^\top \hat{x}_i|}) \in (0, 1/2)$, we can get following inequality

$$-\frac{2\ln 2}{1 + e^{|\theta^\top \hat{x}_i|}} \le \ln\left(1 - \frac{1}{1 + e^{|\theta^\top \hat{x}_i|}}\right) \le -\frac{1}{1 + e^{|\theta^\top \hat{x}_i|}}.$$

That means

$$\frac{1}{N} \sum_{i=1}^{N} \frac{1}{1 + e^{|\theta^\top \hat{x}_i|}} \le \hat{g}(\theta) \le \frac{2\ln 2}{N} \sum_{i=1}^{N} \frac{1}{1 + e^{|\theta^\top \hat{x}_i|}}. \tag{26}$$

On the other hand, we can calculate

$$\|\nabla \hat{g}(\theta)\| = \frac{1}{N} \left\| \sum_{i=1}^{N} \left(\frac{1}{1 + e^{-\theta^\top \hat{x}_i}} - y_i\right) \hat{x}_i \right\|.$$

Due to $\theta$ is a margin vector, we can get

$$\|\nabla \hat{g}(\theta)\| = \frac{1}{N} \left\| \sum_{i=1}^{N} \frac{-\mathrm{sgn}(y_i - 0.5)}{1 + e^{|\theta^\top \hat{x}_i|}} \hat{x}_i \right\|.$$

First, we use the norm inequality, getting

$$\|\nabla \hat{g}(\theta)\| = \frac{1}{N} \left\| \sum_{i=1}^{N} \frac{-\mathrm{sgn}(y_i - 0.5)}{1 + e^{|\theta^\top \hat{x}_i|}} \hat{x}_i \right\| \le \frac{\max_{1 \le i \le N}\{\|\hat{x}_i\|\}}{N} \cdot \sum_{i=1}^{N} \frac{1}{1 + e^{|\theta^\top \hat{x}_i|}}.$$

Second, we assume $\theta^*/\|\theta^*\|$ is the max margin vector of this data set, we getting

$$\|\nabla g(\theta)\| = \frac{1}{N} \left\| \sum_{i=1}^{N} \frac{-\mathrm{sgn}(y_i - 0.5)}{1 + e^{|\theta^\top \hat{x}_i|}} \hat{x}_i \right\| \ge \frac{1}{N} \left( \sum_{i=1}^{N} \frac{1}{1 + e^{|\theta^\top \hat{x}_i|}} \left| \frac{\theta^{*\top} \hat{x}_i}{\|\theta^*\|} \right| \right)$$

$$\ge \frac{d^*}{N} \sum_{i=1}^{N} \frac{1}{1 + e^{|\theta^\top \hat{x}_i|}}.$$

Then we can get

$$\frac{d^*}{N} \sum_{i=1}^{N} \frac{1}{1 + e^{|\theta^\top \hat{x}_i|}} \le \|\nabla g(\theta)\| \le \frac{\max_{1 \le i \le N}\{\|\hat{x}_i\|\}}{N} \cdot \sum_{i=1}^{N} \frac{1}{1 + e^{|\theta^\top \hat{x}_i|}}. \tag{27}$$

Combining *Equation* 26 and *Equation* 27, we can get the result.

### B.3 THE PROOF OF LEMMA A.10

Due to $\theta$ is not a margin vector of the data set $\{x_i, y_i\}_{i=1}^{N}$, we know that there is at least one data $(x_j, y_j)$ has a wrong classification which is formed by $\theta$. That means

$$-y_j \ln(\hat{y}_j) - (1 - y_j) \ln(1 - \hat{y}_j) > \ln 2,$$

so we can get

$$g(\theta) = -\frac{1}{N} \sum_{i=1}^{N} \big(y_i \ln(\hat{y}_i) + (1 - y_i) \ln(1 - \hat{y}_i)\big) > \frac{\ln 2}{N}.$$

$\square$

### B.4 THE PROOF OF LEMMA A.6

*Proof.* Obviously, since $\theta^* := \arg\min_{\{\theta \| |\theta^\top x_1| = |\theta^\top x_2|\}} \|\theta\|$, we can get $\mathrm{rank}\{x_1, x_2\} = \mathrm{rank}\{x_1, x_2, \theta^*\}$. Then we assign $S := \mathrm{span}\{x_1, x_2\}$. For any vector $\theta$, we assign the vector which $\theta$ projects on $S$ as $\theta'$. Without loss of generality, we can think $\theta^{*\top} x_1 = \theta^{*\top} x_2 > 0$. (if $\theta^{*\top} x_i < 0$, we can construct a new vector $x_i' := -x_i$ to substitute $x_i$.) Then we assign

$$\varphi := \arccos \frac{\theta'^\top \theta^*}{\|\theta'\| \|\theta^*\|},$$

$$\varphi_1 := \arccos \frac{\theta^{*\top} x_1}{\|\theta^*\| \|x_1\|}, \ \varphi_2 := \arccos \frac{\theta^{*\top} x_2}{\|\theta^*\| \|x_2\|},$$

$$\phi := \arccos \frac{\theta^\top \theta'}{\|\theta\| \|\theta'\|},$$

In order to prove *Equation* 10, we just need to prove exists $\delta_0' > 0$, making the binary function

$$D(\varphi, \phi) = \frac{\|x_1\|}{\|x_2\|} \cdot \frac{\big|\cos(\varphi_1 - \varphi)\cos(\phi) - \cos(\varphi_1)\big|}{\cos(\varphi_2) - \cos(\varphi_2 + \varphi)\cos(\phi)} < \hat{r}, \ (\forall\, 0 < \varphi, \ \phi < \delta_0'). \qquad (28)$$

Absolutely, when $\varphi$ and $\phi$ are small enough, we can cancel the absolute value, i.e.,

$$D(\varphi, \phi) = \frac{\|x_1\|}{\|x_2\|} \cdot \frac{\big|\cos(\varphi_1 - \varphi)\cos(\phi) - \cos(\varphi_1)\big|}{\cos(\varphi_2) - \cos(\varphi_2 + \varphi)\cos(\phi)}.$$

That means

$$\limsup_{\varphi \to 0, \phi \to 0} D(\varphi, \phi)$$

$$\leq \limsup_{\varphi \to 0, \phi \to 0} \frac{\|x_1\|}{\|x_2\|} \cdot \frac{\big|\cos(\varphi_1 - \varphi)\cos(\phi) - \cos(\varphi_1)\cos(\phi)\big| + \big|\cos(\varphi_1)\cos(\phi) - \cos(\varphi_1)\big|}{\cos(\varphi_2) - \cos(\varphi_2 + \varphi)\cos(\phi)}$$

$$\leq \frac{\|x_1\|}{\|x_2\|} \cdot \max\left\{\frac{\sin(\varphi_1)}{\sin(\varphi_2)}, \frac{\cos(\varphi_1)}{\cos(\varphi_2)}\right\}.$$

That means we can take

$$\hat{r} := 2\frac{\|x_1\|}{\|x_2\|} \cdot \max\left\{\frac{\sin(\varphi_1)}{\sin(\varphi_2)}, \frac{\cos(\varphi_1)}{\cos(\varphi_2)}\right\}$$

to make *Equation* 10 holding.

$\square$

### B.5 THE PROOF OF LEMMA 4.1

*Proof.* The proof is similar to those to obtain the arguments in the proof of Lemma A.6.

$\square$

### B.6 PROOF OF LEMMA 4.2

*Proof.* Given two unary function $y_1(x) = -1/\alpha |\ln x|^\alpha$ $(0 < x < 1/4)$, $y_2(x) = 1$ $(x > 1/2)$. We know that there is a smooth connecting function $y_3(x)$ $(1/4 \le x \le 1/2)$, making the following function

$$y(x) = \begin{cases} -1/\alpha |\ln x|^\alpha, & \text{if } x < \frac{\ln 2}{N} \\ 1, & \text{if } x > 1 \\ y_3(x), & \text{if } \frac{\ln 2}{N} \le x \le 1 \end{cases}$$

is an infinite order continuous function.

We construct a function

$$h(\theta) := y(g(\theta)), \tag{29}$$

and a set $S^{(\hat\delta)} := \{\theta | 0 < g(\theta) < \hat\delta\}$. We make $\hat\delta = (\ln 2)/N$. Then we use the *taylor expansion* and the structure of $g$, getting that for any $\theta^{(1)} \in S^{(\hat\delta)}$ and $\theta^{(2)} \in \mathbb{R}^d$, there exists three positive constants $d_0$, $d_1$ and $d_2$, making

$$
\begin{aligned}
h(\theta^{(2)}) - h(\theta^{(1)}) \le{} & \nabla h(\theta^{(1)})^\top (\theta^{(2)} - \theta^{(1)}) + \frac{d_0}{\|\theta^{(1)}\|^2} \|\theta^{(2)} - \theta^{(1)}\|^2 \\
& + c_0 \|\theta^{(2)} - \theta^{(1)}\|^3,
\end{aligned}
\tag{30}
$$

where $\hat{c}$ is a constant that can not affect the result. For convenience, we assign

$$T_n := \frac{d_0 \alpha_0^2}{\|\theta_n\|^{1+\alpha}}.$$

We construct an event $\mathcal{A}_n := \{\theta_n \in S^{(\hat\delta)}\}$ $(\hat\delta = (\ln 2)/N)$. Combining *Equation 30*, we get

$$
\begin{aligned}
& \mathbf{1}_{\mathcal{A}_n}\big(h(\theta_{n+1}) - h(\theta_n)\big) \le \mathbf{1}_{\mathcal{A}_n} \nabla h(\theta_n)^\top (\theta_{n+1} - \theta_n) + \mathbf{1}_{\mathcal{A}_n} T_n \|\theta_{n+1} - \theta_n\|^2 \\
& + \hat{c}\|\theta_{n+1} - \theta_n\|^3 \\
& = -\mathbf{1}_{\mathcal{A}_n} \frac{\big(\nabla g(\theta_n)\big)^\top \nabla g(\theta_n, \xi_n)}{\sqrt{S_n} g(\theta_n) |\ln(g(\theta_n))|^{1+\alpha}} + \mathbf{1}_{\mathcal{A}_n} \frac{T_n \|\nabla g(\theta_n, \xi_n)\|^2}{S_n} + \mathbf{1}_{\mathcal{A}_n} \frac{c_0 \alpha_0^3 \|\nabla g(\theta_n, \xi_n)\|^3}{S_n^{\frac{3}{2}}}.
\end{aligned}
\tag{31}
$$

Then we get

$$
\begin{aligned}
& \mathbf{1}_{\mathcal{A}_n}\big(h(\theta_{n+1}) - h(\theta_n)\big) \\
& \le -\mathbf{1}_{\mathcal{A}_n} \frac{\big(\nabla g(\theta_n)\big)^\top \nabla g(\theta_n, \xi_n)}{\sqrt{S_n} g(\theta_n) |\ln(g(\theta_n))|^{1+\alpha}} + \mathbf{1}_{\mathcal{A}_n} \frac{T_n \|\nabla g(\theta_n, \xi_n)\|^2}{S_n} + \mathbf{1}_{\mathcal{A}_n} \frac{c_0 \alpha_0^3 \|\nabla g(\theta_n, \xi_n)\|^3}{S_n^{\frac{3}{2}}} \\
& = -\mathbf{1}_{\mathcal{A}_n} \frac{\big(\nabla g(\theta_n)\big)^\top \nabla g(\theta_n, \xi_n)}{\sqrt{S_{n-1}} g(\theta_n) |\ln(g(\theta_n))|^{1+\alpha}} + \mathbf{1}_{\mathcal{A}_n} \frac{\big(\nabla g(\theta_n)\big)^\top \nabla g(\theta_n, \xi_n)}{g(\theta_n) |\ln(g(\theta_n))|^{1+\alpha}} \left( \frac{1}{\sqrt{S_{n-1}}} - \frac{1}{\sqrt{S_n}} \right) \\
& + \mathbf{1}_{\mathcal{A}_n} \frac{T_n \|\nabla g(\theta_n, \xi_n)\|^2}{S_n} + \mathbf{1}_{\mathcal{A}_n} \frac{c_0 \alpha_0^3 \|\nabla g(\theta_n, \xi_n)\|^3}{S_n^{\frac{3}{2}}}.
\end{aligned}
$$

Then we use an identical equation, i.e.,

$$\mathbf{1}_{\mathcal{A}_n} h(\theta_{n+1}) = \mathbf{1}_{\mathcal{A}_{n+1}} h(\theta_{n+1}) + \big(\mathbf{1}_{\mathcal{A}_n} - \mathbf{1}_{\mathcal{A}_{n+1}}\big) h(\theta_{n+1}),$$

getting

$$
\begin{aligned}
& \mathbf{1}_{\mathcal{A}_{n+1}} h(\theta_{n+1}) - \mathbf{1}_{\mathcal{A}_n} h(\theta_n) \\
& \le -\mathbf{1}_{\mathcal{A}_n} \frac{\big(\nabla g(\theta_n)\big)^\top \nabla g(\theta_n, \xi_n)}{\sqrt{S_{n-1}} g(\theta_n) |\ln(g(\theta_n))|^{1+\alpha}} + \mathbf{1}_{\mathcal{A}_n} \frac{\big(\nabla g(\theta_n)\big)^\top \nabla g(\theta_n, \xi_n)}{g(\theta_n) |\ln(g(\theta_n))|^{1+\alpha}} \left( \frac{1}{\sqrt{S_{n-1}}} - \frac{1}{\sqrt{S_n}} \right) \\
& + \mathbf{1}_{\mathcal{A}_n} \frac{T_n \|\nabla g(\theta_n, \xi_n)\|^2}{S_n} + \mathbf{1}_{\mathcal{A}_n} \frac{c_0 \alpha_0^3 \|\nabla g(\theta_n, \xi_n)\|^3}{S_n^{\frac{3}{2}}} - \big(\mathbf{1}_{\mathcal{A}_n} - \mathbf{1}_{\mathcal{A}_{n+1}}\big) h(\theta_{n+1}).
\end{aligned}
\tag{32}
$$

We make the mathematical expectation on the both side of *Equation* 32, getting

$$
\mathbb{E}\left(\mathbf{1}_{\mathcal{A}_{n+1}}h(\theta_{n+1})\right) - \mathbb{E}\left(\mathbf{1}_{\mathcal{A}_n}h(\theta_n)\right)
$$
$$
\leq -\mathbb{E}\left(\mathbf{1}_{\mathcal{A}_n}\frac{\left\|\nabla g(\theta_n)\right\|^2}{\sqrt{S_{n-1}}g(\theta_n)|\ln(g(\theta_n))|^{1+\alpha}}\right) + \mathbb{E}\left(\mathbf{1}_{\mathcal{A}_n}\frac{\left(\nabla g(\theta_n)\right)^\top \nabla g(\theta_n,\xi_n)}{g(\theta_n)|\ln(g(\theta_n))|^{1+\alpha}}\left(\frac{1}{\sqrt{S_{n-1}}} - \frac{1}{\sqrt{S_n}}\right)\right)
$$
$$
+ \mathbb{E}\left(\mathbf{1}_{\mathcal{A}_n}\frac{T_n\|\nabla g(\theta_n,\xi_n)\|^2}{S_n}\right) + \mathbb{E}\left(\mathbf{1}_{\mathcal{A}_n}\frac{c_0\alpha_0^3\|\nabla g(\theta_n,\xi_n)\|^3}{S_n^{\frac{3}{2}}}\right) - \mathbb{E}\left((\mathbf{1}_{\mathcal{A}_n} - \mathbf{1}_{\mathcal{A}_{n+1}})h(\theta_{n+1})\right).
$$
$$\tag{33}$$

For the second item in *Equation* 33 right, through Assumption 3.1, there is

$$
\mathbb{E}\left(\mathbf{1}_{\mathcal{A}_n}\frac{\left(\nabla g(\theta_n)\right)^\top \nabla g(\theta_n,\xi_n)}{g(\theta_n)|\ln(g(\theta_n))|^{1+\alpha}}\left(\frac{1}{\sqrt{S_{n-1}}} - \frac{1}{\sqrt{S_n}}\right)\right) \leq \tilde{\delta}_0\,\mathbb{E}\left(\frac{1}{\sqrt{S_{n-1}}} - \frac{1}{\sqrt{S_n}}\right). \tag{34}
$$

Next we get

$$
-\mathbb{E}\left((\mathbf{1}_{\mathcal{A}_n} - \mathbf{1}_{\mathcal{A}_{n+1}})h(\theta_{n+1})\right)
$$
$$
= -\mathbb{E}\left((\mathbf{1}_{\mathcal{A}_n} - \mathbf{1}_{\mathcal{A}_n}\mathbf{1}_{\mathcal{A}_{n+1}})h(\theta_{n+1})\right) - \mathbb{E}\left((\mathbf{1}_{\mathcal{A}_n}\mathbf{1}_{\mathcal{A}_{n+1}} - \mathbf{1}_{\mathcal{A}_{n+1}})h(\theta_{n+1})\right)
$$
$$
\leq \frac{1}{\ln\left(\min\{\hat{\delta},\frac{1}{2}\}\right)}\mathbb{E}\left((\mathbf{1}_{\mathcal{A}_n} - \mathbf{1}_{\mathcal{A}_n}\mathbf{1}_{\mathcal{A}_{n+1}})\right) + \frac{1}{\ln\left(\min\{\hat{\delta},\frac{1}{2}\}\right)}\mathbb{E}\left((\mathbf{1}_{\mathcal{A}_n}\mathbf{1}_{\mathcal{A}_{n+1}} - \mathbf{1}_{\mathcal{A}_{n+1}})\right)
$$
$$
= \frac{1}{\ln\left(\min\{\hat{\delta},\frac{1}{2}\}\right)}\mathbb{E}\left(\mathbf{1}_{\mathcal{A}_n} - \mathbf{1}_{\mathcal{A}_{n+1}}\right).
$$
$$\tag{35}$$

We make the sum of *Equation* 33, getting

$$
\mathbb{E}\left(\mathbf{1}_{\mathcal{A}_{n+1}}h(\theta_{n+1})\right) - \mathbb{E}\left(I_1 h(\theta_1)\right) \leq -\sum_{k=2}^{n}\mathbb{E}\left(\mathbf{1}_{A_k}\frac{\left\|\nabla g(\theta_k)\right\|^2}{\sqrt{S_{k-1}}g(\theta_k)|\ln(g(\theta_k))|^{1+\alpha}}\right)
$$
$$
+ \sum_{k=2}^{n}\mathbb{E}\left(\mathbf{1}_{A_k}\frac{\left(\nabla g(\theta_k)\right)^\top \nabla g(\theta_k,\xi_k)}{g(\theta_k)|\ln(g(\theta_k))|^{1+\alpha}}\left(\frac{1}{\sqrt{S_{k-1}}} - \frac{1}{\sqrt{S_k}}\right)\right)
$$
$$
+ \sum_{k=2}^{n}\mathbb{E}\left(\mathbf{1}_{A_k}\frac{T_k\|\nabla g(\theta_k,\xi_k)\|^2}{S_k}\right) + \sum_{k=1}^{n}\mathbb{E}\left(\mathbf{1}_{A_k}\frac{c_0\alpha_0^3\|\nabla g(\theta_k,\xi_k)\|^3}{S_k^{\frac{3}{2}}}\right)
$$
$$
- \sum_{k=1}^{n}\mathbb{E}\left((\mathbf{1}_{A_k} - \mathbf{1}_{\mathcal{A}_{k+1}})h(\theta_{k+1})\right).
$$

We can get that

$$
\sum_{n=2}^{+\infty}\mathbb{E}\left(\mathbf{1}_{\mathcal{A}_n}\frac{\left\|\nabla g(\theta_n)\right\|^2}{\sqrt{S_{n-1}}g(\theta_n)|\ln(g(\theta_n))|^{1+\alpha}}\right) \leq \mathbb{E}\left(I_1 h(\theta_1)\right)
$$
$$
+ \sum_{k=1}^{n}\mathbb{E}\left(\mathbf{1}_{A_k}\frac{\left(\nabla g(\theta_k)\right)^\top \nabla g(\theta_k,\xi_k)}{g(\theta_k)|\ln(g(\theta_k))|^{1+\alpha}}\left(\frac{1}{\sqrt{S_{k-1}}} - \frac{1}{\sqrt{S_k}}\right)\right)
$$
$$
+ \sum_{k=1}^{n}\mathbb{E}\left(\mathbf{1}_{A_k}\frac{T_k\|\nabla g(\theta_k,\xi_k)\|^2}{S_k}\right) + \sum_{k=1}^{n}\mathbb{E}\left(\mathbf{1}_{A_k}\frac{c_0\alpha_0\|\nabla g(\theta_k,\xi_k)\|^3}{S_k^{\frac{3}{2}}}\right)
$$
$$
- \sum_{k=1}^{n}\mathbb{E}\left((\mathbf{1}_{A_k} - \mathbf{1}_{\mathcal{A}_{k+1}})h(\theta_{k+1})\right).
$$
$$\tag{36}$$

For the third term in the right side of *Equation* 36, we have

$$\sum_{k=1}^{n} \mathbb{E}\left(\mathbf{1}_{A_k} \frac{T_k \|\nabla g(\theta_k, \xi_k)\|^2}{S_k}\right) = \sum_{k=1}^{n} \mathbb{E}\left(\mathbf{1}_{A_k} \mathbf{1}\left(\frac{1}{\sqrt{S_k}} < \tilde{k}g(\theta_k)\right) \frac{d_0 \|\nabla g(\theta_k, \xi_k)\|^2}{\|\theta_k\|^{1+\alpha} S_k}\right)$$

$$+ \sum_{k=1}^{n} \mathbb{E}\left(\mathbf{1}_{A_k} \mathbf{1}\left(\frac{1}{\sqrt{S_k}} \geq \tilde{k}g(\theta_k)\right) \frac{d_0 \|\nabla g(\theta_k, \xi_k)\|^2}{\|\theta_k\|^{1+\alpha} S_k}\right)$$

$$\leq \sum_{k=1}^{n} \mathbb{E}\left(\mathbf{1}_{A_k} \frac{d_0 \tilde{k}g(\theta_k)\|\nabla g(\theta_k, \xi_k)\|^2}{\|\theta_k\|^2 S_k}\right) + \sum_{k=1}^{n} \mathbb{E}\left(\frac{4d_0 \|\nabla g(\theta_k, \xi_k)\|^2}{S_k \ln^{1+\alpha} S_k}\right).$$

taking proper $\tilde{k}$, we can make

$$\sum_{k=1}^{n} \mathbb{E}\left(\mathbf{1}_{A_k} \frac{T_k \|\nabla g(\theta_k, \xi_k)\|^2}{S_k}\right)$$

$$\leq \frac{T_1 \|\nabla g(\theta_1, \xi_1)\|^2}{S_1} + \frac{1}{2} \sum_{k=2}^{n} \mathbb{E}\left(\mathbf{1}_{A_k} \frac{\|\nabla g(\theta_k)\|^2}{\sqrt{S_{k-1}}g(\theta_k)|\ln(g(\theta_k))|^{1+\alpha}}\right) + \sum_{k=2}^{n} \mathbb{E}\left(\frac{4\hat{d}_0 \|\nabla g(\theta_k, \xi_k)\|^2}{S_k |\ln S_k|^{1+\alpha}}\right)$$

$$\leq \frac{1}{2} \sum_{k=2}^{n} \mathbb{E}\left(\mathbf{1}_{A_k} \frac{\|\nabla g(\theta_k)\|^2}{\sqrt{S_{k-1}}g(\theta_k)|\ln(g(\theta_k)|^{1+\alpha})}\right) + 4\hat{d}_0 \int_{S_2}^{+\infty} \frac{1}{x|\ln x|^{1+\alpha}} dx + \frac{T_1 \|\nabla g(\theta_1, \xi_1)\|^2}{S_1}.$$

$$(37)$$

Substitute *Equation* 34, *Equation* 35 and *Equation* 37 into *Equation* 36, getting

$$\sum_{n=2}^{+\infty} \mathbb{E}\left(\mathbf{1}_{A_n} \frac{\|\nabla g(\theta_n)\|^2}{\sqrt{S_{n-1}}g(\theta_n)|\ln(g(\theta_n))|^{1+\alpha}}\right) < +\infty. \tag{38}$$

For the event $A_n^- := \{\theta_n \notin S^{(\hat{\delta})}\}$ $(\hat{\delta} = (\ln 2)/N)$. Combining *Equation* 30, Through Lemma A.4, we have

$$\sum_{k=2}^{n} \mathbb{E}\left(\mathbf{1}_{A_k^-} \frac{\|\nabla g(\theta_k)\|^2}{\sqrt{S_{k-1}}g(\theta_k)|\ln(g(\theta_k))|^{1+\alpha}}\right) < \tilde{c}_0 \sum_{k=2}^{n} \mathbb{E}\left(\frac{\|\nabla g(\theta_k)\|^2}{\sqrt{S_{k-1}}}\right) < +\infty, \tag{39}$$

where $\tilde{c}_0$ is a constant which can not effect the result. We calculate *Equation* 38+*Equation* 39, getting

$$\sum_{k=2}^{n} \mathbb{E}\left(\frac{\|\nabla g(\theta_k)\|^2}{\sqrt{S_{k-1}}g(\theta_k)|\ln(g(\theta_k))|^{1+\alpha}}\right) < +\infty. \tag{40}$$

$$\square$$

### B.7 PROOF OF LEMMA A.5

*Proof.* We know

$$g(\theta) = -\frac{1}{N} \sum_{i=1}^{N} \left(y_i \ln\left(\frac{1}{1 + e^{-\mathrm{sgn}(y_i - 0.5)\theta^\top x_i}}\right) + (1 - y_i) \ln\left(1 - \frac{1}{1 + e^{-\mathrm{sgn}(y_i - 0.5)\theta^\top x_i}}\right)\right).$$

We defined

$$f(\theta, x_i) := \frac{1}{1 + e^{-\mathrm{sgn}(y_i - 0.5)\theta^\top x_i}},$$

and

$$f_{x_i}(\theta, x_i) := \begin{cases} f(\theta, x_i), & \text{if } y_i = 0, \\ 1 - f(\theta, x_i), & \text{if } y_i = 1. \end{cases} \tag{41}$$

We can calculate the gradient

$$\nabla g(\theta) = -\frac{1}{N} \sum_{i=1}^{N} f_{x_i}(\theta, x_i) x_i.$$

Then we get

$$
\begin{aligned}
-\mathbb{E}\left(\frac{\nabla f(\theta_n)^\top \nabla g(\theta_n, \xi_n)}{\sqrt{S_{n-1}}}\right) &= \mathbb{E}\left(\left(\frac{\hat{\theta}^*\|\theta_n\| - \frac{\theta_n \theta_n^\top \hat{\theta}^*}{\|\theta_n\|}}{(\|\theta_n\|+1)^2}\right)^\top \frac{\nabla g(\theta_n)}{\sqrt{S_{n-1}}}\right) + \mathbb{E}\left(\left(\frac{\hat{\theta}^*}{(\|\theta_n\|+1)^2}\right)^\top \frac{\nabla g(\theta_n)}{\sqrt{S_{n-1}}}\right) \\
&= \mathbb{E}\left(\left(\frac{\hat{\theta}^*\|\theta_n\| - \frac{\theta_n \theta_n^\top \hat{\theta}^*}{\|\theta_n\|}}{(\|\theta_n\|+1)^2}\right)^\top \frac{\nabla g(\theta_n)}{\sqrt{S_{n-1}}}\right) - \mathbb{E}\left(\frac{1}{N\sqrt{S_{n-1}}}\left(\frac{\hat{\theta}^*}{(\|\theta_n\|+1)^2}\right)^\top \sum_{i=1}^N \mathrm{sgn}(y_i - 0.5) f_{x_i}(\theta_n, x_i) x_i\right) \\
&\leq \mathbb{E}\left(\left(\frac{\hat{\theta}^*\|\theta_n\| - \frac{\theta_n \theta_n^\top \hat{\theta}^*}{\|\theta_n\|}}{(\|\theta_n\|+1)^2}\right)^\top \frac{\nabla g(\theta_n)}{\sqrt{S_{n-1}}}\right) \\
&= -\mathbb{E}\left(\frac{1}{N\sqrt{S_{n-1}}}\left(\frac{\hat{\theta}^*\|\theta_n\| - \frac{\theta_n \theta_n^\top \hat{\theta}^*}{\|\theta_n\|}}{(\|\theta_n\|+1)^2}\right)^\top \sum_{i=1}^N \mathrm{sgn}(y_i - 0.5) f_{x_i}(\theta_n, x_i) x_i\right) \\
&= -\mathbb{E}\left(\frac{1}{N\sqrt{S_{n-1}}}\sum_{i=1}^N \mathrm{sgn}(y_i - 0.5) f_{x_i}(\theta_n, x_i)\left(\frac{\hat{\theta}^*\|\theta_n\| - \frac{\theta_n \theta_n^\top \hat{\theta}^*}{\|\theta_n\|}}{(\|\theta_n\|+1)^2}\right)^\top x_i\right) \\
&= \mathbb{E}\left(\frac{1}{N\sqrt{S_{n-1}}}\sum_{i=1}^N \mathrm{sgn}(y_i - 0.5) f_{x_i}(\theta_n, x_i)\frac{\frac{\theta_n^\top \hat{\theta}^* \theta_n^\top x_i}{\|\theta_n\|} - \hat{\theta}^{*\top} x_i \|\theta_n\|}{(\|\theta_n\|+1)^2}\right) \\
&= \mathbb{E}\left(\frac{1}{N\sqrt{S_{n-1}}}\sum_{i=1}^N \mathrm{sgn}(y_i - 0.5) f_{x_i}(\theta_n, x_i)\left(\frac{\theta_n^\top x_i - \hat{\theta}^{*\top} x_i \|\theta_n\|}{(\|\theta_n\|+1)^2} - \frac{\theta_n^\top x_i}{2(\|\theta_n\|+1)^2}\left\|\frac{\theta_n}{\|\theta_n\|} - \hat{\theta}^*\right\|^2\right)\right) \\
&\leq \mathbb{E}\left(\frac{1}{N\sqrt{S_{n-1}}}\sum_{i=1}^N \mathrm{sgn}(y_i - 0.5) f_{x_i}(\theta_n, x_i)\frac{\theta_n^\top x_i - \hat{\theta}^{*\top} x_i \|\theta_n\|}{(\|\theta_n\|+1)^2}\right) + \beta_n,
\end{aligned}
\tag{42}
$$

where

$$
\beta_n := \mathbb{E}\left(\mathbf{1}(\theta_n \text{ is not a margin vector})\frac{1}{N\sqrt{S_{n-1}}}\sum_{i=1}^N \mathrm{sgn}(y_i - 0.5) f_{x_i}(\theta_n, x_i)\frac{|\theta_n^\top x_i|}{2(\|\theta_n\|+1)^2}\left\|\frac{\theta_n}{\|\theta_n\|} - \hat{\theta}^*\right\|^2\right).
$$

through Lemma A.10, we know following inequity

$$
\begin{aligned}
\beta_n &\leq \frac{\max_{1\leq i\leq N}\{\|x_i\|^2\}}{2} \cdot \mathbb{E}\left(\mathbf{1}(\theta_n \text{ is not a margin vector})\frac{1}{\sqrt{S_{n-1}}}\right) \\
&\leq \frac{N^2 \max_{1\leq i\leq N}\{\|x_i\|^2\}}{2k_1^2 \ln^2 2} \cdot \mathbb{E}\left(\mathbf{1}(\theta_n \text{ is not a margin vector})\frac{\|\nabla g(\theta_n)\|^2}{\sqrt{S_{n-1}}}\right) \\
&\leq \frac{N^2 \max_{1\leq i\leq N}\{\|x_i\|^2\}}{2k_1^2 \ln^2 2} \cdot \mathbb{E}\left(\frac{\|\nabla g(\theta_n)\|^2}{\sqrt{S_{n-1}}}\right)
\end{aligned}
$$

For convenient, we assign

$$
H_n := \frac{1}{N\sqrt{S_{n-1}}}\sum_{i=1}^N \psi_i f_{x_i}(\theta_n, x_i)\frac{\theta_n^\top x_i - \theta^{*\top} x_i \|\theta_n\|}{(\|\theta_n\|+1)^2},
\tag{43}
$$

where $\psi_i := \mathrm{sgn}(y_i - 0.5)$. We denote the index of the support vector as $\mathbf{i}_n := \{i | i = \arg\min_{1\leq i\leq N} \psi_i \theta_n^\top x_i / \|\theta_n\|\}$, and $i_n$ is a element of $\mathbf{i}_n$. Then for $H_n$, we have $\exists \hat{k}_0 > 0$, such

that

$$
H_n = \frac{1}{N\sqrt{S_{n-1}}} \sum_{i=1}^{N} \psi_i f_{x_i}(\theta_n, x_i) \frac{{\theta_n}^\top x_i - \hat{\theta}^{*\top} x_i \|\theta_n\|}{(\|\theta_n\|+1)^2}
$$

$$
= \frac{\|\theta_n\|}{N(\|\theta_n+1\|)^2\sqrt{S_{n-1}}} \left( \sum_{i\in\mathbf{i}_n} \psi_i f_{x_i}(\theta_n, x_i)\left(\frac{{\theta_n}^\top x_i}{\|\theta_n\|} - \hat{\theta}^{*\top} x_i\right) + \sum_{i\notin\mathbf{i}_n} \psi_i f_{x_i}(\theta_n, x_i)\left(\frac{{\theta_n}^\top x_i}{\|\theta_n\|} - \hat{\theta}^{*\top} x_i\right) \right)
$$

$$
= \frac{f_{x_i}(\theta_n, x_i)\|\theta_n\|}{N(\|\theta_n+1\|)^2\sqrt{S_{n-1}}} \left( \left( \sum_{i\in\mathbf{i}_n} \psi_i \frac{{\theta_n}^\top x_i}{\|\theta_n\|} - \hat{\theta}^{*\top} x_i \right) + \sum_{i\notin\mathbf{i}_n} \psi_i \frac{f_{x_i}(\theta_n, x_i)}{f_{x_{i_n}}(\theta_n, x_{i_n})}\left(\frac{{\theta_n}^\top x_i}{\|\theta_n\|} - \hat{\theta}^{*\top} x_i\right) \right)
$$

$$
\leq \frac{f_{x_i}(\theta_n, x_i)\|\theta_n\|}{N(\|\theta_n+1\|)^2\sqrt{S_{n-1}}} \left( \left( \sum_{i\in\mathbf{i}_n} \psi_i \frac{{\theta_n}^\top x_i}{\|\theta_n\|} - \hat{\theta}^{*\top} x_i \right) + \hat{k}_0 \sum_{i\notin\mathbf{i}_n} \frac{\psi_i}{e^{(d_{n,i}-d_{n,i_n})(\|\theta_n\|+1)}}\left(\frac{{\theta_n}^\top x_i}{\|\theta_n\|} - \hat{\theta}^{*\top} x_i\right) \right),
$$

where $d_{n,i} := |\theta_n^\top x_i|/\|\theta_n\|$. Through Lemma A.6, we know that there exists $\hat{\delta}_1 > 0$, $\hat{r} > 0$ making when $\|(\theta_n/\|\theta_n\|) - \hat{\theta}^*\| < \mathcal{L} := \min\{\hat{\delta}_1, \tilde{\delta}_0\}$, ($\tilde{\delta}_0$ is defined in Lemma A.9) for any $j \neq i_n$, there is

$$
\left| \frac{\theta^\top x_j}{\|\theta\|} - \hat{\theta}^* x_j \right| < \hat{r}\left| \frac{\theta^\top x_{i_n}}{\|\theta\|} - \hat{\theta}^* x_{i_n} \right|.
$$

We construct two events

$$
\mathcal{C}_n^+ := \left\{ \left\| \frac{\theta_n}{\|\theta_n\|} - \hat{\theta}^* \right\| \geq \mathcal{L} \right\}, \quad \mathcal{C}_n^- := \left\{ \left\| \frac{\theta_n}{\|\theta_n\|} - \hat{\theta}^* \right\| < \mathcal{L} \right\},
$$

and their characteristic function as $\mathbf{1}_{\mathcal{C}_n^+}$. Natruely, we can separate $H_n$ as

$$
H_n = \mathbf{1}_{\mathcal{C}_n^-} H_n + \mathbf{1}_{\mathcal{C}_n^+} H_n. \tag{44}
$$

For $\mathbf{1}_{\mathcal{C}_n^-} H_n$, we have

$$
\begin{aligned}
\mathbf{1}_{\mathcal{C}_n^-} H_n &\leq \mathbf{1}_{\mathcal{C}_n^-} \frac{f_{x_{i_n}}(\theta_n, x_{i_n})\|\theta_n\|}{N(\|\theta_n\|+1)^2\sqrt{S_{n-1}}} \left( \sum_{i\in\mathbf{i}_n} \psi_i\left(\frac{{\theta_n}^\top x_i}{\|\theta_n\|} - \hat{\theta}^{*\top} x_i\right) \right. \\
&\quad + \left. \mathbf{1}_{\mathcal{C}_n^-} \hat{k}_0 \sum_{i\notin\mathbf{i}_n} \frac{\psi_i}{e^{(d_{n,i}-d_{n,i_n})(\|\theta_n\|+1)}}\left(\frac{{\theta_n}^\top x_i}{\|\theta_n\|} - \hat{\theta}^{*\top} x_i\right) \right).
\end{aligned} \tag{45}
$$

In *Equation* 45, we know the first term in the bracket is negative. For the second term, we have

$$
\begin{aligned}
\mathbf{1}_{\mathcal{C}_n^-} \sum_{i=1,i\notin\mathbf{i}_n}^{N} \frac{\psi_i}{e^{(d_{n,i}-d_{n,i_n})(\|\theta_n\|+1)}}\left(\frac{{\theta_n}^\top x_i}{\|\theta_n\|} - \hat{\theta}^{*\top} x_i\right) &= \mathbf{1}_{\mathcal{C}_n^-} \sum_{i=1,i\notin\mathbf{i}_n}^{N} \left( \mathbf{1}\big((d_{n,i} - d_{n,i_n})(\|\theta_n\|+1) < \hat{U}\big) \right. \\
&\quad + \left. \mathbf{1}\big((d_{n,i} - d_{n,i_n})(\|\theta_n\|+1) \geq \hat{U}\big) \right) \frac{\psi_{i_n}}{e^{(d_{n,i}-d_{n,i_n})(\|\theta_n\|+1)}}\left(\frac{{\theta_n}^\top x_i}{\|\theta_n\|} - \hat{\theta}^{*\top} x_i\right).
\end{aligned} \tag{46}
$$

where $\hat{U} > 0$ is an undetermined constant. We know where

$$
(d_{n,i} - d_{n,i_n})(\|\theta_n\|+1) < \hat{U},
$$

which means

$$
\begin{aligned}
\psi_i\left(\frac{{\theta_n}^\top x_i}{\|\theta_n\|} - \hat{\theta}^{*\top} x_i\right) &\leq \mathbf{1}\left(\frac{{\theta_n}^\top x_i}{\|\theta_n\|} - \hat{\theta}^{*\top} x_i \geq 0\right) \cdot \left(\hat{\theta}^{*\top} x_i - \frac{{\theta_n}^\top x_i}{\|\theta_n\|}\right) \\
&\leq \mathbf{1}\left(\frac{{\theta_n}^\top x_i}{\|\theta_n\|} - \hat{\theta}^{*\top} x_i \geq 0\right) \cdot \left(\frac{\theta_n^\top x_i}{\|\theta_n\|} - \frac{\theta_n^\top x_{i_n}}{\|\theta_n\|} + \frac{\theta_n^\top x_{i_n}}{\|\theta_n\|} - \hat{\theta}^{*\top} x_{i_n} + \hat{\theta}^{*\top} x_{i_n} - \hat{\theta}^{*\top} x_i\right) \\
&\leq (d_{n,i} - d_{n,i_n}) \leq \frac{\hat{U}}{\|\theta_n\|+1}.
\end{aligned}
$$

On the other hand, Due to the characteristic function $\mathbf{1}_{\mathcal{C}_n^-}$, we can confine *Equation* 46 on the set $\mathcal{C}_n^-$. That means

$$\mathbf{1}_{\mathcal{C}_n^-}\sum_{i\notin\mathbf{i}_n}\mathbf{1}\big((d_{n,i}-d_{n,i_n})(\|\theta_n\|+1)<\hat{U}\big)\frac{\psi_i}{e^{(d_{n,i}-d_{n,i_n})(\|\theta_n\|+1)}}\left(\frac{\theta_n^\top x_i}{\|\theta_n\|}-\hat{\theta}^{*\top}x_i\right)$$
$$\leq\hat{c}N\frac{\hat{U}}{\|\theta_n\|+1}, \tag{47}$$

and

$$\sum_{i\notin\mathbf{i}_n}\mathbf{1}\big((d_{n,i}-d_{n,i_n})(\|\theta_n\|+1)\geq\hat{U}\big)\frac{\psi_i}{e^{(d_{n,i}-d_{n,i_n})(\|\theta_n\|+1)}}\left(\frac{\theta_n^\top x_i}{\|\theta_n\|}-\hat{\theta}^{*\top}x_i\right)$$
$$\leq\frac{N\hat{r}}{e^{\hat{U}}}\left|\frac{\theta_n^\top x_{i_n}}{\|\theta_n\|}-\hat{\theta}^{*\top}x_{i_n}\right|. \tag{48}$$

We substitute *Equation* 48 and *Equation* 47 into *Equation* 46, getting

$$\sum_{i=1,i\notin\mathbf{i}_n}^N\frac{\psi_i}{e^{(d_{n,i}-d_{n,i_n})(\|\theta_n\|+1)}}\left(\frac{\theta_n^\top x_i}{\|\theta_n\|}-\hat{\theta}^{*\top}x_i\right)\leq\hat{k}_0\hat{c}N\frac{\hat{U}}{\|\theta_n\|+1}$$
$$+\hat{k}_0\frac{N\hat{r}}{e^{\hat{U}}}\left|\frac{\theta_n^\top x_{i_n}}{\|\theta_n\|}-\hat{\theta}^{*\top}x_{i_n}\right|. \tag{49}$$

We substitute *Equation* 49 into *Equation* 45, acquiring

$$\mathbf{1}_{\mathcal{C}_n^-}H_n\leq\mathbf{1}_{\mathcal{C}_n^-}\frac{f_{x_{i_n}}(\theta_n,x_{i_n})\|\theta_n\|}{N(\|\theta_n\|+1)^2\sqrt{S_{n-1}}}\left(\psi_{i_n}\left(\frac{\theta_n^\top x_{i_n}}{\|\theta_n\|}-\hat{\theta}^{*\top}x_{i_n}\right)+\hat{k}_0\hat{c}N\frac{\hat{U}}{\|\theta_n\|+1}\right.$$
$$\left.+\hat{k}_0\frac{N\hat{r}}{e^{\hat{U}}}\left|\frac{\theta_n^\top x_{i_n}}{\|\theta_n\|}-\hat{\theta}^{*\top}x_{i_n}\right|\right).$$

We take the undetermined constant $\hat{U}=\ln\big(2\hat{k}_0N\hat{r}\big)$, getting $\exists\,\tilde{M}>0$, such that

$$\mathbf{1}_{\mathcal{C}_n^-}H_n\leq\mathbf{1}_{\mathcal{C}_n^-}\frac{f_{x_{i_n}}(\theta_n,x_{i_n})\|\theta_n\|}{N(\|\theta_n\|+1)^2\sqrt{S_{n-1}}}\left(\frac{1}{2}\psi_{i_n}\left(\frac{\theta_n^\top x_{i_n}}{\|\theta_n\|}-\hat{\theta}^{*\top}x_{i_n}\right)+\frac{\tilde{M}}{\|\theta_n\|+1}\right). \tag{50}$$

For $\mathbf{1}_{\mathcal{C}_n^+}H_n$ in *Equation* 44, we can use the similar techniques (from *Equation* 45 to *Equation* 50) to acquire $\exists\,\tilde{M}_1>0$, shuch that

$$\mathbf{1}_{\mathcal{C}_n^+}H_n\leq\mathbf{1}_{\mathcal{C}_n^+}\frac{f_{x_{i_n}}(\theta_n,x_{i_n})\|\theta_n\|}{N(\|\theta_n\|+1)^2\sqrt{S_{n-1}}}\left(\frac{1}{2}\psi_{i_n}\left(\frac{\theta_n^\top x_{i_n}}{\|\theta_n\|}-\hat{\theta}^{*\top}x_{i_n}\right)+\frac{\tilde{M}_1}{e^{s'\|\theta_n\|}}\right). \tag{51}$$

Then we calculate *Equation* 51+*Equation* 50, getting $\exists\,r_0>0,\ \tilde{M}_0>0$, such that

$$H_n\leq\frac{f_{x_{i_n}}(\theta_n,x_{i_n})\|\theta_n\|}{N(\|\theta_n\|+1)^2\sqrt{S_{n-1}}}\left(\frac{1}{2}\psi_{i_n}\left(\frac{\theta_n^\top x_{i_n}}{\|\theta_n\|}-\hat{\theta}^{*\top}x_{i_n}\right)+\frac{r_0}{\|\theta_n\|+1}+\frac{\tilde{M}_0}{e^{s'\|\theta_n\|}}\right). \tag{52}$$

With this, we complete the proof. $\qquad\square$

### B.8 PROOF OF THEOREM 4.1

*Proof.* For the sequence $\{S_n\}$, we separate the proof into two situation. The first situation is $S_n<+\infty$. In this situation, we use Lemma A.4 and Lemma A.2, getting

$$\frac{\|\nabla g(\theta_n)\|^2}{\sqrt{S_{n-1}}}\to 0.$$

Combine $\lim_{n\to+\infty}S_n<+\infty$, getting

$$\|\nabla g(\theta_n)\|\to 0. \tag{53}$$

Then we consider the second situation $S_n \to +\infty$. Through *Equation* 13 and *Equation* 19, we can get $g(\theta_{n+1}) - g(\theta_n)$ as

$$g(\theta_{n+1}) - g(\theta_n) \leq \hat{\alpha}_0 \frac{1}{\sqrt{S_{n-1}}} + \hat{T}_n, \tag{54}$$

where $\hat{\alpha}_0 > 0$ is a constant and $\hat{T}_n$ is a sequence which satisfies $\sum_{n=1}^{+\infty} \hat{T}_n < +\infty$ *a.s.*. Then we can get

$$
\begin{aligned}
\sum_{n=2}^{+\infty} \frac{1}{\sqrt{S_{n-1}}} &\geq \frac{1}{a} \sum_{n=1}^{+\infty} \frac{\mathbb{E}\left(\|\nabla g(\theta_n, \xi_n)\|^2 \big| \mathcal{F}_n\right) - M_0 \|\nabla g(\theta_n)\|^2}{\sqrt{S_{n-1}}} \\
&\geq \frac{1}{a} \sum_{n=2}^{+\infty} \frac{\mathbb{E}\left(\|\nabla g(\theta_n, \xi_n)\|^2 \big| \mathcal{F}_n\right)}{\sqrt{S_{n-1}}} - \zeta_0 \\
&= \frac{1}{a} \sum_{n=2}^{+\infty} \frac{\|\nabla g(\theta_n, \xi_n)\|^2}{\sqrt{S_{n-1}}} + \frac{1}{a} \sum_{n=2}^{+\infty} \frac{\mathbb{E}\left(\|\nabla g(\theta_n, \xi_n)\|^2 \big| \mathcal{F}_n\right) - \|\nabla g(\theta_n, \xi_n)\|^2}{\sqrt{S_{n-1}}} - \zeta_0,
\end{aligned}
\tag{55}
$$

where $\zeta_0 := \sum_{n=2}^{+\infty} M_0 \|\nabla g(\theta_n)\|^2 / a\sqrt{S_{n-1}} < +\infty$ *a.s.*. Next we aim to prove $\sum_{n=2}^{+\infty} 1/\sqrt{S_{n-1}} = +\infty$ *a.s.* by contradiction. We assume $\sum_{n=2}^{+\infty} 1/\sqrt{S_{n-1}} < +\infty$ *a.s.*. Then through Lemma A.2, we get that

$$\frac{1}{a} \sum_{n=2}^{+\infty} \frac{\mathbb{E}\left(\|\nabla g(\theta_n, \xi_n)\|^2 \big| \mathcal{F}_n\right) - \|\nabla g(\theta_n, \xi_n)\|^2}{\sqrt{S_{n-1}}}$$

is convergence *a.s.*. Substitute it into *Equation* 55, acquiring

$$
\begin{aligned}
\frac{1}{a} &\sum_{n=2}^{+\infty} \frac{\|\nabla g(\theta_n, \xi_n)\|^2}{\sqrt{S_{n-1}}} \\
&\leq \sum_{n=2}^{+\infty} \frac{1}{\sqrt{S_{n-1}}} - \frac{1}{a} \sum_{n=2}^{+\infty} \frac{\mathbb{E}\left(\|\nabla g(\theta_n, \xi_n)\|^2 \big| \mathcal{F}_n\right) - \|\nabla g(\theta_n, \xi_n)\|^2}{\sqrt{S_{n-1}}} + \zeta_0 < +\infty.
\end{aligned}
\tag{56}
$$

Howeve, we know

$$\frac{1}{a} \sum_{n=2}^{+\infty} \frac{\|\nabla g(\theta_n, \xi_n)\|^2}{\sqrt{S_{n-1}}} > \frac{1}{a} \int_{S_1}^{+\infty} \frac{1}{\sqrt{x}} dx = +\infty.$$

It contradicts with *Equation* 56. That means

$$\sum_{n=2}^{+\infty} \frac{1}{\sqrt{S_{n-1}}} = +\infty.$$

Combining it with

$$\sum_{n=2}^{+\infty} \frac{\|\nabla g(\theta_n)\|^2}{\sqrt{S_{n-1}}} < +\infty \quad a.s., \tag{57}$$

we acquire that there is a subsequence $\{\|\nabla g(\theta_{k_n})\|^2\}$ of $\{\|\nabla g(\theta_n)\|^2\}$ which satisfies that

$$\lim_{n \to +\infty} \left\|\nabla g(\theta_{k_n})\right\|^2 = 0. \tag{58}$$

Next we aim to prove $\lim_{n \to +\infty} \|\nabla g(\theta_n)\|^2 = 0$. It is equivalent to prove that $\{\|\nabla g(\theta_n)\|^2\}$ has no positive accumulation points, that is to say, $\forall e_0 > 0$, there are only finite values of $\{\|\nabla g(\theta_n)\|\}$ larger than $e_0$. And obviously, we just need to prove $\forall 0 < e_0 < r$, there are only finite values of $\{\|\nabla g(\theta_n)\|\}$ larger than $r$. We prove this by contradiction. We suppose $\exists\, 0 < e < a$, making the set $S = \{\|\nabla g(\theta_n)\|^2 > a\}$ be an infinite set. We assign the *Lipschitz coefficient* of $\nabla g(\theta)$ $(\theta \in \mathbb{R}^d)$ as $c$. Then we assign $b = e/8c$ and define $o = min\{b, e/4\}$. Due to *Equation* 58, we get there exists a subsequence $\{\theta_{p_n}\}$ of $\{\theta_n\}$ which satisfies $\|\nabla g(\theta_{p_n})\| < o$. We rank $S$ as a subsequence $\{\|\nabla g(\theta_{m_n})\|^2\}$ of $\{\|\nabla g(\theta_n)\|^2\}$. Then there is an infinite subsequence

$\{\|\nabla g(\theta_{m_{i_n}})\|^2\}$ of $\{\|\nabla g(\theta_{m_n})\|^2\}$ such that $\forall n \in \mathbb{N}_+, \exists l, \ n_{p_n} \in (m_{i_l}, m_{i_{l+1}})$. For convenient, we abbreviate $\{m_{i_n}\}$ as $\{i_n\}$. And we construct another infinite sequence $\{q_n\}$ as follows

$$q_1 = \max\left\{n : p_1 < n < \min\{m_{i_l : m_{i_l} > p_1}\}, \|\nabla g(\theta_n)\| \le o\right\},$$

$$q_2 = \min\left\{n : n > q_1, \|\nabla g(\theta_n)\| > e\right\},$$

$$q_{2n-1} = \max\left\{n : \min\{m_{i_l} : m_{i_l} > q_{2n-3}\} < n < \min\{m_l : m_l > \min\{m_{i_l} : m_{i_l} > q_{2n-3}\},\right.$$
$$\left.\|\nabla g(\theta_n)\| \le o\right\},$$

$$q_{2n} = \min\left\{n : n > q_{2n-1}, \|\nabla g(\theta_n)\| > e\right\}.$$

Now we prove that $\exists N_0$, when $q_{2n} > N_0$, it has $e < \|\nabla g(\theta_{q_{2n}})\| < r$. The left side is obvious (the definition of $q_{2n}$). And for the right side, we know $\|\nabla g(\theta_{q_{2n}-1})\| \le e$. It follows from *Equation* 1 that

$$\|\theta_{n+1} - \theta_n\|^2 = \frac{\alpha_0^2}{S_n}\|\nabla g(\theta_n, \xi_n)\|^2$$

$$\le \frac{\alpha_0^2}{S_{n-1}}\left(\|\nabla g(\theta_n, \xi_n)\|^2 - \mathbb{E}\left(\|\nabla g(\theta_n, \xi_n)\|^2\big|\mathcal{F}_n\right)\right)$$

$$+ \frac{\alpha_0^2}{S_{n-1}}\left(M_0\|\nabla g(\theta_n)\|^2 + a\right).$$

Through previous consequences we can easily find that

$$\sum_{n=2}^{+\infty}\left(\frac{\alpha_0^2}{S_{n-1}}\left(\|\nabla g(\theta_n, \xi_n)\|^2 - \mathbb{E}\left(\|\nabla g(\theta_n, \xi_n)\|^2\big|\mathcal{F}_n\right)\right) + \frac{\alpha_0^2 M_0\|\nabla g(\theta_n)\|^2}{S_{n-1}}\right)$$
$$< +\infty \quad a.s..$$

Note that $\alpha_0^2 a / S_{n-1} \to 0, \quad a.s..$ We conclude

$$\|\theta_{n+1} - \theta_n\| \to 0 \quad a.s.. \tag{59}$$

Then we get $\left|\|\nabla g(\theta_{n+1})\|^2 - \|\nabla g(\theta_n)\|^2\right| \le \left|\|\nabla g(\theta_{n+1})\| - \|\nabla g(\theta_n)\|\right|^2 \le \|\nabla g(\theta_{n+1}) - \nabla g(\theta_n)\|^2 \le c\|\theta_{n+1} - \theta_n\| \to 0 \ a.s..$ Then through Lemma A.7, we get

$$\|\nabla g(\theta_n)\|^2 \le 2cg(\theta_n) \ (n \in [q_{2n-1}, q_{2n}]).$$

Then we get

$$e - o < \|\nabla g(\theta_{q_{2n}})\|^2 - \|\nabla g(\theta_{q_{2n-1}})\|^2 < 2cg(\theta_{q_{2n}}) - \|\nabla g(\theta_{q_{2n-1}})\|^2$$

$$= \left(2c\sum_{i=0}^{q_{2n}-q_{2n-1}-1} g(\theta_{q_{2n-1}+i+1}) - g(\theta_{q_{2n-1}+i})\right) + 2cg(\theta_{q_{2n-1}}) - \|\nabla g(\theta_{q_{2n-1}})\|^2.$$

From *Equation* 54, we obtain

$$g(\theta_{q_{2n-1}+i+1}) - g(\theta_{q_{2n-1}+i}) \le \hat{\alpha}_0 \frac{1}{\sqrt{S_{q_{2n-1}+i}}} + \hat{T}_n.$$

So there is

$$e - o < \sum_{i=0}^{q_{2n}-q_{2n-1}-1} \frac{\hat{\alpha}_0}{\sqrt{S_{q_{2n-1}+i}}} + \sum_{i=0}^{q_{2n}-q_{2n-1}-1} \hat{T}_{q_{2n-1}+i}$$
$$+ 2cg(\theta_{q_{2n-1}}) - \|\nabla g(\theta_{q_{2n-1}})\|^2. \tag{60}$$

Due to $\|\nabla g(\theta_{q_{2n-1}})\|^2 < o < b$, so we get that $g(\theta_{q_{2n-1}}) < e/8c$. Substitute it into *Equation* 60. We get

$$\sum_{i=0}^{q_{2n}-q_{2n-1}-1} \frac{1}{\sqrt{S_{q_{2n-1}+i}}} > \hat{\alpha}_0 - \sum_{i=0}^{q_{2n}-q_{2n-1}-1} \hat{T}_{q_{2n-1}+i}. \tag{61}$$

Due to $\sum_{n=1}^{+\infty} \hat{T}_n$ is convergence almost surely. So we get that $\sum_{i=0}^{q_{2n}-q_{2n-1}-1} \hat{T}_{q_{2n-1}+i} \to 0$ $a.s.$ by *Cauchy's test for convergence*. Combining $1/\sqrt{S_{q_{2n-1}+i}} \to 0$ $a.s.$, we get

$$\sum_{i=1}^{q_{2n}-q_{2n-1}-1} \frac{1}{\sqrt{S_{q_{2n-1}+i}}} > \hat{\alpha}_0 - \frac{1}{\sqrt{S_{q_{2n-1}}}} - \sum_{i=0}^{q_{2n}-q_{2n-1}-1} \hat{T}_{q_{2n-1}+i} \to \frac{\hat{\alpha}_0}{2} \quad a.s., \tag{62}$$

so there is

$$\sum_{n=1}^{+\infty} \left( \sum_{i=1}^{q_{2n}-q_{2n-1}-1} \frac{1}{\sqrt{S_{q_{2n-1}+i}}} \right) = +\infty \quad a.s.. \tag{63}$$

But on the other hand, we know $\|\nabla g(\theta_{q_{2n-1}+i})\| > o$ $(i > 0)$. Together with *Equation 57*, we get

$$\sum_{n=1}^{+\infty} \left( \sum_{i=1}^{q_{2n}-q_{2n-1}-1} \frac{1}{\sqrt{S_{q_{2n-1}+i}}} \right) < \frac{1}{o} \sum_{n=1}^{+\infty} \left( \sum_{i=1}^{q_{2n}-q_{2n-1}-1} \frac{\left\|\nabla g(\theta_{q_{2n-1}+i})\right\|^2}{\sqrt{S_{q_{2n-1}+i}}} \right)$$

$$< \frac{1}{o} \sum_{n=3}^{n} \frac{\left\|\nabla g(\theta_n)\right\|^2}{\sqrt{S_{n-1}}} < +\infty \quad a.s.. \tag{64}$$

It contradicts with *Equation 63*, so we get that $\|\nabla g(\theta_n)\| \to 0$ $a.s..$ Combining *Equation 53*, we get $\|\nabla g(\theta_n)\| \to 0$ no matter $S_n < +\infty$ $a.s.$ or $S_n = +\infty$. Through Lemma A.10 and Lemma A.8, we get $g(\theta_n) \to 0$ $a.s..$ In the case of linear separable data set, $g(\theta n) \to 0$ $a.s.$ implies $\|\theta_n\| \to +\infty$ $a.s..$ $\qquad\square$

### B.9 PROOF OF THEOREM 4.2

*Proof.* We assign $\hat{\theta}^* := \theta^*/\|\theta^*\|$. Then, we assign

$$f(\theta) := 1 - \frac{\theta^\top \hat{\theta}^*}{\|\theta\| + 1}. \tag{65}$$

Then we use the *taylor expansion* on $f(\theta_{n+1}) - f(\theta_n)$, getting

$$f(\theta_{n+1}) - f(\theta_n) \le \nabla f(\theta_n)^\top (\theta_{n+1} - \theta_n) + T_n \|\theta_{n+1} - \theta_n\|^2$$

$$= -\frac{\alpha_0 \nabla f(\theta_n)^\top \nabla g(\theta_n, \xi_n)}{\sqrt{S_n}} + \frac{T_n \alpha_0^2 \|\nabla g(\theta_n, \xi_n)\|^2}{S_n}$$

$$\le -\frac{\alpha_0 \nabla f(\theta_n)^\top \nabla g(\theta_n, \xi_n)}{\sqrt{S_{n-1}}} + \frac{\alpha_0 \hat{\theta}^{*\top} \nabla g(\theta_n, \xi_n)}{(\|\theta_n\| + 1)^2 \sqrt{S_{n-1}}} \tag{66}$$

$$+ \left( \frac{\hat{\theta}^* \|\theta_n\| - \frac{\theta_n \theta^\top \hat{\theta}^*}{\|\theta_n\|}}{(\|\theta_n\| + 1)^2} \right)^\top \alpha_0 \nabla g(\theta_n, \xi_n) \left( \frac{1}{\sqrt{S_{n-1}}} - \frac{1}{\sqrt{S_n}} \right) + \frac{T_n \alpha_0^2 \|\nabla g(\theta_n, \xi_n)\|^2}{S_n},$$

where

$$T_n := \hat{c}_0 \left( \frac{1}{(\|\theta_n\| + 1)^2} + \frac{1}{(\|\theta_{n+1}\| + 1)^2} \right),$$

where $\hat{c}_0$ is a constant which can not effect the result. For convenience, we assign

$$G_n := \left| \left( \frac{\hat{\theta}^* \|\theta_n\| - \frac{\theta_n \theta^\top \hat{\theta}^*}{\|\theta_n\|}}{(\|\theta_n\| + 1)^2} \right)^\top \alpha_0 \nabla g(\theta_n, \xi_n) \left( \frac{1}{\sqrt{S_{n-1}}} - \frac{1}{\sqrt{S_n}} \right) \right| + \frac{T_n \alpha_0^2 \|\nabla g(\theta_n, \xi_n)\|^2}{S_n}$$

$$+ \frac{\alpha_0 \hat{\theta}^{*\top} \nabla g(\theta_n, \xi_n)}{(\|\theta_n\| + 1)^2 \sqrt{S_{n-1}}} + \frac{N^2 \max_{1 \le i \le N}\{\|x_i\|^2\}}{2k_1^2 \ln^2 2} \cdot \frac{\|\nabla g(\theta_n)\|^2}{\sqrt{S_{n-1}}}.$$

Then we make the mathematical expectation of *Equation 66*, getting

$$\mathbb{E}\left( f(\theta_{n+1}) \right) - \mathbb{E}\left( f(\theta_n) \right) \le \alpha_0 \mathbb{E}\left( H_n \right) + \mathbb{E}\left( G_n \right), \tag{67}$$

where $H_n$ is defined in *Equation* 43. Then we make a sum of *Equation* 67, acquiring

$$\alpha_0 \sum_{k=2}^{n} \mathbb{E}\left(-H_k\right) \leq \mathbb{E}\left(f(\theta_2)\right) + \sum_{k=2}^{n} \mathbb{E}\left(G_k\right).$$

obviously,

$$
\begin{aligned}
\sum_{k=2}^{+\infty} \left\| \mathbb{E}\left(G_k\right) \right\| &\leq \sum_{k=2}^{+\infty} \mathbb{E}\left\|G_k\right\| \\
&\leq \sum_{k=2}^{+\infty} \mathbb{E}\left\| \left(\frac{\hat{\theta}^*\|\theta_n\| - \frac{\theta_n \theta^\top \hat{\theta}^*}{\|\theta_n\|}}{(\|\theta_n\| + 1)^2}\right)^\top \alpha_0 \nabla g(\theta_n, \xi_n)\left(\frac{1}{\sqrt{S_{n-1}}} - \frac{1}{\sqrt{S_n}}\right) \right\| + \sum_{n=2}^{+\infty} \mathbb{E}\left(\frac{T_n \alpha_0^2 \|\nabla g(\theta_n, \xi_n)\|^2}{S_n}\right) \\
&+ \sum_{n=2}^{+\infty} \mathbb{E}\left(\frac{\alpha_0 \hat{\theta}^{*\top} \nabla g(\theta_n, \xi_n)}{(\|\theta_n\| + 1)^2 \sqrt{S_{n-1}}}\right) + \frac{N \max_{1 \leq i \leq N}\{\|x_i\|^2\}}{4c \ln 2} \cdot \sum_{n=2}^{+\infty} \mathbb{E}\left(\frac{\|\nabla g(\theta_n)\|^2}{\sqrt{S_{n-1}}}\right),
\end{aligned}
\tag{68}
$$

and for the first term on the right side of the above inequality, we have $\exists\, \hat{T} > 0$, such that

$$
\begin{aligned}
\sum_{k=2}^{+\infty} \mathbb{E}&\left\| \left(\frac{\hat{\theta}^*\|\theta_n\| - \frac{\theta_n \theta^\top \hat{\theta}^*}{\|\theta_n\|}}{(\|\theta_n\| + 1)^2}\right)^\top \alpha_0 \nabla g(\theta_n, \xi_n)\left(\frac{1}{\sqrt{S_{n-1}}} - \frac{1}{\sqrt{S_n}}\right) \right\| \\
&\leq \hat{T} \sum_{k=2}^{+\infty} \mathbb{E}\left(\|\nabla g(\theta_n, \xi_n)\|\left(\frac{1}{\sqrt{S_{n-1}}} - \frac{1}{\sqrt{S_n}}\right)\right).
\end{aligned}
$$

Through Assumption 3.1 and Lemma A.8, we can get that $\exists\, \hat{T}_0 > 0,\ \hat{T}_1 > 0$, such that

$$
\begin{aligned}
\hat{T} \sum_{n=2}^{+\infty} \mathbb{E}&\left(\|\nabla g(\theta_n, \xi_n)\|\left(\frac{1}{\sqrt{S_{n-1}}} - \frac{1}{\sqrt{S_n}}\right)\right) \\
&\leq \hat{T} \sum_{n=2}^{+\infty} \mathbb{E}\left(I\big(\|\nabla g(\theta_n)\| \leq \delta_0\big)\|\nabla g(\theta_n, \xi_n)\|\left(\frac{1}{\sqrt{S_{n-1}}} - \frac{1}{\sqrt{S_n}}\right)\right) \\
&+ \hat{T} \sum_{n=2}^{+\infty} \mathbb{E}\left(I\big(\|\nabla g(\theta_n)\| > \delta_0\big)\|\nabla g(\theta_n, \xi_n)\|\left(\frac{1}{\sqrt{S_{n-1}}} - \frac{1}{\sqrt{S_n}}\right)\right) \\
&\leq \hat{T}_0 \frac{1}{\sqrt{S_1}} + \hat{T}_1 \sum_{n=2}^{+\infty} \mathbb{E}\left(\frac{\|\nabla g(\theta_n)\|^2}{\sqrt{S_{n-1}}}\right) < +\infty.
\end{aligned}
\tag{69}
$$

For the second term in *Equation* 68, Through Assumption 3.1 and Lemma 4.2, we have

$$
\begin{aligned}
\sum_{n=2}^{+\infty} \mathbb{E}&\left(\frac{T_n \alpha_0^2 \|\nabla g(\theta_n, \xi_n)\|^2}{S_n}\right) \\
&\leq \hat{c}_0 \left(\sum_{n=1}^{+\infty} \mathbb{E}\left(\frac{\alpha_0^2 \|\nabla g(\theta_n, \xi_n)\|^2}{(\|\theta_n\| + 1)^2 S_n}\right) + \sum_{n=1}^{+\infty} \mathbb{E}\left(\frac{\alpha_0^2 \|\nabla g(\theta_n, \xi_n)\|^2}{(\|\theta_{n+1}\| + 1)^2 S_n}\right)\right) < +\infty.
\end{aligned}
$$

For the third and forth term of equation 68, we can use Lemma A.4 and Lemma 4.1 to prove their are convergence. That means $\exists\, \hat{K}_1 > 0$, such that

$$\sum_{k=1}^{n} \mathbb{E}\left(-H_k\right) \leq \hat{K}_1 < +\infty.
\tag{70}$$

Through equation 52, we get

$$H_n \leq \frac{f_{x_{i_n}}(\theta_n, x_{i_n})\|\theta_n\|}{N(\|\theta_n\| + 1)^2 \sqrt{S_{n-1}}}\left(\frac{1}{2}\psi_{i_n}\left(\frac{\theta_n^\top x_{i_n}}{\|\theta_n\|} - \hat{\theta}^{*\top} x_{i_n}\right) + \frac{r_0}{\|\theta_n\| + 1} + \frac{\tilde{M}_0}{e^{s'\|\theta_n\|}}\right).
\tag{71}$$

Substitute *Equation* 71 into *Equation* 70, we getting

$$\sum_{n=2}^{+\infty} \mathbb{E}\left(\frac{\psi_{i_n}\|\theta_n\|f_{x_{i_n}}(\theta_n, x_{i_n})}{N(\|\theta_n\| + 1)^2\sqrt{S_{n-1}}}\left(\hat{\theta}^{*\top}x_{i_n} - \frac{\theta_n^{\top}x_{i_n}}{\|\theta_n\|}\right)\right)$$

$$\leq 2\hat{K}_1 + 2\sum_{n=2}^{+\infty} \mathbb{E}\left(\mathbf{1}_{\mathcal{C}_n^-}\frac{\|\theta_n\|f_{x_{i_n}}(\theta_n, x_{i_n})}{N(\|\theta_n\| + 1)^2\sqrt{S_{n-1}}}\cdot\left(\frac{r_0}{\|\theta_n\| + 1}\right)\right) + \sum_{n=2}^{+\infty} \mathbb{E}\left(\frac{\|\theta_n\|f_{x_{i_n}}(\theta_n, x_{i_n})}{N(\|\theta_n\| + 1)^2\sqrt{S_{n-1}}}\cdot\frac{\hat{M}_0}{e^{s'\|\theta_n\|}}\right).$$

Then we calculate the third term of *Equation* 67. We know when $\theta_n \in A_n^-$, there is

$$\frac{\|\theta_n\|}{\cdot(\|\theta_n\| + 1)^3} \leq \tilde{k}_0\frac{1}{\ln^2(g(\theta_n))} = \tilde{k}_0\frac{1}{|\ln(g(\theta_n))|^{1+1}},$$

where $\tilde{k}_0$ is a constant where can not effect the result. Combine Lemma 4.2, We can get

$$\sum_{n=2}^{+\infty} \mathbb{E}\left(\mathbf{1}_{\mathcal{C}_n^-}\frac{f_{x_{i_n}}(\theta_n, x_{i_n})}{N\|\theta_n\|\sqrt{S_{n-1}}}\cdot\left(\frac{r_0}{\|\theta_n\| + 1}\right)\right) \leq \tilde{k}_1\sum_{n=2}^{+\infty} \mathbb{E}\left(\frac{g(\theta_n)}{\sqrt{S_{n-1}}|\ln(g(\theta_n))|^{1+1}}\right) < +\infty.$$

Similarly, through Lemma 4.2, we can get

$$\sum_{n=2}^{+\infty} \mathbb{E}\left(\frac{\|\theta_n\|f_{x_{i_n}}(\theta_n, x_{i_n})}{N(\|\theta_n\| + 1)^2\sqrt{S_{n-1}}}\cdot\frac{\hat{M}_0}{e^{s'\|\theta_n\|}}\right) < +\infty.$$

That means we can get

$$\sum_{n=2}^{+\infty} \mathbb{E}\left(\frac{\psi_{i_n}\|\theta_n\|f_{x_{i_n}}(\theta_n, x_{i_n})}{N(1 + \|\theta_n\|)^2\sqrt{S_{n-1}}}\left(\hat{\theta}^{*\top}x_{i_n} - \frac{\theta_n^{\top}x_{i_n}}{\|\theta_n\|}\right)\right) < +\infty,$$

We simplify the above inequality, getting

$$\sum_{n=2}^{+\infty} \mathbb{E}\left(\frac{\|\theta_n\|f_{x_{i_n}}(\theta_n, x_{i_n})}{N(\|\theta_n\| + 1)^2\sqrt{S_{n-1}}}\left|\hat{\theta}^{*\top}x_{i_n} - \frac{\theta_n^{\top}x_{i_n}}{\|\theta_n\|}\right|\right) < +\infty.$$

Through Lemma A.1, we have

$$\sum_{n=2}^{+\infty}\frac{\|\theta_n\|f_{x_{i_n}}(\theta_n, x_{i_n})}{N(\|\theta_n\| + 1)^2\sqrt{S_{n-1}}}\left|\hat{\theta}^{*\top}x_{i_n} - \frac{\theta_n^{\top}x_{i_n}}{\|\theta_n\|}\right| < +\infty \quad a.s.. \tag{72}$$

We back to *Equation* 66. We make a sum of *Equation* 66, getting

$$f(\theta_{n+1}) = f(\theta_1) + \sum_{k=1}^{n}\left(\frac{\hat{\theta}^*(\|\theta_k\| + 1) - \frac{\theta_k\theta_k^{\top}\hat{\theta}^*}{\|\theta_k\|}}{(\|\theta_k\| + 1)^2}\right)^{\top}\frac{\alpha_0\nabla g(\theta_k, \xi_k)}{\sqrt{S_k}}$$
$$+ \sum_{k=1}^{n}\frac{\hat{c}\alpha_0^2\|\nabla g(\theta_n, \xi_n)\|^2}{S_n}. \tag{73}$$

For the first series sum, we have

$$\sum_{k=2}^{n}\left(\frac{\hat{\theta}^*(\|\theta_k\| + 1) - \frac{\theta_k\theta_k^{\top}\hat{\theta}^*}{\|\theta_k\|}}{(\|\theta_k\| + 1)^2}\right)^{\top}\frac{\alpha_0\nabla g(\theta_k, \xi_k)}{\sqrt{S_k}} = \sum_{k=2}^{n}\left(\frac{\hat{\theta}^*(\|\theta_k\| + 1) - \frac{\theta_k\theta_k^{\top}\hat{\theta}^*}{\|\theta_k\|}}{(\|\theta_k\| + 1)^2}\right)^{\top}\frac{\alpha_0\nabla g(\theta_k)}{\sqrt{S_{k-1}}}$$

$$- \sum_{k=2}^{n}\left(\frac{\hat{\theta}^*(\|\theta_k\| + 1) - \frac{\theta_k\theta_k^{\top}\hat{\theta}^*}{\|\theta_k\|}}{(\|\theta_k\| + 1)^2}\right)^{\top}\left(\frac{\alpha_0\nabla g(\theta_k, \xi_k)}{\sqrt{S_{k-1}}} - \frac{\alpha_0\nabla g(\theta_k, \xi_k)}{\sqrt{S_k}}\right) + \sum_{k=2}^{n}\zeta_k,$$

where $\{\zeta_n\}$ is a martingale difference sequence. Through *Equation* 69, *Equation* 72 and Lemma A.2, we can get

$$\sum_{k=2}^{n}\left(\frac{\hat{\theta}^*(\|\theta_k\| + 1) - \frac{\theta_k\theta_k^{\top}\hat{\theta}^*}{\|\theta_k\|}}{(\|\theta_k\| + 1)^2}\right)^{\top}\frac{\alpha_0\nabla g(\theta_k, \xi_k)}{\sqrt{S_k}}$$

convergence a.s.. Meanwhile, through *Equation* 72, we have

$$\sum_{k=2}^{n} \left| \frac{\hat{c}\alpha_0^2 \|\nabla g(\theta_k, \xi_k)\|^2}{S_k} \right| \leq \sum_{k=2}^{n} \left( \frac{1}{\|\theta_k\|^2} + \frac{1}{\|\theta_{k+1}\|^2} \right) \frac{\hat{c}_0 \alpha_0^2 \|\nabla g(\theta_k, \xi_k)\|^2}{S_k} < +\infty \ \ a.s..$$

That means

$$\sum_{k=1}^{n} \frac{\hat{c}\alpha_0^2 \|\nabla g(\theta_k, \xi_k)\|^2}{S_k}$$

is absolute convergence a.s.. Naturally, it is convergence a.s.. Until now, we already prove two series sums in *Equation* 73 are both convergence a.s.. That means $f(\theta_n)$ is convergence a.s.. We assign

$$c := \lim_{n \to +\infty} f(\theta_n) \ \ a.s.,$$

where $c < +\infty$ is a random variable about the trajectory. Through Theorem 4.1, we know

$$\|\theta_n\| \to +\infty.$$

That means

$$\lim_{n \to +\infty} \frac{\|\theta_n\| + 1}{\|\theta_n\|} = 1 \ \ a.s..$$

$$\lim_{n \to +\infty} 1 - \frac{\theta^\top \hat{\theta}^*}{\|\theta\|} = \lim_{n \to +\infty} f(\theta_n) = c,$$

so we can get that

$$\lim_{n \to +\infty} \left\| \frac{\theta_n}{\|\theta_n\|} - \hat{\theta}^* \right\|^2 = 2c \ \ a.s.$$

Next we aim to prove that $c = 0$ by contradiction. We assume $c > c' > 0$. Then we can conclude that

$$\lim_{n \to +\infty} \left| \hat{\theta}^{*\top} x_{i_n} - \frac{\theta_n^\top x_{i_n}}{\|\theta_n\|} \right| > r(c') > 0 \ \ a.s..$$

We can further conclude that

$$\sum_{n=2}^{+\infty} \frac{\|\theta_n\| f_{x_{i_n}}(\theta_n, x_{i_n})}{N(\|\theta_n\| + 1)^2 \sqrt{S_{n-1}}} \left| \hat{\theta}^{*\top} x_{i_n} - \frac{\theta_n^\top x_{i_n}}{\|\theta_n\|} \right| > r(c') \sum_{n=2}^{+\infty} \frac{f_{x_{i_n}}(\theta_n, x_{i_n})}{N\|\theta_n\| \sqrt{S_{n-1}}} \ \ a.s.. \tag{74}$$

We can get

$$\sum_{n=2}^{+\infty} \frac{f_{x_{i_n}}(\theta_n, x_{i_n})}{N\|\theta_n\| \sqrt{S_{n-1}}} > \hat{r}' \sum_{n=2}^{+\infty} \frac{\|\theta_n\| f_{x_{i_n}}(\theta_n, x_{i_n})}{N(\|\theta_n\| + 1)^2 \sqrt{S_{n-1}}},$$

where $\hat{r}' > 0$ is a constant. Then we can get

$$\sum_{n=2}^{+\infty} \frac{\|\theta_n\| f_{x_{i_n}}(\theta_n, x_{i_n})}{N(\|\theta_n\| + 1)^2 \sqrt{S_{n-1}}} > q_1 \sum_{n=1}^{+\infty} \left( \ln \|\theta_{n+1}\| - \ln \|\theta_n\| \right) - q_2 \sum_{n=1}^{+\infty} \frac{\|\nabla g(\theta_n, \xi_n)\|^2}{\|\theta_n\|^2 S_n} = +\infty \ a.s.,$$

where $q_1 > 0$ and $q_2$ are two constants will can not effect the result. That means

$$\sum_{n=2}^{+\infty} \frac{\|\theta_n\| f_{x_{i_n}}(\theta_n, x_{i_n})}{N(\|\theta_n\| + 1)^2 \sqrt{S_{n-1}}} \left| \hat{\theta}^{*\top} x_{i_n} - \frac{\theta_n^\top x_{i_n}}{\|\theta_n\|} \right| = +\infty,$$

which is contradict with *Equation* 73. That means

$$\lim_{n \to \infty} \left\| \frac{\theta_n}{\|\theta_n\|} - \hat{\theta}^* \right\| = 0 \ \ a.s.,$$

that is

$$\frac{\theta_n}{\|\theta_n\|} \to \frac{\theta^*}{\|\theta^*\|} \ \ a.s..$$

With this, we complete the proof.

$\square$

### B.10 Generalizing to Tight Exponential-tail Loss

We show how our technique remains applicable when dealing with tight exponential-tail loss. This demonstration reinforces the enduring relevance of Lemma A.8 under the exponential-tail loss setting.

Before we give the main steps, we give the definition of 'tight exponential tail loss', by adopting the definition in Gunasekar et al. (2018).

**Definition** Consider a general classification problem, i.e., $\mathcal{L}(w) = \sum_{n=1}^{N} l(y_n w^T x_n)$, where $\{x_n, y_n\}_{n=N}^{2}$ is a dataset and labels $y_n \in \{-1, 1\}$ . Suppose the gradient $l'(u)$ satisfies:

$$
\begin{aligned}
\forall\, u > u_+ :\ & l'(u) \le c(1 + e^{-u_+ + u}) e^{-au}, \\
\forall\, u > u_- :\ & l'(u) \ge c(1 - e^{-u_- - u}) e^{-au},
\end{aligned}
\tag{75}
$$

where $u_+ > 0$, $u_- > 0$, $a > 0$ are three constants, and $\lim_{u \to +\infty} l(u) = \lim_{u \to +\infty} l'(u) = 0$ , $l'(u) < 0$ , then we call $\mathcal{L}$ a tight exponential-tail loss (refer to Assumption 2, Definition 2, and Assumption 3 in Gunasekar et al. (2018) for reference).

By solving the corresponding differential equation, we derive

$$
\forall\, u > u_+ :\ l(u) \le c' e^{-au}, \quad \forall\, u > u_- :\ l(u) \ge c'' e^{-au}.
\tag{76}
$$

By combining the aforementioned inequalities and $\nabla \mathcal{L}(w) = \sum_{n=1}^{N} l'(w(t)^T x_n) y_n x_n$, we can derive a similar result to Lemma A.8 under the 'tight exponential tail loss' setting. We will then clarify it.

The left side of Lemma A.8 is derived as follows. Since $\{x_n, y_n\}$ is a linearly separable dataset, it has a maximum margin vector $\omega^*$. The margin vector $\omega^*$ satisfies the separation of $y_n = 1$, $(\omega^*)^T x_n > 0$ and $y_n = -1$, $(\omega^*)^T x_n < 0$. In addition, it has a lower bound $r := \min_n \{\|(\omega^*)^T x_n\|\}$ . Then, we acquire

$$
\|\nabla \mathcal{L}(\theta)\| = \left\| \sum_{n=1}^{N} l'(y_n \omega(t)^T x_n) y_n x_n \right\| \ge \left| (\omega^*)^T \sum_{n=1}^{N} l'(y_n \omega(t)^T x_n) y_n x_n \right|
$$

$$
= \| \sum_{n=1}^{N} l'(y_n \omega(t)^T x_n) y_n (\omega^*)^T x_n \|.
$$

Due to the conditions $y_n = 1$, $(\omega^*)^T x_n > 0$ and $y_n = -1$, $(\omega^*)^T x_n < 0$, all the signs of $\{y_n (\omega^*)^T x_n\}$ are the same. Then, we get

$$
\|\nabla \mathcal{L}(\theta)\| \ge \left\| \sum_{n=1}^{N} l'(y_n \omega(t)^T x_n) y_n \omega^{*T} x_n \right\| = \sum_{n=1}^{N} \|l'(y_n \omega(t)^T x_n)\| \cdot \|y_n \omega^{*T} x_n\|
$$

$$
\ge r \sum_{n=1}^{N} \|l'(y_n \omega(t)^T x_n)\|.
$$

For the right side of Lemma A.8, we use the triangle inequality, i.e.,

$$
\|\nabla \mathcal{L}(w)\| = \left\| \sum_{n=1}^{N} l'(w(t)^T x_n) y_n x_n \right\| \le \hat{r} \sum_{n=1}^{N} |l'(w(t)^T x_n)|,
$$

where $\hat{r} > 0$ is a given scalar.

According to *Equation* 76 and *Equation* 75, we can observe that Lemma A.8 still holds.

