# OpenReview forum: "The Implicit Bias of Stochastic AdaGrad-Norm on Separable Data"
_ICLR.cc/2024/Conference — Submitted to ICLR 2024_

### Official Review · Reviewer_jNzZ · 2023-10-29

**Soundness:** 3 good
**Presentation:** 3 good
**Contribution:** 3 good
**Rating:** 6
**Confidence:** 2

**Summary:**

In this paper authors provide analysis for the stochastic AdaGrad-Norm method, applied to binary classification problem with linearly separable data. The main motivation of the authors is as follows. Due to the specific nature of stochastic AdaGrad-Norm, the existing techniques, used to analyse other stochastic or deterministic first-order methods, is inapplicable in this case. Firstly, they show that this method converges to max-margin solution almost surely. Secondly, they get the almost surely convergence rate of max-margin vector. To obtain these results authors assume, that the variance of the stochastic gradient is upper-bounded by squared norm of the full gradient, and if loss function is upper-bounded, then its stochastic gradient is also upper-bounded. This is a more general assumption, then existing assumption about regular sampling noise.

**Strengths:**

1. The problem of existence of implicit bias of optimization methods for binary classification problem with separable data is being studied since 2017-2018. In this paper authors extend this theory on stochastic AdaGrad-Norm method.
2. Authors clearly describe, why existing analysis does not work for this method.
3. Authors provide a sketch for long proof of one of the theorems and provide its intuition.

**Weaknesses:**

Lack of experimental results. It worths to show numerically, that indeed AdaGrad-Norm converges to zero error for this problem.

**Questions:**

Questions:
1. Proof of Lemma A.4. Could you please describe in more details, why $\|\nabla^2 g(\theta)\| = \Theta(\|\nabla g(\theta)\|)$ in the neighborhood of the stationary point?

Also I have some minor remarks.
1. Maybe move problem formulation to introduction, because it is rather hard to understand the introduction part, if you are not very familiar with topic.
2. Page 2. You haven't introduced $f$ and $g$.
3. Page 2, row 8. Seems like, you forgot $\mathbb E$ sign in front of $\frac{\alpha_0}{\sqrt S_n} f(\theta_n) \nabla g(\theta_n)$
4. Same place, you forgot transpose sign
5. Page 2, last paragraph before **Related Works**. Change $\zeta$ to $\xi$
6. Page 3, **Contributions**, second part, fourth line. Probably, you forgot "*" over $\theta$-s in the second term under the norm.
7. Page 3, **Contributions**, second part, sixth line. You forgot second "|" in the closing gap of the norm
8. Page 4. Probably, it is better to remove "and Main Results" from the name of the 3rd section.
9. Page 4, Main Results, first paragraph. Since Theorem 4.1 is from another paper, I think, it is better to remove "Our" from first line of the paragraph.
10, Page 5, **Intuition of theorem**. What do you mean by "(requiring additional validation)"?
11. Page 5. Maybe remove index $n$ in $\psi_{n,i}$, because it does not depend on $n$ and only makes equations messier.
11. Page 6, Lemma 4.1, first line. Probably, there should be "there is a vector $x_\theta$".
12. Page 6, Lemma 4.1. What is $U(..., ...)$? It seems like you meant $\delta_0$-neighborhood, but still define it earlier, please.

---

> ### Author Response · Authors · 2023-11-15
> **Author rebuttal**
>
> We thank the reviewer for the feedback. We are glad to have our contributions and novelties acknowledged. The detailed comments greatly help us to revise our manuscript. Amendments have been made according to the comments. We provide a response to the question below.
>
>
> > Question/Comment 1: Proof of Lemma A.4. Could you please describe in more details, why $$\\|\nabla^2 g\\|=\Theta(\\|\nabla g\\|)$$ in the neighborhood of the stationary point?
>
> In this problem, the stationary point of the loss function is at infinity, and it is not all infinite points, but rather infinite points along a certain margin vector direction. Therefore, vectors near the stationary point must be a margin vector of the dataset, resulting in a loss function form similar to $e^{-x}$ (x is sufficiently large). Hence, this conclusion can be drawn.

---

> ### Author Response · Authors · 2023-11-22
> **Thank you**
>
> We thank the reviewer again for your time and effort in reviewing our paper! Your feedback is very helpful to us in improving the manuscript. As the discussion period is coming to a close, we would like to note that we remain open to any further feedback and are committed to making additional improvements.

---

> ### Comment · Reviewer_jNzZ · 2023-11-22
> **Some of mentioned remarks are not fixed**
>
> Thank you for your answers. Sorry, I was rather occupied on the rebuttal process for other papers. Now I've looked through the revised version of your paper, and I see that you haven't fixed the issues, that I mentioned in my review.
>
> > 1. Maybe move problem formulation to introduction, because it is rather hard to understand the introduction part, if you are not very familiar with topic.
>
> I still think, that problem formulation part should be either moved or merged into introduction section before literature review. Since at first you have to introduce the problem you are working with, and only then show the other existing approaches to its solving.
>
> > 3. Page 2, row 8. Seems like, you forgot $\mathbb E$ sign in front of $\frac{\alpha_0}{\sqrt S_n} f(\theta_n) \nabla g(\theta_n)$
>
> Not fixed
>
> > 4. Same place, you forgot transpose sign
>
> I see, that you do not need a transpose sign here. Thus, you need to remove one from the third line in $\frac{\alpha_0}{\sqrt S_n} f(\theta_n)^T \nabla g(\theta_n)$
>
> > 11. Page 5. Maybe remove index $n$ in $\psi_{n,i}$, because it does not depend on $n$ and only makes equations messier.
>
> Not fixed. Or if it is deliberately left as it is, please clarify this moment.
>
> Also, if you fix any of my remarks, pleas highlight them with color: I see that you highlight only some of your fixes.

---

> ### Author Response · Authors · 2023-11-22
> **Follow-up updates**
>
> We thank the reviewer for the follow-up comments. We are currently working on moving the formulation to the introduction. We fully agree that the problem needs to be clearly defined before discussing the related works and the contributions. Especially, the contribution part even uses the term $\theta^*/\\|\theta^*\\|$. We will update the manuscript again in a few hours. We wanted to respond to the rest of the questions first.
>
> > Page 2, row 8. Seems like, you forgot $\mathbb{E}$ sign in front of $\frac{\alpha_{0}^{2}}{S_{n}}\big\|\nabla g(\theta_{n},\xi_{n})\big\|^{2}$
>
> The sentence was equivalently "the expectation of $\frac{\alpha_{0}^{2}}{S_n}\big\\|\nabla g(\theta_n,\xi_n)\big\\|^{2}$" cannot be $\frac{\alpha_{0}^{2}}{S_n}{\mathbb{E}_\xi}_n\Big(\big\\|\nabla g(\theta_n,\xi_n)\big\\|^{2}\Big)$. We did not put the $\mathbb{E}$ sign before the first term because of the phrase "the expectation of ". The sentence was indeed confusing, and now we have revised it by breaking it into two sentences.
>
> > Same place, you forgot transpose sign
>
> Indeed we forgot the transpose sign. We have fixed that (in the same highlighted sentence).
>
> > Page 5. Maybe remove index $n$ in $\psi_{n,i}$, because it does not depend on $n$ and only makes equations messier.
>
> It was deliberately left as it is. We wanted to explain the reason. In fact, this quantity is only independent of $n$ when $\theta_n$ is a margin vector. However, during training, $\theta_n$ is not initially a margin vector. Therefore, considering the overall update process, it is rigorous to include the $n$ index.

---

> ### Author Response · Authors · 2023-11-22
> **Updated formulation**
>
> We wanted to follow up on our revision of the formulation. We now break down the introduction into 5 parts: the beginning part, "Formulation", "Challenges", "Related Works", and "Contributions". The definition of the $\mathcal{L}^{2}$ max-margin solution $\theta^*/\\|\theta^*\\|$ and the loss function $g(\theta)$ are defined in the "Formulation" part of the introduction (highlighted in blue). Up to the point of (1) plus "Formulation", we should have everything needed in the introduction section defined. Because of this change we contracted some definitions in Section 2 so they are not restated (also highlighted in blue).
>
> We also merged two paragraphs in the introduction with the technical contribution part as the "Contributions" paragraph. Now we clearly state the main results before briefly introducing our main techniques in response to the main challenges (highlighted in blue).
>
> We thank the reviewer again for suggesting this edition and we are happy to answer further questions and make further updates :)

---

> ### Author Response · Authors · 2023-11-23
>
> Regarding the issue with $\psi_{n,i}$, it was our oversight. We have already removed $n$ in the revised version. :) :)

---

> > ### Comment · Reviewer_jNzZ · 2023-11-23
> > **Paper exceeds allowed length**
> >
> > Thanks for your comments and changes. But now your paper length is 10 pages, which exceeds the maximal limit of 9 pages for ICLR 2024. As far as I understand, if you don't fix it, it will be rejected automatically.

---

> ### Author Response · Authors · 2023-11-23
> **Paper length update**
>
> We thank the reviewer a lot for pointing the paper length issue out. We have contracted two equations in the proof of Theorem 4.3 to in-line equation formal, and the manuscript should now abide by the length constraint. We are committed to making any further clarifications and updates.

---

> > ### Comment · Reviewer_jNzZ · 2023-12-04
> > **Reduce of the score and confidence**
> >
> > I thank the authors and other reviewers for a hard work. Sorry, I did not have enough time to look through proofs. But, looking at discussion of reviewer Ckt3 with authors, it seems like there are lots of typos or mistakes. Actually, I've point out some of them in the main text, but it seemed to me, that they are minor, and there are not so many of them. Due to these facts, I came to conclusion, that I've overestimated the paper. Thus, I reduce both its rating and my confidence level.

---

### Official Review · Reviewer_VfNS · 2023-10-30

**Soundness:** 2 fair
**Presentation:** 3 good
**Contribution:** 2 fair
**Rating:** 6
**Confidence:** 3

**Summary:**

The paper examines the implicit bias of Stochastic Adagrad within the framework of linear separable data. Specifically, the authors consider linear classification applied to data that can be linearly separated, aiming to determine the hyperplane that correctly classifies the data. Out of all hyperplanes that correctly separate the data, the authors center their attention on the maximum-margin vector, which optimally widens the gap between positive and negative data points. The main contribution of the paper lies in establishing that once Stochastic Adagrad-Norm is coupled with the logistic loss, convergence towards the maximum-margin vector is achieved.

**Strengths:**

Achieving convergence towards the maximum-margin vector is a crucial objective for the sake of generalization, as emphasized in prior research. Additionally, Stochastic Adagrad norm is a widely utilized method in practical applications. From this perspective, it is an interesting question whether Stochastic Adagrad norm exhibits implicit bias in favor of the maximum-margin maximizer. Furthermore, the technical contributions of the paper are nicely presented in the main part of the paper and they seem to admit an interesting technical depth (I have not checked the appendix in detail though).  Finally, the paper effectively extends the recent findings of [1] demonstrating that Adagrad norm converges towards a margin vector that however may not correspond to the maximum-margin solution.


[1]  On the convergence of mSGD and AdaGrad for stochastic optimization, Jin et al., ICML 2022

**Weaknesses:**

As also noted by the authors, prior studies within the same context have already established implicit biases for methods like Gradient Descent and SGD. From this perspective, I find the presented results somewhat unsurprising, leading to doubts about their significance. Additionally, I find the established convergence rate to be a weak point. As far as I comprehend, Theorem 4.3 suggests that Stochastic Adagrad-Norm requires $O(2^{1/\epsilon})$ iterations to converge to an $\epsilon$-optimal solution for the respective logistic regression problem. Another less important concern pertains to the fact that the provided convergence results are based on the best-iterate rather than the last-iterate of Stochastic Adagrad Norm.


Minor: There is room for improvement in terms of clarity in some sections of the paper. For instance, the presentation of the max-margin solutions could benefit from additional details and more formal definitions. Furthermore, it's worth noting that the concept of a "margin vector" is never formally defined in the paper.

**Questions:**

1. As far as I understand, Assumption 3.1 holds for the standard estimator that samples data points uniformly at random. If this is the case I think it would be better to remove Assumption 3.1 and provide a Lemma stating that the assumption is satisfied for the standard estimator.

2. Are you aware of a first-order method that does not exhibit implicit bias to max-margin solutions in such linear classification problems? In the case there is such a method, I think it would be worth mentioning it so as to emphasize that implicit bias is not a general property of first-order methods.

---

> ### Author Response · Authors · 2023-11-15
> **Author rebuttal**
>
> We thank the reviewer for the feedback. We are glad to receive recognition of our contributions and very constructive suggestions from the reviewer. We have revised the manuscript according to the reviewers' suggestions and the revisions are marked on the new pdf. We now provide detailed responses to the questions.
>
> > Question 1: As far as I understand, Assumption 3.1 holds for the standard estimator that samples data points uniformly at random. If this is the case I think it would be better to remove Assumption 3.1 and provide a Lemma stating that the assumption is satisfied for the standard estimator.
>
> We very much appreciate the suggestion. In fact, by removing Assumption 3.1 and putting the standard estimator into the setting, the presentation is clean and becomes easier to follow. Nevertheless, the noise model we work on is broader than the mini-batch setting. An example is reinforcement learning with bounded rewards, where the gradient is then bounded by a constant that is independent of $g$ [1]. This could potentially broaden the applicable cases of our results.
> For a better presentation, we have revised the manuscript, and added in the beginning of Section 3 the main purpose of the noise model as well as the section. We believe that with your suggestion, the revised presentation could bring up more clarity.
>
> > Question 2: Are you aware of a first-order method that does not exhibit implicit bias to max-margin solutions in such linear classification problems? In the case there is such a method, I think it would be worth mentioning it so as to emphasize that implicit bias is not a general property of first-order methods.
>
> Answer: Implicit bias is indeed not a general property of first-order methods. For example, the authors in [2] have observed that by using mirror descent, it is possible to obtain different L-p max margins.
>
>
> [1] Kaiqing Zhang, Alec Koppel, Hao Zhu, and Tamer Basar. Global convergence of policy gradient methods to (almost) locally optimal policies. SIAM Journal on Control and Optimization, 58(6): 3586–3612, 2020.
>
> [2] Sun H, Ahn K, Thrampoulidis C, et al. Mirror descent maximizes generalized margin and can be implemented efficiently. Advances in Neural Information Processing Systems, 2022, 35: 31089-31101.

---

> > ### Comment · Reviewer_VfNS · 2023-11-21
> > **Response to the authors**
> >
> > Thank you for your response. I do not have any further questions.

---

> ### Author Response · Authors · 2023-11-22
> **Thank you**
>
> We thank the reviewer again for your time and effort in reviewing our paper. We very much enjoyed the discussion with the reviewer.

---

### Official Review · Reviewer_Ckt3 · 2023-10-31

**Soundness:** 1 poor
**Presentation:** 1 poor
**Contribution:** 2 fair
**Rating:** 3
**Confidence:** 4

**Summary:**

This paper mainly concerns with the implicit bias property of the stochastic AdaGrad (normed variant) algorithm on linear classification problem. In particular, its main result shows that for linearly separable data, AdaGrad-Norm converges in direction to the $L_2$ maximum-margin solution. Furthermore, a second result gives a convergence rate to the solution induced by the implicit bias of AdaGrad-Norm.

**Strengths:**

This result (if true, see below), would be a nice addition to the study of implicit bias of various optimization algorithms. In the literature, the stochastic setting is considered quite difficult. And the normed version of AdaGrad should be much more challenging than the diagonal variant because the scale matrix $S$ (more commonly referred as $G$ in literature) is no longer separable in coordinates. Thus, it is nice to see the authors attempted this more difficult problem.

**Weaknesses:**

However, I do not believe the results are correct. There are some fatal and elementary errors and the paper shows no sign of proof-reading.

First, the result does not pass an eye test. In [1, 2], the implicit bias of the diagonal variant of AdaGrad depends on various factors such as initialization. Therefore, it is very surprising that the more complex normed version variant would have an implicit bias that do not depend on such conditions.

Then, there are several extremely elementary mistakes:

1. In the equation block in the "Intuition of the theorem" part of Section 4, what is the second gradient taken with respect to? If it is taken wrt $\theta_n$, then the author clearly forgot to apply the chain rule to $\theta_n / \lVert {\theta_n} \rVert$. Also, the value of $i_n$ is in general not unique.

2. In the second equation of the proof of Theorem 4.3, the gradient of $\lVert \theta \rVert^\alpha$ is computed incorrectly, again seemed to be an omission of chain rule. Also, the dimensions in the middle and right expressions do not match.

Next, the formatting of the paper is wrong. It seems that the author put most of the paper into a enumerate block, thus affect the indentation throughout the paper. It is clear to me that no proof-reading was done.

With these observations in mind, I am convinced that this paper is deeply flawed and I cannot trust the author's claims.

Additionally, there are some minor mistakes that further hurt the quality of the paper.

3. Many quantities were not adequately defined before they were being used. For instance, Lipschitzness of $\nabla g$ was not mentioned until Section 4. In the same equation block I pointed to in my first comment, $k_n$ was not defined. In equation (3), $\hat{c}$ was not defined.

4. The first two sentences in Section 4 seems to suggest that Theorem 4.1 is original, but it is in fact not. They need to be rephrased to not confuse the readers.

5. Lemmas A.8 and A.10 where invoked in Section 3 without being stated first. And in Lemma A.8, the term "margin vector" was not defined.

6. The set defined at the top of page 4 is not rigorous. In fact, the global minimum is simply not attainable.

[1] Gunasekar, Suriya, et al. "Characterizing implicit bias in terms of optimization geometry." International Conference on Machine Learning. PMLR, 2018.

[2] Qian, Qian, and Xiaoyuan Qian. "The implicit bias of adagrad on separable data." Advances in Neural Information Processing Systems, 2019.

**Questions:**

None, I think a complete overhaul would be needed.

---

> ### Author Response · Authors · 2023-11-15
> **Author rebuttal (Part 1)**
>
> We thank the reviewer for the detailed feedback. We have revised the manuscript and we now provide the responses to the comments below. Some comments are likely misunderstandings of our statements, which in our opinion could be clarified by our response.
>
> The reviewer provides very negative feedback, primarily because of the belief that our proof is wrong. To our best understanding from the comments, this belief is due to the combination of (1) a gut feeling and eye pass based on existing results on AdaGrad-diagonal; (2) the presentation of the idea in a non-rigorous way in the "Intuition of the theorem" part of Section 4; and (3) fixable mistakes in the proofs. For (1), we believe that such gut feeling might not be accurate. For (2), we did this for the purpose of isolating the important part of the argument. Those were to demonstrate ideas and are not part of the proofs. For (3), we thank the reviewer very much for pointing the mistakes out, but the mistakes are fixable and do not affect the overall correctness of the proofs.
>
> We now provide detailed explanations on the aforementioned issues and on how our manuscript has been revised.
>
> > Question/Comment 1: However, I do not believe the results are correct. There are some fatal and elementary errors and the paper shows no sign of proof-reading. First, the result does not pass an eye test. In [1, 2], the implicit bias of the diagonal variant of AdaGrad depends on various factors such as initialization. Therefore, it is very surprising that the more complex normed version variant would have an implicit bias that do not depend on such conditions.
>
> The algorithm we consider in our work is the Adagrad-Norm algorithm, which is different from the AdaGrad-diagonal algorithm. In this algorithm, all random gradient components share the same learning rate $-\frac{\alpha_{0}}{\sqrt{S_{n}}}$ during updates. Therefore, this algorithm, intuitively, is supposed to have the same implicit bias as GD and SGD. We specifically explained this on above Theorem 4.3 the reason AdaGrad-diagonal has a different implicit bias compared to AdaGrad-Norm.
>
> > Question/Comment 2: In the equation block in the "Intuition of the theorem" part of Section 4, what is the second gradient taken with respect to? If it is taken wrt $\theta_{n}$, then the author clearly forgot to apply the chain rule to $\theta_{n}/\\|\theta_{n}\\|$. Also, the value of  $i_{n}$ is in general not unique.
>
> The intuition block, as its name suggests, is just an intuitive understanding, and the expressions used here are not rigorous. For a rigorous derivation using the chain rule, please refer to Equation (41) in the appendix. Additionally, regarding $i_n$, it is indeed not unique. We deliberately skipped this detail in the intuition part. For a rigorous definition of $i_{n}$, please refer to step 2 in the proof sketch of Theorem 4.2, where $i_n$ is rigorously defined. We provide the complete process of the derivative here for your reference.
>
> $$\nabla \Bigg\\|\frac{\theta}{\\|\theta\\|}-\frac{\theta^*}{\\|\theta^*\\|}\Bigg\\|^2=2\bigg(\frac{\theta}{\\|\theta\\|}-\frac{\theta^*}{\\|\theta^*\\|}\bigg)^{\top}\nabla\frac{\theta}{\\|\theta\\|}=2\bigg(\frac{\theta}{\\|\theta\\|}-\frac{\theta^*}{\\|\theta^*\\|}\bigg)^{\top}\frac{\textbf{1}\\|\theta\\|-\frac{\theta\theta^{\top}}{\\|\theta\\|}}{\\|\theta\\|^{2}}=-2\Bigg(\frac{\frac{\theta^*\\|\theta\\|}{\\|\theta^*\\|}-\frac{\theta\theta^{\top}\frac{\theta^*}{\\|\theta^*\\|}}{\\|\theta\\|}}{\\|\theta\\|^{2}\}\Bigg)=2\frac{\theta-\frac{\theta^*\\|\theta\\|}{\\|\theta^*\\|}}{\\|\theta\\|^{2}}-\frac{\theta}{\\|\theta\\|^2}\Bigg\\|\frac{\theta}{\\|\theta\\|}-\frac{\theta^*}{\\|\theta^*\\|}\Bigg\\|^{2}.$$
>
> $$-\Bigg(\nabla \Bigg\\|\frac{\theta}{\\|\theta\\|}-\frac{\theta^*}{\\|\theta^*\\|}\Bigg\\|^2\Bigg)^{\top}\nabla g(\theta)=2\frac{1}{N}\sum_{i=1}^{N}\text{sgn}(y_{i}-0.5)f_{x_{i}}(\theta,x_{i})\Bigg(\frac{{\theta}^{\top}x_{i}-{(\frac{\theta^*}{\\|\theta^*\\|})^\top}x_{i}\\|\theta\\|}{\\|\theta\\|^{2}}-\frac{{\theta}^{\top}x_{i}}{2\\|\theta_{n}\\|^{2}}\bigg\\|\frac{\theta}{\\|\theta\\|}-{\hat{\theta}}^*\bigg\\|^{2}\Bigg)\le 2\frac{1}{N}\sum_{i=1}^{N}\text{sgn}(y_{i}-0.5)f_{x_{i}}(\theta,x_{i})\Bigg(\frac{{\theta}^{\top}x_{i}-{(\frac{\theta^*}{\\|\theta^*\\|})^\top}x_{i}\\|\theta\\|}{\\|\theta\\|^{2}}\Bigg).$$
>
>
> We wanted to note that we do agree that writing an informal equation, even in the intuition block, is not ideal. In fact, in the revised version, the intuition block specifically states the drop of the gradient chain and refers the reader to the appendix if they are looking for a full, rigorous statement. We thank the reviewer very much for pointing this out.

---

> ### Author Response · Authors · 2023-11-15
> **Author rebuttal (Part 2)**
>
> > Question/Comment 3: In the second equation of the proof of Theorem 4.3, the gradient of $\\|\theta\\|^\alpha$ is computed incorrectly, again seemed to be an omission of chain rule. Also, the dimensions in the middle and right expressions do not match.
>
> We did make a mistake in calculating the gradient of $\\|\theta\\|^\alpha$, and we thank the reviewer for pointing this out. Fortunately, this mistake is fixable in a few lines (revised in the updated manuscript) and does not affect the overall derivation of the proof.
>
> > Question/Comment 4:  Many quantities were not adequately defined before they were being used. For instance, Lipschitzness of $\nabla g$ was not mentioned until Section 4. In the same equation block I pointed to in my first comment, $k_n$ was not defined. In equation (3), $\hat{c}$ was not defined.
>
> We have made amendments to the definitions to improve the presentation. The revisions are highlighted in the revised pdf.
>
> > Question/Comment 5: The first two sentences in Section 4 seems to suggest that Theorem 4.1 is original, but it is in fact not. They need to be rephrased to not confuse the readers.
>
> It has been revised.
>
> > Question/Comment 6: Lemmas A.8 and A.10 where invoked in Section 3 without being stated first. And in Lemma A.8, the term "margin vector" was not defined.
>
> The term 'margin vector' refers to a vector that satisfies the conditions $\\{\theta\mid\text{sgn}(y_i-0.5)\cdot\theta^{\top}x_i>0\\}$. The definition has been updated in the manuscript.
>
> > Question/Comment 7: The set defined at the top of page 4 is not rigorous. In fact, the global minimum is simply not attainable.
>
> The global optimal points lie at infinitely far points in the directions parallel to at least one margin vector. This can be rigorously represented as follows:
> $$\lim_{\\|\theta\\|\rightarrow+\infty,\ \theta/\\|\theta\\|\in S}g(\theta)=0,$$ where $$S:=\\{\alpha\|\\|\alpha\\|=1,\ \text{sgn}(y_i-0.5)\cdot\alpha^{\top}x_i>0\\}.$$

---

> ### Comment · Reviewer_Ckt3 · 2023-11-17
>
> I would like to thank the authors for the prompt response.
>
> First, I would like to explain why I arrived at my strongly negative assessment of this paper. I primarily based my review on the obvious and elementary mathematical errors present in the intuitive block of Theorem 4.2 and on the second line of the proof of Theorem 4.3 (*which the authors do not contest*). Then, not passing my initial eye-test only reinforces my impression that the results are wrong. There are other issues with the clarity of the writing, which are less important but again showed that the authors did not carefully proof-read their submission.
>
> Regarding the revision, I would thank the authors for attempting to address all of the concerns that I listed. However, I find the changes as unsatisfactory.
>
> 1. Clarifications to the notations were added to where they first appeared instead of being centralized. Thus, I find the improvements to the readability of this paper to be inadequate. Also, I could not find any fixes to my 3rd comment (labeled as the 4th in your rebuttal).
>
> 2. While the revised version does now correctly note that the chain rule was skipped, there is no intuitive explanation on why this yields a good approximation.
>
> 3. I took a cursory look at the revised proof of Theorem 4.3 and did not notice any new errors. But half of the proof was changed and I am not going to call this as "a few lines." Plus, I am simultaneously in discussion with authors of several other ICLR submissions, so right now it is not possible for me to sit down and check the new proof line-by-line.
>
> Overall, I find that the authors only did the *minimal* changes to respond to my concerns. And given the poor state of the original submission, this is not giving me much confidence. I will remain recommending a **rejection** of this paper (but I did adjust the scores slightly) for the following reasons:
>
> 1. As I said before, I find some of the changes to be unsatisfactory.
>
> 2. It is my understanding that my review should primarily be based on the initial submission and the rebuttal is mainly for clarifications. The mathematical mistakes presented in the original submission were unacceptable and they will remain as the basis of my review. While the authors did propose fixes to these mistakes and they seem to be okay on a high level, the rebuttal period is much shorter than the review window and thus I cannot review these changes to the level of quality I like. Even if the new proofs are indeed correct, they still need to undergo a full review process from scratch.

---

> ### Comment · Reviewer_Ckt3 · 2023-11-17
>
> I just want to expand on my second point because it may sound too harsh.
>
> I want to emphasize that whether due to poor writing or actual mathematical mistakes, a good chuck of the technical portion of the original submission was incorrect and the readers may very well interpret those as fatal mistakes. This does raise *significant doubt* about the quality of the proofs for rest of the paper. And given that the revision were quite barebone and did not even touch some of my minor comments (which the authors *promised to change but did not deliver*), I feel that the authors still did not exercise due diligence and carefully check their proofs.
>
> Therefore, the only way for me to feel comfortable with the correctness of the results is if I do a careful read of this manuscript from the beginning to the end. Since this is clearly not practical given the amount allotted for rebuttal, I am not inclined to change my assessment in a major way. But I did lower my confidence by quite a bit, so hopefully the AC can properly take account of my concerns.

---

> ### Author Response · Authors · 2023-11-18
> **Author rebuttal (Part 3)**
>
> A detailed question regarding the constants was not answered in Parts 1 and 2 of the rebuttal. Here we wanted to supplement it.
>
> > Many quantities were not adequately defined before they were being used. For instance, Lipschitzness of $\nabla g$ was not mentioned until Section 4. In the same equation block I pointed to in my first comment, $k_n$ was not defined. In equation (3), $\hat c$ was not defined.
>
> Regarding $\hat{k}_{0},$ this constant is an existential constant. We only need to prove the existence of a constant $\hat{k}_0$ that satisfies $$\frac{{f_x}_i(\theta_n,x_i)}{{{f_x}_i}_n(\theta_n,{x_i}_n)}\le \hat{k}_0 \cdot \frac{1}{e^{({d_n,}_i-{{d_n,}_i}_n)(\\|\theta_n\\|+1)}}.$$ In fact, we have $$\frac{{f_x}_i(\theta_n,x_i)}{{{f_x}_i}_n(\theta_n,{x_i}_n)}=\frac{1+e^{\|\theta_n^\top {x_i}_n\|}}{1+e^{\|\theta_n^\top x_i\|}}\le \frac{1}{e^{\|\theta_n^\top {x_i}_n\|-\|\theta_n^\top x_i\|}}\cdot \frac{1+e^{-\|\theta_n^\top {x_i}_n\|}}{1+e^{-\|\theta_n^\top x_i\|}}\le \frac{1}{e^{\|\theta_n^\top {x_i}_n\|-\|\theta_n^\top x_i\|}}\cdot 2=2\frac{1}{e^{({d_n,}_i-{{d_n,}_i}_n)(\\|\theta_n\\|)}}=2\frac{1}{e^{({d_n,}_i-{{d_n,}_i}_n)(\\|\theta_n\\|+1)}}\cdot \frac{1}{e^{(-{d_n,}_i+{{d_n,}_i}_n)}}.$$ Since the dataset is given, there must exist an upper bound for $\frac{1}{e^{(-{d_n,}_i+{{d_n,}_i}_n)}}.$ Therefore, such a $\hat{k}_0$ must exist. Regarding $\hat{c},$ there was a typographical error in the original text. The correct notation of $\hat{c}$ should be $T_n.$ We have made the necessary correction in the revised version and indicated the location where $T_n$ is defined.

---

> ### Comment · Reviewer_Ckt3 · 2023-11-18
>
> Just to be clear, I had serious doubts about the mathematical formalism of the initial submission, and making piecemeal changes will not sway that opinion.
>
> Regarding small technical details and the writing, I already stated above that
>
> > Clarifications to the notations were added to where they first appeared instead of being centralized. Thus, I find the improvements to the readability of this paper to be inadequate.
>
> On top of this, the issues I pointed out in my 3rd and 5th comments are still not fully fixed. Plus, my list wasn't exhaustive -- I remember that there were a lot more typos, e.g. in the first line of the "intuition of the theorem" part on page 5, it should say "the direction of $\nabla \theta_n$" instead of "the direction of $\theta_n$." I simply do not feel that the paper is getting any more polished than before.
>
> Lastly, my last comment regarding the definition at the top of page 4 is still stands. The revised definition is still wrong, as minimizers simply do not exist. Also, following that definition, why do the authors claim that $\ell_2$ max margin direction has better generalization, as I recall this is still an open question and not even consistent with the empirical observations in [1]?
>
> EDIT: Another comment, the revised manuscript now exceeds the page limit. Reminder that the page limit is the same for initial submission and camera-ready (unlike NeruIPS).
>
> [1] H. Sun, K. Ahn, C. Thrampoulidis, and N. Azizan. Mirror descent maximizes generalized margin and can be implemented efficiently. Advances in Neural Information Processing Systems, 2022.

---

> > ### Author Response · Authors · 2023-11-18
> >
> > The global optimal point of this problem extends to infinity along the direction of a certain margin vector, as the function $y=1/x$. If we only consider $\mathbb{R}^d$ without adding the point at infinity, it indeed does not exist. Therefore, studying the limit of $\theta_n$ alone is meaningless because it tends to infinity. It is more meaningful to study the limit of the direction $\theta_n/\\|\theta_n\\|$.
> >
> > We made the definition in page 4 for the sake of convenience in understanding.

---

> > > ### Author Response · Authors · 2023-11-18
> > >
> > > As for the question of whether $L_2$ max margin can lead to better generalization, it is not the focus of our article. Our main focus is to demonstrate that AdaGrad-Norm can achieve $L_2$ max margin. The claim that $L_2$ max margin has better generalization is based on a previous paper [1], which points out that $L_2$ max margin explains why the predictor continues to improve even when the training loss is already extremely small.
> > >
> > > [1] Soudry D, Hoffer E, Nacson M S, et al. The implicit bias of gradient descent on separable data[J]. The Journal of Machine Learning Research, 2018, 19(1): 2822-2878.

---

> ### Author Response · Authors · 2023-11-20
>
> For the issue you mentioned about the unclear definition on page 4, we have made revisions in the updated version, and we have also added a property earlier to describe the Lipschitz continuity of $ \nabla g$. Additionally, regarding the $ \nabla\theta_{n}$ mentioned in the intuitive block, there is no problem with our description. We are referring to the change in the direction of $\theta_{n}$.

---

> ### Comment · Reviewer_Ckt3 · 2023-11-22
>
> Thank you for your responses.
>
> > Additionally, regarding the $\nabla \theta_n$ mentioned in the intuitive block, there is no problem with our description. We are referring to the change in the direction of $\theta_n$.
>
> You are right, this is my bad.
>
> > The claim that max margin has better generalization is based on a previous paper [1], which points out that $L_2$ max margin explains why the predictor continues to improve even when the training loss is already extremely small.
>
> I have read this particular reference (Soudry et al 2018) quite a few times, and I do not think they ever claimed that convergence to the $\ell_2$ max-margin is necessarily good for generalization. From my understanding of the implicit regularization literature, implicit regularization may explain generalization by controlling the "effective capacity" of the learned models [2]. From this perspective, there is nothing special about the $\ell_2$ max-margin other than that it happens to the implicit bias of gradient descent on linear classification. Plus, the second part of your argument could also be easily applied to another algorithm which converges towards max-magin wrt different norms, and this phenomenon has been observed in [3].
>
> While I understand that this is not a key point in your paper, I want to raise this issue because I have observed that your writing can be quite imprecise. If you cannot clearly explain a point and back it up with citation if necessary, then you should not be making such claim in the first place.
>
> > For the issue you mentioned about the unclear definition on page 4, we have made revisions in the updated version
>
> I read your revision. As we agree that global optima do not exist, then you still putting out this incorrect characterization in the revised definition?
>
> > We made the definition in page 4 for the sake of convenience in understanding.
>
> I do not like this explanation. If I see a equation block, then I assume it is a formal mathematical statement. I think writing down an informal statement without clearly advertising as such would only confuse the readers. Same for your intuition block for Theorem 4.2, which is still poorly explained.
>
>
> Finally, I checked the issues raised by Reviewer jNzZ. And quite a few of them are *still not addressed* in the latest revision. For example, issue #4, second part of #9, and #10 are unresolved despite them requiring very simply fixes.
>
>
> [1] Soudry, Daniel, et al. "The implicit bias of gradient descent on separable data." The Journal of Machine Learning Research 19.1, 2018.
>
> [2] Zhang, Chiyuan, et al. "Understanding deep learning (still) requires rethinking generalization." Communications of the ACM 64.3, 2021.
>
> [3] H. Sun, K. Ahn, C. Thrampoulidis, and N. Azizan. Mirror descent maximizes generalized margin and can be implemented efficiently. Advances in Neural Information Processing Systems, 2022.

---

> ### Comment · Reviewer_Ckt3 · 2023-11-22
>
> I found a little free time today and decided to take another look at the proofs. I found another elementary error within minutes: in the first line of equation (41) on page 23, the derivative $\nabla f$ is computed incorrectly. As this error was propagated through the whole equation, this is most likely not a typo. This is yet another time where I found an elementary mistake within the first few steps of a proof.
>
> **I think at this point, there is a clear trend that this paper contains an abundance of errors and thus the author MUST carefully revisit their proofs.**
>
> Combining with my observation that the authors have yet to fully address the issues raised by me and other reviewers (see above), I have no doubt that this paper **falls short** of the standards for publication. I recommend the authors to carefully proofread their work.

---

> > ### Author Response · Authors · 2023-11-22
> >
> > We have already modified the definition of the global optimum point in the revised version to its limit form. We have explained that when $θ_n$ approaches infinity along a certain margin vector, the loss function tends to zero, which is not contradictory to the non-existence of a global optimum point. If you find the statement about "infinity point" vague, we can replace it with a precise mathematical analysis in the revised version. Please refer to our latest revision. Regarding your concern about the lack of rigor in the 'intuitive block' equation, we are planning to remove the intuitive block module in the revised version to avoid misleading the readers.
> >
> >
> >
> > As for the issue you mentioned about equation (41), that is indeed a typo. We mistakenly wrote $\le$ as $=$. You can refer to our final revised version.

---

> ### Comment · Reviewer_Ckt3 · 2023-11-22
>
> > If you find the statement about "infinity point" vague, we can replace it with a precise mathematical analysis in the revised version. Please refer to our latest revision.
>
> The latest version finally looks correct. I am not sure why the authors were so hesitant to be rigorous in the "problem formulation" section, but at least I am happy this particular concern is resolved.
>
> > As for the issue you mentioned about equation (41), that is indeed a typo. We mistakenly wrote $\ge$ as $=$. You can refer to our final revised version.
>
> Okay. Good to hear that it is not major. As a comment to your revision, it may not clear to a reader as why the second term is negative. This is a very minor point, but I suggest you clarify this.
>
> However, I have other issues with equation (41). Your Lemma A.5 is concerned with *stochastic* Adagrad, and so the expectation is taken over the random variable $\nabla g(\theta_n, \xi_n) / \sqrt{S_{n-1}}$. Then why did you switch to *full-batch* Adagrad on the first line of equation (41) (note that the randomness $\xi_n$ in the gradient disappeared)? **It seems that your claim does not match what you are trying to prove.**
>
> **EDIT:** the issue above has been resolved.
>
>
> I thank the authors for their engagement throughout the discussion. However, resolving the issues regarding the quality of writing AND mathematical rigor has turned into a game of Whac-A-Mole. Since most of the mistakes were quite elementary or due to sloppy writing, my belief is that most of our conversation should have occurred between the authors themselves *before* the submission. Therefore, I am confident that there are many more errors waiting to be fixed.
>
> Overall, I maintain my original assessment that this manuscript is **far from ready** for publication. I again recommend the authors to **carefully proofread** their work.

---

> > ### Author Response · Authors · 2023-11-22
> >
> > We thank the reviewer for raising further questions. We believe (41) (42 in the updated manuscript) is correct. In fact, in probability theory, the expectation of conditional expectation is equal to the expectation. In fact, for any function $h(\theta_n),$ we have $$E(h(\theta_n)^\top\nabla g(\theta_n,\xi_n)/\sqrt{S_{n-1}})=E(E(h(\theta_n)^\top\nabla g(\theta_n,\xi_n)/\sqrt{S_{n-1}}\|F_n))=E(h(\theta_n)^\top E(\nabla g(\theta_n,\xi_n)/\sqrt{S_{n-1}}\|F_n))=E(h(\theta_n)^\top\nabla g(\theta_n)/\sqrt{S_{n-1}}).$$
> > Does this address the reviewer's question?
> >
> > Regarding why the second term of (41) (42 in the updated manuscript) is negative, we have revised the manuscript to add a line to explain this (highlighted in blue).

---

> > > ### Comment · Reviewer_Ckt3 · 2023-11-22
> > >
> > > > We believe (41) (42 in the updated manuscript) is correct.
> > >
> > > Thanks for the clarification. I did not expect that two steps were combined into one line.
> > >
> > > >  we have revised the manuscript to add a line to explain this (highlighted in blue).
> > >
> > > I believe that one very common way is to number each line and then explain the steps in words following the equation block. Another step that requires some explanation is the second last line of (42), where I believe you used the geometric interpretation of the dot product and then applied double angle formula?
> > >
> > > As about why I implored the authors to **proofread** their work, I actually noticed more errors on page 23 and waited to see if the authors could fix those on their own. Evidently that did not happen, so I will just list these errors and close the discussion.
> > >
> > > 1. Final inequality in equation (42) is incorrect because $\theta_n$ may not be a margin vector.
> > >
> > > 2. In the next equation block involving $H_n$, you switched between $\hat{\theta}^{* \top} x_i$ or $\hat{\theta}^{* \top} x_{i, n}$ for a few times. Which is the quantity you want? Former is the mathematically correct one, but it seems that you intend to use the later?
> > >
> > > 3. In equation (42), there are several places where you wrote $\theta$ instead of $\theta_n$.
> > >
> > > 4. In the statement of Lemma A.5, $f$ was referring to two unrelated functions, one of which was not undefined until the bottom of page 22.
> > >
> > > 5. The sentence "Absolutely, there exists "\hat{\delta}_0 > 0" is nonsensical because this variable is not mentioned elsewhere. I got confused here and so I stopped reading this proof.
> > >
> > > They are a mix of minor and more fundamental errors, but I will leave the fixes to the author. I want to emphasize that I am not a free editing service, so the authors should spend time debugging the proofs line-by-line. Based on the error/mistake density I observed, I think **rejection** is more than fair.
> > >
> > > P.S. I appreciate that the paper has in general improved since the authors began incorporating the reviewers' feedback. I wish the authors would bring similar effort into polishing rest of the paper.

---

> ### Author Response · Authors · 2023-11-23
>
> Thank you again for your careful inspection. Below, we will reply to each of your points.
>
> Question 1: Final inequality in equation (42) is incorrect because $\theta_n$ may not be a margin vector.
>
> Answer 1: In Theorem 4.1, we have actually proven that $\theta_n$ converges to a region composed of margin vectors. Therefore, this inequality holds for almost all $\theta_n$. However, in the revised version, we adopt a more rigorous statement. We construct a remainder term $$\beta_n:=\mathbb{E}\Bigg(\textbf{1}(\text{$\theta_{n}$ is not a margin vector})\frac{1}{N\sqrt{S_{n-1}}}\sum_{i=1}^{N}\text{sgn}(y_{i}-0.5)f_{x_{i}}(\theta_{n},x_{i})\frac{\\|{\theta_{n}}^{\top}x_{i}\\|}{2(\\|\theta_{n}\\|+1)^{2}}\bigg\\|\frac{\theta_{n}}{\\|\theta_{n}\\|}-{\hat{\theta}}^{*}\bigg\\|^{2}\Bigg),$$ and, through a simple derivation, obtain $$\beta_{n}\le\frac{N^2\max_{1\le i\le N}\\{\\|x_i\\|^2\\}}{2k_1^2\ln^2 2}\cdot \mathbb{E}\bigg(\frac{\\|\nabla g(\theta_{n})\\|^{2}}{\sqrt{S_{n-1}}}\bigg).$$ Then, according to Lemma A.4, we can obtain $\sum_{n=2}^{+\infty}\beta_n<
> +\infty.$ allowing us to incorporate this remainder term into $G_n$ in subsequent proofs without affecting the final result. You can refer to our latest revised version for details on the remainder term.
>
> Question 2: In the next equation block involving $H_n$, you switched between $\hat{\theta}^*\top x_i$
>  or $\hat{\theta}^{*\top}x_{i}$
>  for a few times. Which is the quantity you want? Former is the mathematically correct one, but it seems that you intend to use the later?
>
> Answer: Thank you for pointing that out. It was a small typo. We actually need the former statement, and we have already corrected it in the revised version.
>
> Question 3: In equation (42), there are several places where you wrote  $\theta$ instead of $\theta_n$.
>
> Answer: Indeed, we have already corrected it in the revised version.
>
> Question 4: In the statement of Lemma A.5, $f$ was referring to two unrelated functions, one of which was not undefined until the bottom of page 22.
>
> Answer: We have added the definition.
>
> Question 5： The sentence "Absolutely, there exists "\hat{\delta}_0 > 0" is nonsensical because this variable is not mentioned elsewhere. I got confused here and so I stopped reading this proof.
>
> Answer: This is a leftover term from the previous version of the proof. We have already removed it.

---

> ### Author Response · Authors · 2023-11-23
> **Paper length update**
>
> We thank the reviewer again for pointing out that the manuscript exceeds the max page limit in a previous comment. We have now updated the manuscript so it is now within the page limit. It is only the formatting that is updated while the content stays the same as the last revision.
>
> We are committed to clarifying any questions and making further improvements to the manuscript.

---

### Official Review · Reviewer_6py2 · 2023-11-03

**Soundness:** 3 good
**Presentation:** 2 fair
**Contribution:** 2 fair
**Rating:** 5
**Confidence:** 2

**Summary:**

For the stochastic AdaGrad-Norm equipped with am affine variance noise, the authors demonstrate that its almost surely convergence result to the L2 max-margin solution, considering the classification problem with the cross-entropy loss on a linearly separable data set.

**Strengths:**

For the stochastic AdaGrad-Norm equipped with am affine variance noise, the authors demonstrate that its almost surely convergence result to the L2 max-margin solution, considering the classification problem with the cross-entropy loss on a linearly separable data set.

**Weaknesses:**

Although the authors explore a broader noise model, the underlying motivation for the proposed model appears to be somewhat lacking in strength. The pertinent reference to the affine variance noise model is missing. It remains uncertain whether this noise model is practical within the simple model classification problem studied in this paper, and also the existing deep learning datasets or architectures.

The authors study the classification problem with the cross-entropy loss on a linearly separable data set, but the motivation of the paper is to understand the implicit bias of algorithms in training deep neural networks model, which has some extra benign testing phenomenon.

No numerical experiments are provided to complement the derived theory. Particularly, whether the intriguing phenomenon happens in Adagrad-Norm optimization on training deep learning model is unclear to me.

As claimed by the authors, for deterministic AdaGrad-Diagonal, (Soudry et al., 2018; Gunasekar et al., 2018; Qian & Qian, 2019) claim that it does not converge to the L2 max-margin solution as the non-adaptive methods do (e.g. SGD, GD). Thus, investigating the implicit bias of Adagrad-Norm may not be so important.

It looks to me proof techniques relies on (Jin et al. 2022), and (Soudry et al., 2018; Gunasekar et al., 2018; Qian & Qian, 2019; Wang et al. 2021b).  It would be beneficial to distinctly highlight the main novelties of their methods. Also the comparisons with the existing work (Wang et al. 2021b) is unclear to me.

**Questions:**

see the above

---

> ### Author Response · Authors · 2023-11-15
> **Author rebuttal (Part 1)**
>
> We thank the reviewer for the feedback. We are glad that the reviewer acknowledges our main contribution, which is to show AdaGrad-Norm's almost surely convergence result to the L2 max-margin solution, considering the classification problem with the cross-entropy loss on a linearly separable data set. We wanted to note that this problem is important, as reviewer  VfNS has pointed out, "Achieving convergence towards the maximum-margin vector is a crucial objective for the sake of generalization, as emphasized in prior research". Meanwhile, we wanted to clarify that this result was not available before, with either the affine variance noise or the mini-batch noise model.
> We are happy to provide detailed responses to the reviewer's questions and comments as below.
>
> > Question/Comment 1: though the authors explore a broader noise model, the underlying motivation for the proposed model appears to be somewhat lacking in strength. The pertinent reference to the affine variance noise model is missing. It remains uncertain whether this noise model is practical within the simple model classification problem studied in this paper, and also the existing deep learning datasets or architectures.
>
> As we have clarified, our main contribution is to prove the almost surely convergence of AdaGrad-Norm, which was not available before with any noise model. The result is new even with the regular mini-batch noise, which is a special case of our case. Immediately below Assumption 3.1, we stated "Regular Sampling Noise" to explain this.
>
> In terms of additional use cases of the affine variance noise, an example is the bounded gradient model, where $\\|\nabla g(\theta,\xi_n)\\|^{2} \le a.$ The case will be useful in reinforcement learning with bounded rewards, where the gradient is then bounded by a constant that is independent of $g$. The work in [1] is an example that uses the assumption.
>
> > Question/Comment 2: The authors study the classification problem with the cross-entropy loss on a linearly separable data set, but the motivation of the paper is to understand the implicit bias of algorithms in training deep neural networks model, which has some extra benign testing phenomenon.
>
> Our focus is on over-parameterization, so it is natural for us to choose a simple form of over-parameterization, such as logistic regression on separable datasets. Meanwhile, investigating over-parameterization is one of our motivations. Another motivation is to investigate whether stochastic adaptive learning rate algorithms exhibit the same implicit bias as conventional first-order algorithms. We believe the message our manuscript delivers is new to both the subarea of over-parameterization and that of stochastic adaptive learning rate algorithms.
>
> > Question/Comment 3: No numerical experiments are provided to complement the derived theory. Particularly, whether the intriguing phenomenon happens in Adagrad-Norm optimization on training deep learning model is unclear to me.
>
> The goal of this work is to better understand and characterize stochastic AdaGrad-Norm, and is not to develop a new optimization algorithm. The algorithm and empirical performance remain the same as in previous works and implementations. We therefore did not repeat the empirical study in our manuscript.

---

> ### Author Response · Authors · 2023-11-15
> **Author rebuttal (Part 2)**
>
> > Question/Comment 4: As claimed by the authors, for deterministic AdaGrad-Diagonal, (Soudry et al., 2018; Gunasekar et al., 2018; Qian & Qian, 2019) claim that it does not converge to the L2 max-margin solution as the non-adaptive methods do (e.g. SGD, GD). Thus, investigating the implicit bias of Adagrad-Norm may not be so important.
>
> We are discussing the stochastic AdaGrad-Norm algorithm rather than the deterministic AdaGrad-Norm algorithm. These two algorithms are quite different, especially in their convergence analysis. In this sense our result is important, and to some extent even surprising, because it shows the opposite statement given that deterministic AdaGrad-Diagonal has been proved to not converge to the L2 max-margin solution.
>
> Meanwhile, exploring the mathematical properties of stochastic adaptive algorithms is indeed a challenging problem, as highlighted in the difficulties (I) - (IV) mentioned in the first chapter of our manuscript. These difficulties are not present in deterministic adaptive algorithms. The challenges also highlight our contributions from the technical perspective.
>
> > Question/Comment 5: It looks to me proof techniques relies on (Jin et al. 2022), and (Soudry et al., 2018; Gunasekar et al., 2018; Qian & Qian, 2019; Wang et al. 2021b). It would be beneficial to distinctly highlight the main novelties of their methods. Also the comparisons with the existing work (Wang et al. 2021b) is unclear to me.
>
> In comparison to (Jin et al., 2022), the method used by them to handle last-iterate convergence cannot be applied to address implicit bias. The techniques in (Jin et al., 2022) become unsuitable due to the nuanced nature of our constructed Lyapunov function, i.e., $\\|\theta_{n}/\\|\theta_{n}\\|-\theta^*/\\|\theta^*\\|\\|^{2}$. In fact, the terms $\nabla(\\|\theta_{n}/\\|\theta_n\\|-\theta^*/\\|\theta^*\\|\\|^{2})^{\top}\nabla g(\theta_n,\xi_n)/\sqrt{S_{n}}$ and $\\|\theta_{n}/\\|\theta_{n}\\|-\theta^*/\|\theta^*\\|\\|^{2}$ lack a clear and evident quantitative relationship, making it difficult for us to obtain Equation (2) of our manuscript.
>
> In comparison to (Soudry et al., 2018; Gunasekar et al., 2018; Qian & Qian, 2019; Wang et al., 2021b), there are several major differences in the setting. First, the referred works do not consider randomness, which makes handling stochastic algorithms more challenging. Second, the referred works prove implicit bias by establishing the dual definition of the max-margin vector (i.e., the second definition of the max-margin vector in our paper), whereas we use the first definition to prove our results. From the technical perspective, our proofs are essentially different.
>
> [1] Kaiqing Zhang, Alec Koppel, Hao Zhu, and Tamer Basar. Global convergence of policy gradient methods to (almost) locally optimal policies. SIAM Journal on Control and Optimization, 58(6): 3586–3612, 2020.

---

> ### Author Response · Authors · 2023-11-22
> **Have we addressed your concerns?**
>
> We thank the reviewer again for your time and effort in reviewing our paper! As the discussion period is coming to a close, we would like to know if we have resolved your concerns expressed in the original reviews. We remain open to any further feedback and are committed to making additional improvements if needed. If you find that these concerns have been resolved, we would be grateful if you would consider reflecting this in your rating of our paper :)

---

### Meta-Review · Area_Chair_27r6 · 2023-12-12

**Metareview:**

Authors study stochastic AdaGrad-norm on a linear classification task. They show the implicit bias towards max-margin solution on linearly separable data.

This paper was reviewed by 4 reviewers and received the following Rating/Confidence scores: 5 / 2, 6 / 3, 6 / 2, 3 / 4.

I think the paper is overall interesting and provides good insight. However, two out of four reviewers found highlighted various weaknesses, specifically the proof techniques borrowed from other papers are not explicitly discussed. Another reviewer pointed to a few minor issues but argued that paper requires further proof-reading.

AC thinks that the paper has potential but requires significant revision, recommending reject for this ICLR.

**Justification For Why Not Higher Score:**

Paper requires significant revision before publication.

**Justification For Why Not Lower Score:**

n/a

---

### Decision · Program_Chairs · 2024-01-16

Reject